# The proteasome modulates endocytosis specifically in glomerular cells to promote kidney filtration

Wiebke Sachs[1,2], Lukas Blume [1,2], Desiree Loreth [1,2], Lisa Schebsdat[1,2], Favian Hatje[1,2], Sybille Koehler[2,3], Uta Wedekind[1,2], Marlies Sachs[1,2], Stephanie Zieliniski[1,2], Johannes Brand[1,2], Christian Conze [4], Bogdan I. Florea [5], Frank Heppner [6], Elke Krüger [7], Markus M. Rinschen [2,3], Oliver Kretz[2,3], Roland Thünauer[4,8,9] & Catherine Meyer-Schwesinger [1,2] ✉

Kidney filtration is ensured by the interaction of podocytes, endothelial and mesangial cells. Immunoglobulin accumulation at the filtration barrier is pathognomonic for glomerular injury. The mechanisms that regulate filter permeability are unknown. Here, we identify a pivotal role for the proteasome in a specific cell type. Combining genetic and inhibitor-based human, pig, mouse, and *Drosophila* models we demonstrate that the proteasome maintains filtration barrier integrity, with podocytes requiring the constitutive and glomerular endothelial cells the immunoproteasomal activity. Endothelial immunoproteasome deficiency as well as proteasome inhibition disrupt the filtration barrier in mice, resulting in pathologic immunoglobulin deposition. Mechanistically, we observe reduced endocytic activity, which leads to altered membrane recycling and endocytic receptor turnover. This work expands the concept of the (immuno)proteasome as a control protease orchestrating protein degradation and antigen presentation and endocytosis, providing new therapeutic targets to treat disease-associated glomerular protein accumulations.

The kidney assures adequate blood filtration and urine production within the glomerulus by the functional interplay of three resident glomerular cell types, namely visceral epithelial cells (podocytes), glomerular endothelial cells (GEnCs) and mesangial cells[1]. Parietal epithelial cells constitute the inner layer of Bowman's capsule which surrounds the glomerular convolute. The glomerular filtration barrier (GFB), which imparts both size- and charge-selective properties is composed of podocytes on the urinary side and GEnCs on the blood side. Both cell types are separated by the glomerular basement membrane[2]. Podocytes envelop the glomerular capillaries with an intricate network of major- and foot processes. Foot processes between neighboring podocytes interdigitate, to ultimately form a specialized form of cell-cell junction, the slit diaphragm[3]. To add to the high hydraulic conductivity and charge selectivity of the GFB, GEnCs are highly specialized with an elaborate fenestration and glycocalyx. Mesangial cells (MCs) are contractile cells that constitute the central stalk of the glomerulus to provide structural support and, as specialized pericytes, indirectly participate in filtration by reducing the glomerular surface area by contraction[4]. Under physiologic conditions, kidneys are perfused with ~680 ml blood plasma per minute to

[1]Institute of Cellular and Integrative Physiology, Center for Experimental Medicine, University Medical Center Hamburg-Eppendorf, Hamburg, Germany. [2]Hamburg Center of Kidney Health, Hamburg, Germany. [3]Nephrology, III Medical Clinic, Department of Internal Medicine, University Medical Center Hamburg-Eppendorf, Hamburg, Germany. [4]Leibniz Institute of Virology, Hamburg, Germany. [5]Bio-Organic Synthesis Group, Leiden University, Leiden, The Netherlands. [6]Institute of Neuropathology, Charité, Berlin, Germany. [7]Institute of Medical Biochemistry and Molecular Biology, University Medicine Greifswald, Greifswald, Germany. [8]Technology Platform Light Microscopy (TPLM), University Hamburg, Hamburg, Germany. [9]Advanced Light and Fluorescence Microscopy (ALFM) Facility at the Centre for Structural Systems Biology (CSSB), Hamburg, Germany. ✉e-mail: c.meyer-schwesinger@uke.de

generate 180 l of mostly protein-free urinary ultrafiltrate per day. Even though plasma is extremely rich in proteins[5], the deposition of protein along the GFB is neglectable, suggesting that mechanisms must exist to prevent abundant plasma proteins from clogging the GFB. Besides mechanic processes governing renal filter properties to protein[6,7], glomerular cell types themselves might contribute to glomerular protein clearance, such as MCs by phagocytosis[8,9] or podocytes by transcytosis[10].

Typical for glomerular injury is the intracellular[11] as well as the extracellular deposition of proteins. Injurious macromolecules originating from the plasma usually deposit in glomerular cell type specific subendothelial-, subepithelial- or mesangial patterns. The mechanisms underlying these pathologic protein depositions are elusive. As alterations in protein degradation through the ubiquitin-proteasome system (UPS) and the autophagosome-lysosome pathway (ALP) have been observed in glomerular injury[12–14], protein degradation systems might physiologically contribute to keeping the glomerulus free from injurious protein deposits. However, the homeostatic dependence of glomerular cell types on these two main protein degradation systems (Fig. 1A), especially the proteasome system, is unknown. As a major protein quality control system, the UPS comprises a cascade of E1-E2-E3 enzymes that ubiquitinate substrate proteins, as well as of

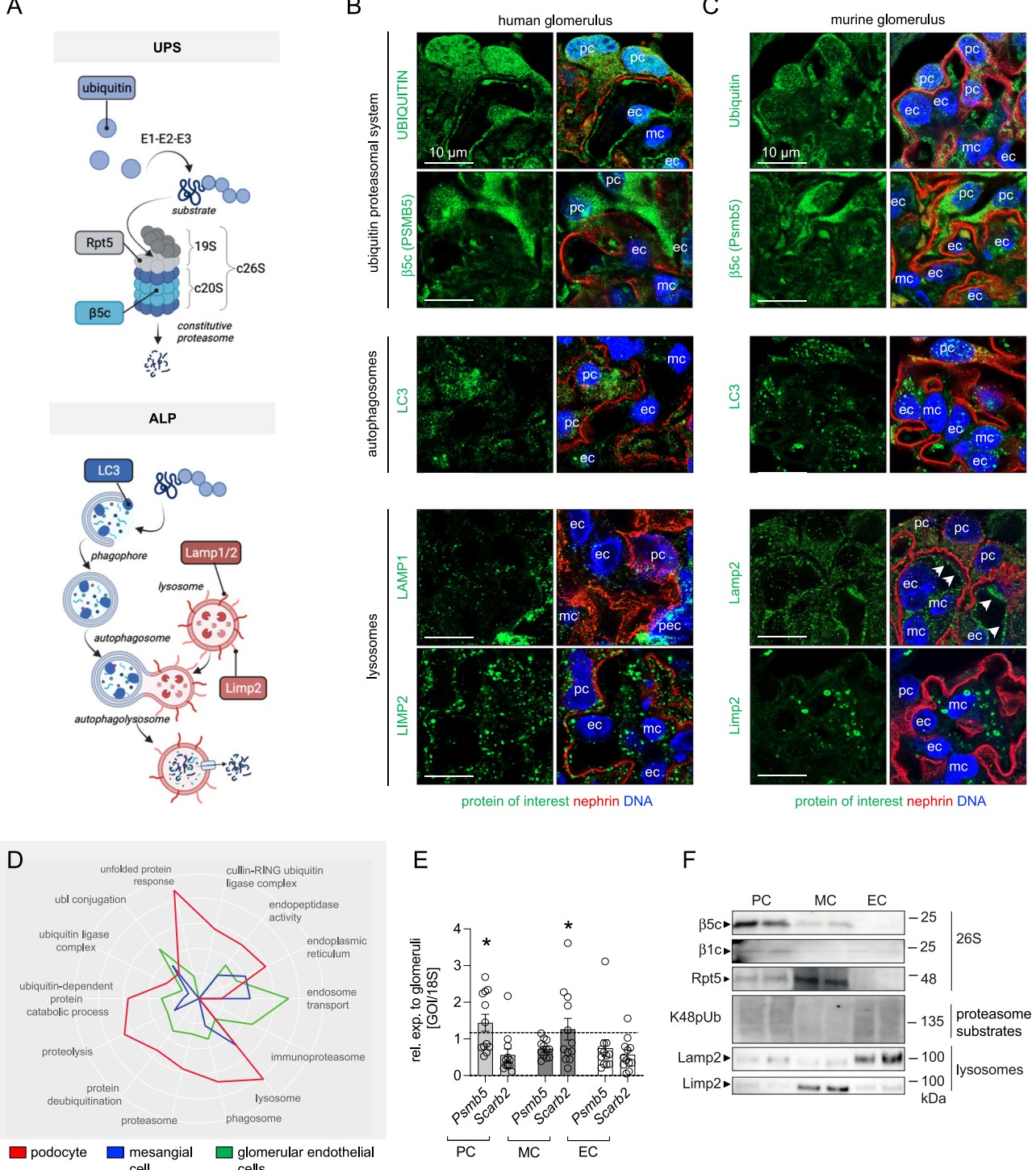

**Fig. 1 | Glomerular cell-specific distribution of proteasomal and lysosomal proteins in human and murine glomeruli. A** Schematic depiction of the ubiquitin proteasome system (UPS) and autophagosome-lysosome pathway (ALP) with individual marker proteins used for the expression analyses within glomerular cell types. Ubiquitin = polypeptide involved in dynamic regulation of protein function, localization, and stability; Rpt5 = proteasome regulatory subunit 6A (*Psmc3*) of the proteasome 19S regulatory particle; β5c (*Psmb5*) = main proteolytic subunit of the constitutive 20S core particle; LC3 = microtubule-associated protein 1A/1B-light chain 3; Lamp1 and Lamp2 = lysosomal-associated membrane proteins 1 or 2; Limp2 (*Scarb2*) = lysosomal integral membrane protein 2. Distribution of marker proteins (green) by high-resolution confocal images in a healthy human (**B**) and murine (**C**) glomerulus in relation to the slit diaphragm protein nephrin (red) and DNA (blue); pc podocyte, mc mesangial cell, ec glomerular endothelial cell, pec parietal epithelial cell, white arrows point toward endothelial lining filled with Lamp2-positive lysosomes, *n* = 3 individuals. **D–F** Podocytes (PC), mesangial cells (MC) and glomerular endothelial cells (EC) were bulk-isolated from glomeruli. **D** Proteomic analyses depict molecular properties of glomerular cell types as shown by the radar plot, whereby two-fold changes of distinct uniprot key words are plotted. Protein values were obtained by label-free quantification results using the MaxQuantLFQ algorithm[72]. **E** Relative transcript levels quantified via qRT-PCR of *Psmb5* (encoding for β5c) and *Scarb2* (encoding for Limp2) normalized to 18S as home keeper in relation to total glomerular transcript levels (dashed line), mean ± SEM, *\*p* = 0.0292 (PC *Psmb5*), *\*p* = 0.048 (MC *Scarb2*), one-way ANOVA with Bonferroni post-test for multiple comparisons, *n* = 12 of 2 pooled independent experiments. **F** Protein abundance from isolated glomerular cell types determined by immunoblot, equal loading was ensured by loading equal numbers of FACS-sorted PCs, MCs, and ECs, *n* = 3 independent experiments. Scheme was created with BioRender.com. Source data are provided as a Source Data file.

deubiquitinating enzymes which edit ubiquitin chains from substrate proteins. In general, ubiquitination determines the cellular fate of proteins including their activity, membrane localization, and/or their degradation by the proteasome[15]. The proteasome, as a complex protease, contains two general assemblies: a proteolytic chamber (20S core particle) harboring the proteolytically active β-subunits, and a regulatory particle (19S regulatory particle). The 20S and 19S particles are functionally linked by a gated protein translocation channel to form the 26S (20S associated with one 19S regulatory particle) or 30S (20S associated with two 19S regulatory particles) proteasome[13] for ubiquitin-dependent degradation. Degradation through the 20S proteasome is ATP and ubiquitin-independent[16]. The 20S core particle exists in multiple structural isoforms depending on the incorporated proteolytically active subunits. As such the constitutive proteasome harbors the β5c, β2c, and β1c subunits to form the constitutive 20S core particle (c20S), whereas the immunoproteasome replaces the proteolytic c20S subunits by the β5i (Lmp7), β1i, and β2i subunits upon IFNγ or TNFα stimulation to constitute the immuno-20S core particle (i20S). Further, hybrid constitutions between the c20S and i20S exist. Related to structural differences between the proteolytic constitutive- and immuno-β-subunits, different cleavage preferences are determined by the nature of the substrate specific pockets[17].

The proteasome is required for the controlled removal of over 80% of cellular proteins and hence is essential for the maintenance of a dynamic and balanced proteome. Despite its major importance for cellular proteostasis, our knowledge regarding the physiologic significance of the proteasome in glomerular cells is very scant even though proteome disbalances and pathological protein accumulations are known features of glomerular injury. The existence of complex differential proteostasis principles within the glomerular syncytium can be suspected. As such, resident glomerular cell types are differentially affected in genetic, metabolic, and inflammatory disorders that affect proteostasis, and this not only between cell types but also between species[12–14]. Understanding the physiology of glomerular cell proteasome processivity is urgently needed to drive our pathophysiologic understanding of glomerular injury leading to end stage renal disease and dialysis. Using techniques specifically established for these investigations, we systematically dissect the functional significance of the proteasome for glomerular proteostasis in a unique cell type specific approach. Our findings expand the concept of the proteasome as a control protease for protein degradation and antigen presentation to an orchestrator of endocytic protein clearance at the kidney filter.

## Results

### Differential expression of degradation systems in resident glomerular cell types

Our knowledge about the glomerular cell-dependence on proteostatic principles is scant. Therefore, comparative analyses of ubiquitin proteasome (UPS) and autophagosome-lysosome (ALP)

marker protein (Fig. 1A) expression patterns were performed in human and murine glomerular cell types. High-resolution confocal analyses revealed an accentuated abundance of ubiquitin in podocytes and GEnCs and of the main proteolytic β5c subunit of the c20S proteasome in podocytes, a finding appreciable in healthy human (Fig. 1B and Supplementary Fig. 1) as well as murine (Fig. 1C and Supplementary Fig. 2) glomeruli. The expression pattern of LC3-positive autophagosomes was balanced between all three resident murine glomerular cell types whereas in humans, podocytes exhibited a prominent LC3 staining pattern. Contrasting, the expression pattern of lysosome subtypes differed between species and glomerular cell types. Whereas murine MCs exhibited an accentuated expression of Limp2-positive lysosomes and GEnCs of Lamp2-positive lysosomes, podocytes exhibited a comparably low expression pattern for both lysosome subtypes. In human glomeruli, the expression pattern of LAMP1- and LIMP2-positive lysosomes was balanced between the three resident intraglomerular cells. Parietal epithelial cells exhibited a strong LAMP1 expression pattern.

As the distribution of ubiquitin and of the β5c-proteasome subunit was most prominent in human and murine podocytes, we refined our analyses by assessing the molecular properties of podocytes, GEnCs, and MCs in accessible human (KPMP, Kidney Precision Medicine Project, accessed 12/04/2023; https://www.kpmp.org) and published mouse glomerular single cell RNAseq databases[18] with focus on UPS transcripts (Supplementary Figs. 3 and 4) and by using a proteomic, qPCR, and immunoblotting approach in bulk-isolated murine glomerular cell types[19] (Fig. 1D–F). As illustrated within the radar plot (Fig. 1D), our analyses support the view of murine podocytes as cells with a high proteasome abundance. In contrast to GEnCs and MCs, murine podocytes exhibited enriched levels of a multitude of c20S proteasome subunits including Psmb5 (β5c), Psmb6 (β1c), and Psmb7 (β2c) in bulk proteomic data (Supplementary Fig. 3A) and single cell RNAseq analyses (Supplementary Fig. 3B), an expression pattern mostly transcriptionally substantiated in human glomerular cell types (Supplementary Fig. 3C). This preferential podocyte expression of standard proteasomal subunits was validated on the transcriptional level (*Psmb5*, Fig. 1E) and on the protein level by immunoblotting for β5c, β1c, and for the 19S regulatory particle subunit Rpt5 in mice (Fig. 1F). Glomerular cell types were loaded in a cell number adapted manner for comparison, as they do not express the same quantity of common housekeepers (Supplementary Fig. 5).

Corroborating the histological analyses, bulk proteomic analysis of ALP marker proteins also revealed a glomerular cell type distinct clustering in mice (Supplementary Fig. 6A) supporting the view of GEnCs as cells with endosomal abundance, and mesangial and podocytes as cells with abundance in lysosomes. This distribution pattern diverged on the single cell RNAseq level (Supplementary Fig. 6B, C). Nonetheless, the specific histologic distribution of Limp2 (abundant

within MCs and GEnCs) and Lamp2 (abundant within GEnCs) lyso-somes in mice could be validated on the transcriptional level for *Scarb2* (transcript of Limp2, Fig. 1E and Supplementary Fig. 6B) and on the protein level for Limp2 and Lamp2 (Fig. 1F). Single cell RNAseq rea-nalyzes substantiated the enhanced expression of *SCARB2* in human podocytes (Supplementary Fig. 6C).

In summary, human and mouse podocytes have a high constitutive proteasome protein abundance, in contrast to GEnCs and MCs. The glomerular cell distribution and abundance of lysosome subtypes (i.e., Limp2$^+$ lysosomes) varies between species. In the following investigations we focused on an in-depth dissection of the physiological significance of the proteasome system within glomerular cell types.

## Proteasome constitutions differ between podocytes and GEnCs

The proteasome is a highly versatile multi-subunit complex, which exists in different constitutions depending on the proteolytic sub-units incorporated within the 20S core particle, especially of the main proteolytic β5 subunits. Cells exposed to oxidative stress and/ or inflammatory stimuli have been shown to induce the immuno-proteasome (marked by the main proteolytic β5i subunit), which differs from the constitutive proteasome by an enhanced proteo-lytic efficiency and a different cleavage specificity, thus generating peptides best for MHC-I presentation[20]. Whereas podocytes exhibit a strong abundance of the constitutive proteolytic c20S β-subunits, our proteomic analyses marked a strong abundance of all proteo-lytic immunoproteasome β-subunits in GEnCs, namely of Psmb8 (β5i, Lmp7) as well as of Psmb10 (β2i) and Psmb9 (β1i) (Fig. 2A). The strong β5i abundance in murine GEnCs was quantitively validated by immunoblot (Fig. 2B) and qPCR (Fig. 2C) and localized to GEnCs in both human and mouse glomeruli (Fig. 2D). Analysis of single cell RNAseq datasets confirmed the preferential expression of proteo-lytic immunoproteasome subunits in murine and human GEnCs (Supplementary Fig. 3B, C). Together, these analyses depict a con-served dissection of β5c and β5i expression among podocytes and GEnCs across species. Since protein levels of the proteolytic β-subunits do not necessarily reflect their actual activity, we per-formed proteasomal subunit-specific activity measurements using activity-based probes in glomerular cell types. The pan-proteasome activity-based probe MVB003 (which binds to all proteolytic β-subunits and does not differentiate between β5c and β5i activity) depicted a prominent β5c/β5i-subunit activity in GEnCs followed by podocytes (Fig. 2E). GEnCs also exhibited activity of the β1i- and β2i-subunits, which was not present in podocytes or MCs. The use of the β5i-specific activity-based probe GB514 substantiated the high activity of β5i in GEnCs. To visualize the differing extent of pro-teasome activity between glomerular cell types in vivo, we estab-lished an ex vivo activity measurement assay applying MVB003 in pig glomeruli. Corroborating our findings in the lysate in-gel activities, proteasome activity was strongest in pig GEnCs fol-lowed by podocytes. MCs distinguished themselves by the absence of MVB003 signal (Fig. 2F, G).

Taken together our data show that within the glomerular syncy-tium GEnCs and podocytes have the strongest proteasome activity. Whereas podocytes exhibit mostly constitutive proteasome-related activity, GEnCs distinguish themselves with an additional immuno-proteasome activity.

## 20S processivity ensures morphologic and functional glo-merular filtration barrier integrity

The functional significance of the GEnCs-specific immunoproteasome activity was assessed by establishing an inducible endothelial cell-specific β5i (Lmp7) knockout mouse (Lmp7$^{\Delta EnC}$) (Fig. 3A). The suc-cessful endothelial induction of β5i-deficiency following tamoxifen administration was validated by the loss of β5i protein expression in glomerular endothelial cells by immunofluorescence (Fig. 3B) and

immunoblotting of bulk-isolated GEnCs (Fig. 3C). Strikingly, Lmp7$^{\Delta EnC}$-derived GEnCs exhibited an increased β5c protein abundance com-pared to littermate control cells (Fig. 3C, quantification lower graph), suggesting a compensatory upregulation of this constitutive protea-some subunit. We assessed whether endothelial β5i-loss affected sur-face expression of Major Histocompatibility complex class I (MHC-I), as constitutive Lmp7-knockout mice are known to exhibit a reduced surface expression of MHC-I due to altered peptide generation for MHC-I related antigen-presentation[21]. In line, inducible Lmp7$^{\Delta EnC}$-derived GEnCs exhibited a reduced MHC-I surface expression (Fig. 3D, quantification lower graph). Unexpectedly, despite an increased abundance of β5c, endothelial loss of the β5i-subunit resulted in strong phenotypic alterations. To this end, GEnCs isolated from Lmp7$^{\Delta EnC}$ kidneys exhibited an increased cell size compared to control GEnCs (Fig. 3E), which was not due to differences in the cell cycle state of the cells (Fig. 3F and Supplementary Fig. 7). Further, β5i-deficient GEnCs showed a loss of fenestration by ultrastructural analyses (Fig. 3G, quantification left lower graph). Interestingly, β5i-deficiency also affected general GFB morphology (Fig. 3G and Supplementary Fig. 8), as mild but significant podocyte foot process effacement (Fig. 3G, middle lower graph) and glomerular basement membrane splitting with electron densities and irregularity in thickness (Fig. 3G, right lower graph) were appreciated. GEnC β5i-deficiency did not translate to an enhanced abundance of proteasome substrates (lysine-48-polyubiquitinated proteins) by immunoblotting (Fig. 3H), sug-gesting an intact removal of proteins by the compensatory upregu-lated β5c subunit. Further, the observed GFB alterations in Lmp7$^{\Delta EnC}$ mice functionally did not translate to a measurable leakiness to albu-min (Fig. 3I), which might be related to an increased reabsorption of filtered albumin by proximal tubular cells.

In a next step, we established the significance of the proteolytic β-subunits for podocyte proteostasis. Genetic targeting of β5c is not feasible to date, therefore we set out to modulate constitutive β-subunit activities by chemical modulation using epoxomicin. This pan-proteasome inhibitor binds covalently to all proteolytic β-subunits, with a prevalence for the β5c- and β5i-subunits[22]. Based on our expression analyses, in vivo application of epoxomicin to mice inhibits the abundant proteolytic βc-subunits of the c20S in podocytes and the abundant proteolytic βi-subunits of the i20S in GEnCs. We also included comparative analyses of the lysosome system by using leupeptin A as a chemical inhibitor of lysosomal serine and cysteine proteases such as cathepsin B, H and L[23], hypothesizing that this should mostly affect MC proteostasis based on our histological and protein-biochemical findings. To this end, male BALB/c mice were successfully treated with the above-mentioned inhibitors over the course of 4 days (Supplementary Fig. 9), allowing us to dissect the differential effects of proteasome and lysosome inhibition on glomerular cell type proteostasis. No obvious symptoms such as diarrhea or weight loss were discernable in the 4 days of inhibitor treatment, mice with proteasome inhibi-tion were, however, less agile and exhibited dulled fur.

In general, both inhibitors did not alter general kidney function, as blood-urea-nitrogen (BUN) levels were not elevated compared to vehicle-treated mice (Fig. 4A). However, GFB functionality and mor-phology were affected by proteasome but not lysosome inhibition. As such, proteasome-inhibited mice exhibited a leakiness of the GFB to albumin (Fig. 4B), as well as an enlargement of their glomerular tuft area (Fig. 4C). Quantification of podocyte numbers per glomerular tuft area depicted the presence of a relative podocytopenia in pro-teasome inhibition, which related to the increase in glomerular tuft area in the setting of normal absolute podocyte number (Fig. 4C). Contrasting the normal light microscopic evaluations of periodic acid Schiff (PAS) stained kidney sections of leupeptin A-treated mice, epoxomicin-treated mice demonstrated subtle changes in glomerular morphology, such as a decrease in podocyte structural integrity, cell

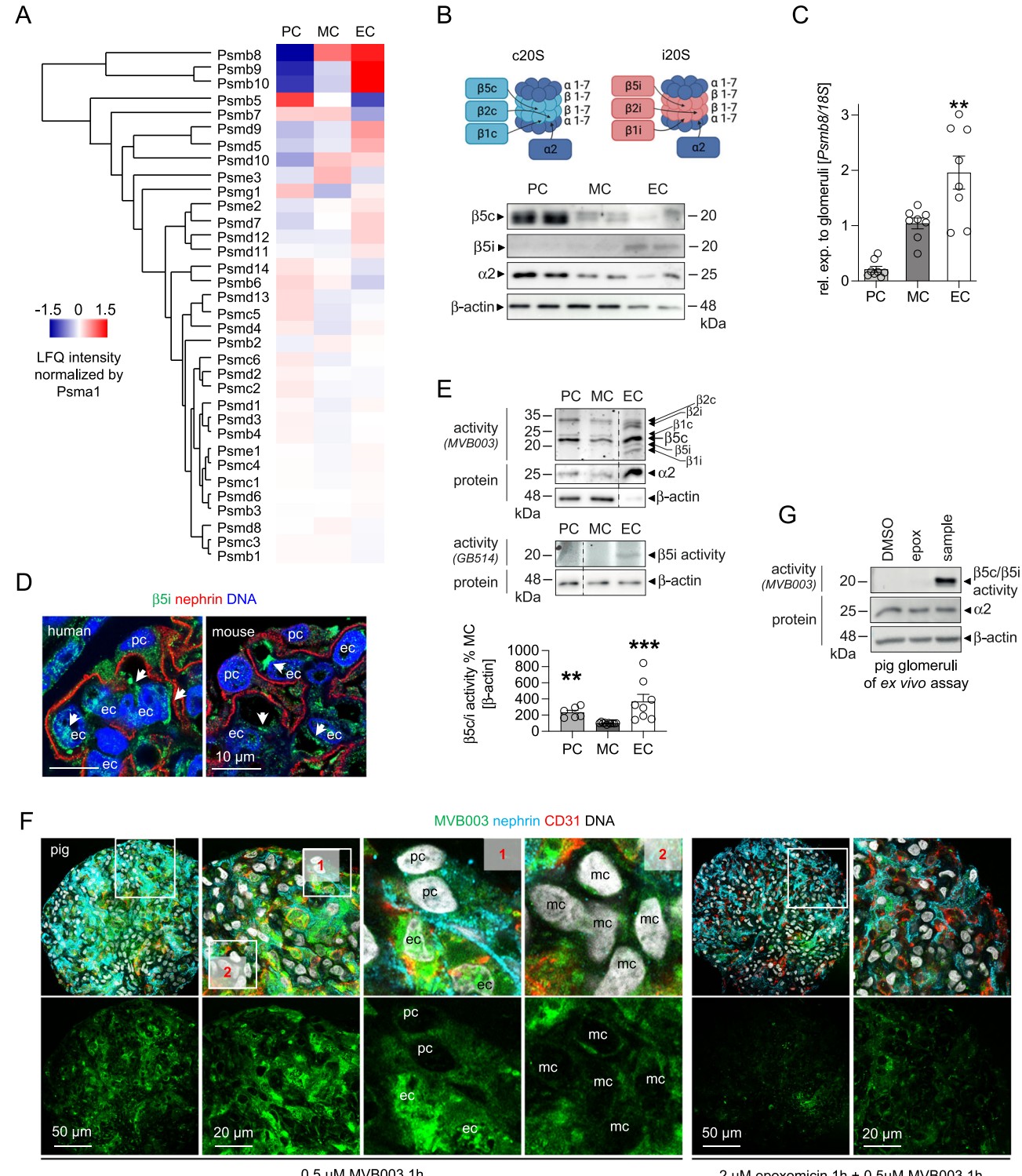

**D** β5i nephrin DNA

**F** MVB003 nephrin CD31 DNA

0,5 μM MVB003 1h          2 μM epoxomicin 1h + 0,5μM MVB003 1h

swelling, enhanced cytoplasmic vacuolization, and less defined cell borders (Fig. 4D). Ultrastructural investigations further confirmed that podocytes were affected by proteasome inhibition, as nephrin meanders were focally broadened by high-resolution microscopy (Fig. 4E) and foot processes focally effaced by electron microscopy (Fig. 4F, panel 1) in comparison to the normal glomerular ultrastructure in vehicle control mice (Supplementary Fig. 10A). Interestingly, nephrin meanders in leupeptin A treated mice were tighter than in vehicle-treated mice (Supplementary Fig. 10B). Further, supporting the ultrastructural GFB findings of Lmp7^ΔEnC mice, loss of endothelial fenestrations and focal splitting of the GBM were appreciated indicating

that proteasome inhibition did not solely affect podocyte but also GEnC proteostasis. Additionally, an accumulation of electron-dense material within the GBM was noted in proteasome-inhibited mice (Fig. 4F, panel 2). Strikingly, MCs exhibited no ultrastructural alterations upon proteasome inhibition (Fig. 4G), however, upon lysosome inhibition enlarged lysosomes with electron-dense storage material were found in MCs (Fig. 4H), while podocyte and GEnC ultrastructure was unaffected (Fig. 4I).

Together, these results demonstrate that inhibition of 20S proteasome processivity causes changes in podocyte and GEnC integrity leading to functional GFB impairment.

**Fig. 2 | Glomerular cell-specific proteasomal activities.** Glomerular cell types were analyzed for the main proteasomal β5-subunit abundance and activity. Micrographs were analyzed from 3 individual experiments, with 3 micrographs per group. **A** Proteomic label-free quantification of proteasome subunit abundance additionally normalized to Psma1 of the structural proteasome 20S core particle; podocytes (PC), mesangial cells (MC) and glomerular endothelial cells (EC). **B** Scheme depicting localization of the proteolytic β-subunits within the constitutive proteasome (c20S) and the immunoproteasome (i20S). Abundance of the c20S subunit β5c, of the i20S subunit β5i, and of the (c + i)20S subunit α2 from cell number adapted cells by immunoblotting. **C** Expression of *Psmb8* (encoding for the β5i) quantified via qRT-PCR, mean ± SEM, **$p = 0.0096$, one-way ANOVA with Bonferroni post-test for multiple comparisons, $n = 8$ of 2 independent experiments. **D** Distribution of the β5i-subunit (green) in healthy human and murine glomerular cells analyzed by high-resolution confocal images in relation to the slit diaphragm protein nephrin (red) and DNA (blue), white arrows point toward β5i expressing GEnCs (ec), pc podocyte, $n = 3$ individual experiments. **E** In-gel activity assay from cell number adapted cells determining the proteolytic c20S and i20S β-subunit activity using the pan-reactive activity-based probe MVB003 and the β5i-subunit-specific activity-based probe GB514. Immunoblots to α2 and/or β-actin from the same activity gels are shown. Graph: quantification of β5c/β5i activity via MVB003, mean ± SEM, **$p = 0.009$, ***$p = 0.0004$, one-way ANOVA with Dunn's post-test for multiple comparisons ($n = 6$ PC, MC, $n = 8$ EC). **F** Proteasomal activity assay in isolated pig glomeruli by ex vivo incubation with MVB003 for 1 h with or without the proteasome inhibitor epoxomicin. Glomerular cell types were demarcated using CD31 (red) for GEnCs and nephrin (turquoise) for podocytes, DNA (white); pc podocyte, mc mesangial cell, ec glomerular endothelial cell. Note strongest activity in GEnCs followed by podocytes (enlarged panel 1). MCs exhibit lowest proteasomal activity (enlarged panel 2). **G** In-gel activity measurement of the in (**F**) visualized pig glomerular samples to control for specificity of the (by confocal microscopy) depicted proteasomal activity. Scheme was created with BioRender.com. Source data are provided as a Source Data file.

## 20S alteration results in intra- and extracellular glomerular protein accumulation, which is not prevented by the autophagosome lysosome pathway

Alteration of 20S functionality in the "endothelial genetic ablation" and in the "global chemical" approach both led to an increase in cell/glomerular tuft size. As no obvious increase in glomerular cell number or cell cycle could be observed by histology, flow cytometry, or FACS-sort of glomerular cell types, this finding prompted us to hypothesize that the increase in cell/glomerular tuft size could be the result of protein accumulations due to degradative issues rather than cell proliferation. Indeed, our investigations were significant for an enhanced abundance of proteasome substrates (lysine-48-polyubiquitinated proteins) in glomeruli of proteasome-inhibited mice, which could be histologically localized specifically to podocytes in comparison to GEnCs and MCs (Supplementary Fig. 9B, C). Of note, our cumulative investigations indicate that proteostasis disturbances within podocytes of proteasome-inhibited mice were not compensated by the autophagosome-lysosome pathway (Supplementary Figs. 9 and 11). However, a firm conclusion, necessitates additional in vivo flux measurements in proteasome inhibited mice.

Intriguingly, besides the intracellular signs of altered glomerular proteostasis, an extracellular accumulation of immunoglobulin was present in glomeruli with impaired 20S function. Importantly, confocal investigations discerned mouse (ms)IgG deposition at the GFB with an accentuation underneath the slit diaphragm of podocytes only in proteasome-inhibited mice, whereas a mesangial msIgG accumulation was present in both proteasome- and lysosome-inhibited mice (Supplementary Fig. 12). As glomerular tuft swelling in the setting of glomerulonephritis is mostly the result of extracellular protein accumulation, especially of immunoglobulins[24], this observation requested further in-depth analyses. For this purpose, mice received "trackable" non-immune rabbit (rb)IgG and 7 days later were treated with either vehicle, epoxomicin, or leupeptin A for four consecutive days (Fig. 5A). Protein A/G resin-mediated precipitation of rbIgG from number-adapted glomeruli showed that a significant amount of rbIgG was deposited in glomeruli of epoxomicin- and to a lesser extent of leupeptin A-treated mice (Fig. 5B). Corroborating the msIgG deposition pattern, only epoxomicin treatment resulted in a strong linear accumulation of rbIgG in the subepithelial space of the GFB, with an accentuation in areas of the slit diaphragm, whereas mesangial rbIgG deposition was comparable between both inhibitors (Fig. 5C, D, panels 1, 2).

As proteasome inhibition resulted in subepithelial immunoglobulin accumulation, we further evaluated, whether β5i-deficiency in GEnCs also resulted in altered immunoglobulin clearance from the GFB. For these investigations, the amount of msIgG was quantified via immunoblot in bulk-isolated GEnC and non-endothelial control kidney cells from Lmp7^ΔEnC and control kidneys. A significant amount of msIgG was present in GEnCs derived from Lmp7^ΔEnC mice compared to controls (Fig. 5E). The msIgG accumulation localized within GEnCs when co-localization was performed with Lycopersicon esculentum lectin to demarcate the endothelial glycocalyx and to the podocyte foot process protein THSD7A (Fig. 5F).

In summary, our analyses show that 20S functionality impacts glomerular intra- and extracellular protein accumulation especially of podocytes and GEnCs, with impairment resulting in glomerular tuft and GEnC swelling, relative podocytopenia and proteinuria, which are not prevented by a compensatory autophagosome-lysosome upregulation.

## Proteasome functionality affects podocyte endocytosis

Immunoglobulin clearance from the GFB was affected in the setting of chemical and genetic proteolytic 20S modulation. We hypothesized that one underlying mechanism could be an altered endocytic protein clearance from the GFB. MCs are known to have a high phagocytic potential[4], however, are the glomerular cell type that depends least on proteasome function. GEnC endocytic/transcytotic activity of albumin through caveolae[25] is starting to be understood, and podocytes are known to exhibit an elaborate endocytic machinery related to neuronal cells[26]. This machinery is required for the turnover/recycling of proteins of the slit diaphragm such as nephrin[27], or of immunoglobulins[28], and for transcytosis of molecules such as albumin[10]. As proteasome impairment substantially affected podocytes and resulted in subepithelial IgG accumulation that was accentuated at the slit diaphragm, we assessed whether proteasome functionality impacted the endocytic propensity of podocytes. Indeed, pulse-chase experiments monitoring the uptake of different endocytic substrates by primary podocytes demonstrated a decreased uptake of transferrin (clathrin-mediated endocytosis[29]) and rbIgG (neonatal Fc receptor (FcRN) mediated endocytosis[28]) in the presence of epoxomicin-pretreatment (Fig. 6A and Supplementary Fig. 13). These findings were corroborated for caveolin-mediated endocytosis of FITC-albumin[30] in garland nephrocytes from *Drosophila* (Fig. 6B), which exhibit a high similarity in morphology and molecular make-up with mammalian podocytes[31] and are widely accepted as a model system to study podocyte endocytic mechanisms[32]. Pretreatment of nephrocytes for 15 min with epoxomicin reduced FITC-albumin endocytosis compared to vehicle, when visualized by confocal microscopy and quantified via the mean fluorescent intensity per nephrocyte area. Since proteasome inhibition affected different forms of podocyte endocytosis, live-cell imaging was performed in human podocytes to assess whether proteasome inhibition affected cellular uptake of plasma membrane, which was visualized after application of wheat germ agglutinin (WGA) to the culture medium to mark membrane glycoproteins (Fig. 6C and Supplementary Movies 1 and 2). To this end, a strong reduction of endocytic plasma membrane

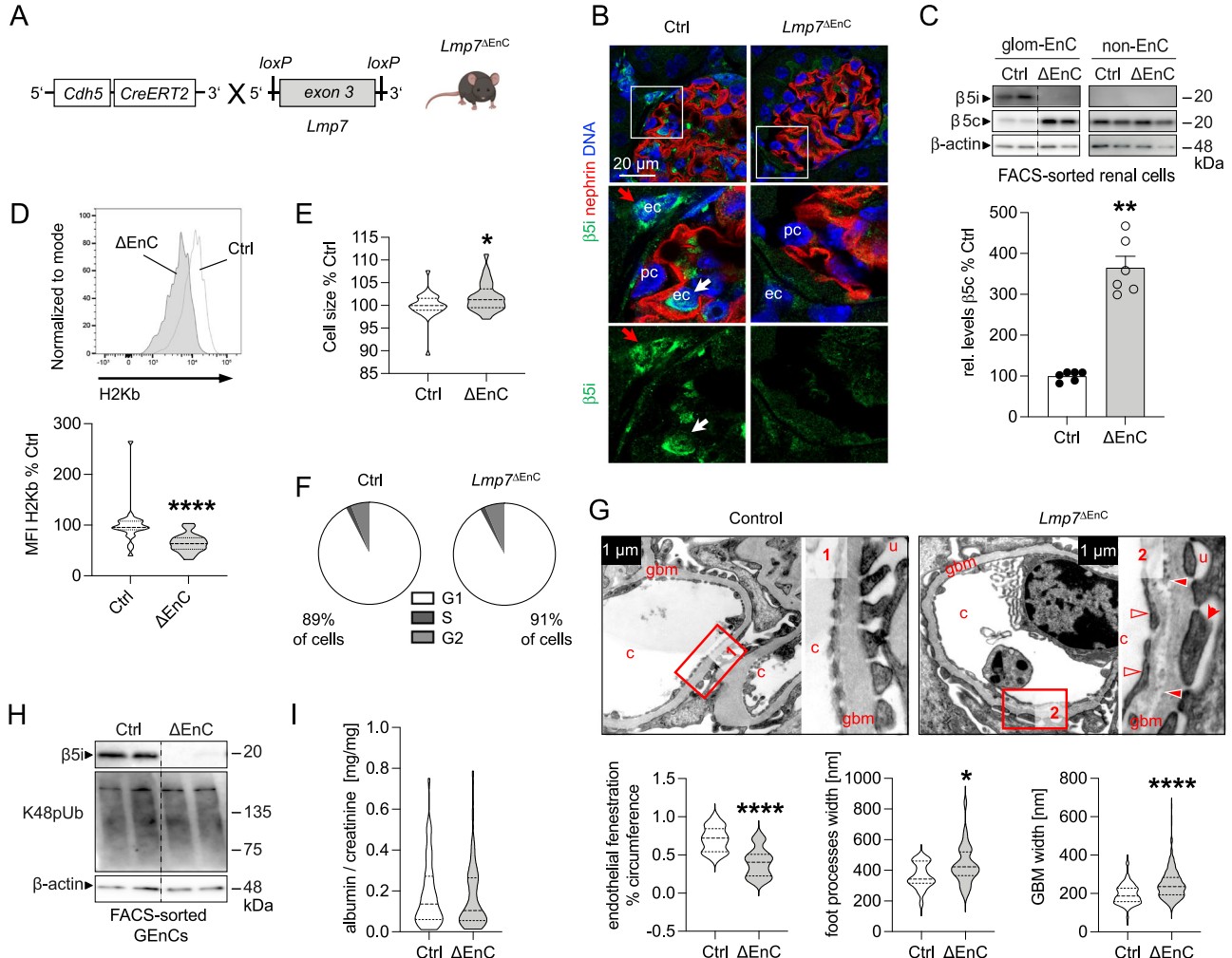

**Fig. 3 | Endothelial cell-specific β5i-deficiency results in morphological alterations of the glomerular filtration barrier.** Statistical analyses of all graphs: Two-sided Mann–Whitney $U$, mean ± SEM, $*p < 0.05$, $**p < 0.01$, $***p < 0.001$, $****p < 0.0001$, $n \geq 3$ per group, pooled values from 1 (EM) to 3 independent experiments (rest). **A** A tamoxifen-inducible endothelial cell-specific *Lmp7* (β5i) knockout mouse was generated with the cre-lox system. Naïve mice were analyzed 5–50 weeks after induction of knockout. **B** Knockout was evaluated by high-resolution confocal microscopy to β5i (green) in relation to nephrin (red) and DNA (blue) in glomeruli from *Lmp7*$^{\Delta EnC}$ and control littermates, arrows point toward β5i-expressing endothelial cells within (glomerular endothelial cell, white arrow) and outside (peritubular endothelial cell, red arrow) of the glomerulus. **C** Immunoblot of β5c and β5i abundance in bulk-isolated glomerular endothelial cells (GEnCs) in comparison to non-endothelial kidney cells (non-EnC) of the preparations from *Lmp7*$^{\Delta EnC}$ (ΔEnC) compared to control (Ctrl) kidneys; lower graph exhibits relative densitometric analysis of β5c levels in *Lmp7*-deficiency compared to control, $n = 6$ per group, $**p = 0.0022$. **D** Flow cytometry plot depicting surface levels of H-2Kb

MHC class I alloantigen in GEnCs in relation to controls. Lower graph exhibits quantification. **E** Cell size determination of GEnCs isolated by FACS-sort using the mean FSC-A, $*p = 0.0485$. **F** Cell cycle analysis with propidium iodide staining via flow cytometry in bulk-isolated GEnCs, pie charts: % of total cells in different cell cycle stages. **G** EM analyses exhibit loss of endothelial fenestrations (empty arrow heads, quantification lower left graph) in GEnCs, podocyte foot process effacement (red arrows, quantification lower middle graph), and focal splitting of the glomerular basement membrane with irregularity in thickness (filled arrow heads, quantification lower right graph) and electron dense depositions in *Lmp7*$^{\Delta EnC}$ mice; c capillary lumen, u urinary space, gbm glomerular basement membrane, overview in Supplementary Fig. 8, $*p = 0.0222$. **H** Immunoblot of lysine (K)48-poly-ubiquitinated proteasome substrates (K48pUb) in bulk-isolated GEnCs from *Lmp7*$^{\Delta EnC}$ (ΔEnC) compared to control (Ctrl) kidneys. **I** Albuminuria measured by ELISA to albumin and normalized to corresponding creatinine. Scheme was created with BioRender.com. Source data are provided as a Source Data file.

---

internalization was noted over 50 min of live-cell imaging in epoxomicin-treated human podocytes. Only a few large WGA-positive vesicles were formed with a strongly reduced intracellular motility compared to the abundant number of vesicles with vivid movement in the vehicle-treated podocytes. High-resolution confocal localization of the in vivo applied WGA together with filamentous (F)-actin and lysine-48 polyubiquitinated proteins (K48-pUb) after 60 min of live-cell imaging demonstrated that proteasome inhibition preserved a strong WGA signal at the plasma membrane and at thin processes, confirming the decreased uptake of WGA-bound glycoproteins of the plasma membrane. Additionally, an accentuated signal for proteasome substrates (K48-pUb) was present at the plasma membrane and at the thin

processes of proteasome-inhibited podocytes suggesting the presence of altered ubiquitination/degradation processes at the membrane (Fig. 6D).

Taken together, our results show that proteasome inhibition leads to an impairment of endocytic processes via a general alteration of plasma membrane dynamics in podocytes.

### Plasma membrane abundance of endocytic receptors is modulated by proteasome functionality

We then set out to identify whether the plasma membrane recycling of endocytic receptors related to protein/immunoglobulin clearance from the GFB was modulated by proteasome functionality. IgG

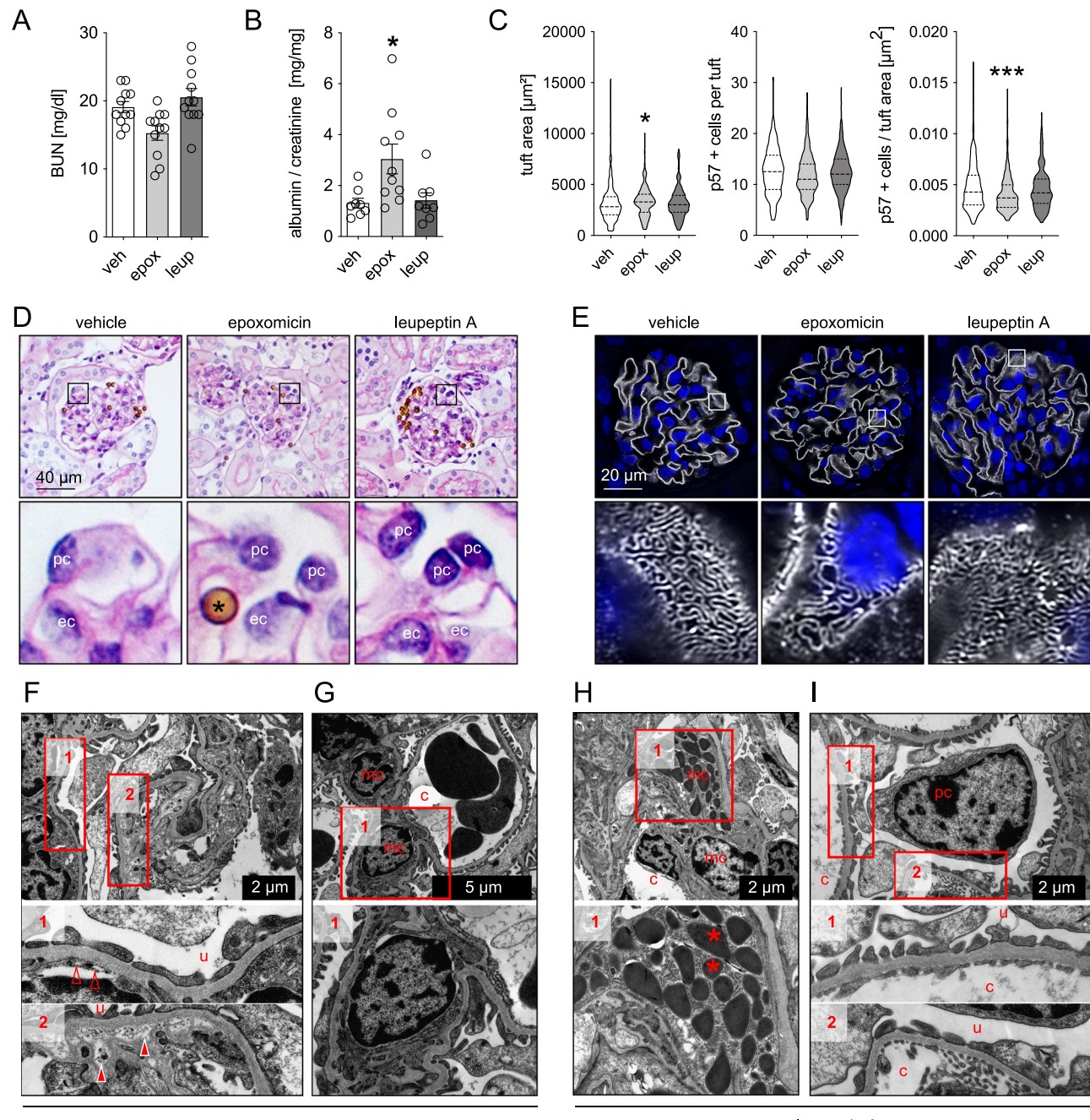

**Fig. 4 | Proteasome inhibition results in morphological and functional alterations of the glomerular filtration barrier.** Mice were treated with the irreversible proteasome inhibitor epoxomicin (0.5 µg/g bodyweight), the lysosomal inhibitor leupeptin A (40 µg/g bodyweight) or equal volumes of DMSO (vehicle, 125 µl) on four consecutive days. Urine, serum, and kidneys were collected and analyzed. Micrographs were analyzed from 3 individual experiments, with 3 micrographs per group. **A** Blood-urea nitrogen (BUN) measurement to assess renal function, $n = 11$ per group. **B** Albuminuria measured by ELISA to urinary albumin and normalized to corresponding creatinine to assess for glomerular filtration barrier functionality, $n = 8$ (veh, leup), $n = 10$ (epox). **C** Glomerular tuft area and absolute or relative podocyte number were quantified by immunohistochemical staining for the podocyte marker p57 in kidney paraffin sections ($n \geq 16$ per group). All shown statistical analyses: one-way ANOVA with Dunn's multiple comparison test,

mean ± SEM, *$p < 0.05$, ***$p < 0.001$, $n \geq 11$ per group, pooled data from 3 independent experiments. **D** Glomerular morphology assessed by PAS staining, *magnetic bead originating from glomerular isolation procedure, pc podocyte, ec glomerular endothelial cell. **E** High-resolution confocal micrograph of immunofluorescent staining for the slit diaphragm protein nephrin (white) and DNA (blue, Hoechst) to assess for podocyte foot process effacement. **F**–**I** Electron microscopical ultra-structural analyses exhibit focal podocyte foot process effacement and loss of endothelial fenestrations (empty arrow heads) as well as focal splitting of the glomerular basement membrane with accumulation of electron dense material (filled arrow heads) in epoxomicin-treated mice. c capillary lumen, u urinary space. **F, G** In leupeptin A treated mice (**H, I**) mesangial cells (mc) show electron dense lysosomal storage (asterisks). Source data are provided as a Source Data file.

turnover depends on the IgG interaction with Fc receptors. To this end, expression analyses of the Fc receptor transcripts in the different glomerular cell types demonstrated that only *Fcgrt*, *Fcgr1*, and *Lrp2* were expressed at a substantial level within podocytes, GEnCs, and

MCs (Supplementary Fig. 14A). The neonatal Fc receptor (FcRN, encoded by *Fcgrt*) is essential for endocytosis and recycling of IgG and albumin and has previously been involved in glomerular IgG accumulations in a constitutive[33] and a podocyte-specific[28] knockout

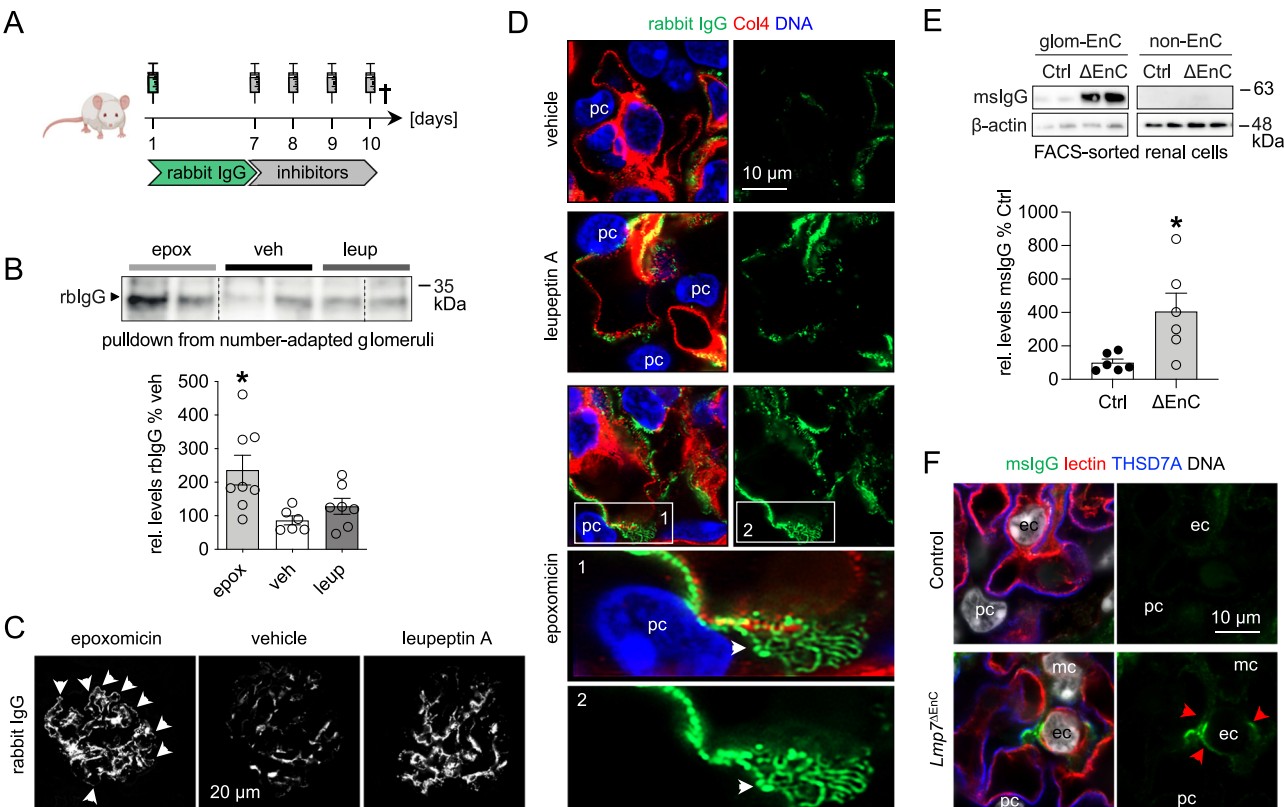

**Fig. 5 | Proteasome impairment results in glomerular IgG accumulation.**
**A** Scheme depicting the experimental setup. Mice were treated with the proteasome inhibitor epoxomicin (epox, 0.5 µg/g bodyweight), the lysosomal inhibitor leupeptin A (leup, 40 µg/g bodyweight) or equal volumes of DMSO (veh, vehicle, 125 µl) on four consecutive days following an initial injection with 160 µl of unspecific rabbit IgG (rbIgG) or PBS. Mice were perfused and kidneys were removed on day 10. Micrographs were analyzed from 3 individual experiments, with 3 micrographs per group. **B** Immunoblot quantification of deposited rbIgG following rbIgG-pulldown from isolated glomeruli. Graph exhibits densitometric analysis, mean ± SEM, $n = 6$ (veh), $n = 7$ (leup), $n = 8$ (epox), pooled data from 2 independent experiments, $*p = 0.013$, one-way ANOVA with Bonferroni's post-test for multiple comparisons. **C** Confocal micrographs depicting rbIgG (white) deposition pattern in glomeruli on day 10, note the linear deposition at the GFB (white arrows) of the epoxomicin-treated mouse. **D** High-resolution confocal micrographs resolving rabbit IgG (green) accumulation pattern in epoxomicin-treated mice at the GFB. White arrows point toward linear meandering rbIgG accumulations in the

subepithelial space. The glomerular basement membrane is marked by collagen type 4 (red) and DNA is depicted in blue (Hoechst). **E** $Lmp7^{\Delta EnC}$ and control littermates were analyzed after induction of β5i knockout. Following kidney perfusion with PBS, glomerular (GEnCs) and non-endothelial kidney cells were FACS-sorted. Immunoblot quantification of msIgG, densitometric analysis relative to control littermate cell populations, mean ± SEM, $n = 6$ mice of 2 pooled independent experiments, $*p = 0.0152$, two-sided Mann–Whitney $U$. Equal loading was ensured by loading equal numbers of FACS-sorted endothelial cell populations between genotypes. **F** Experimental kidneys of $Lmp7^{\Delta EnC}$ and controls were additionally perfused with AF647-Lycopersicon esculentum lectin (here: red) to demarcate the endothelial glycocalyx. Following zinc-fixation, sections were stained for msIgG (green) and the podocyte (pc) foot process protein THSD7A to enable localization of msIgG, DNA (white, Hoechst). Red arrows point toward msIgG accumulation underneath the glycocalyx within GEnCs (ec) in the $Lmp7^{\Delta EnC}$ mouse, mc mesangial cell. Scheme was created with BioRender.com. Source data are provided as a Source Data file.

approach. Even though total FcRN transcript and protein levels were not regulated in β5i-deficient GEnCs (Supplementary Fig. 14B, C), a prominent granular/vesicular cytoplasmic FcRN localization was histologically appreciated within Lmp7$^{\Delta EnC}$ GEnCs compared to control (Fig. 7A). Corroborating these findings, FcRN transcript levels within proteasome-inhibited glomerular cell types, as well as total glomerular FcRN protein levels were not affected (Supplementary Fig. 14D, E). Again, prominent FcRN accumulations were perinuclearly discernible by high-resolution microscopy in a vesicular/granular pattern especially in podocytes, and to a lesser extent in GEnCs (Fig. 7B). Revisiting FcRN protein abundance in bulk-isolated glomerular cell types demonstrated increased FcRN levels in proteasome-inhibited podocytes but not in GEnCs (Supplementary Fig. 14F). The in vitro pulse-chase experiments in primary podocytes of Fig. 6A additionally exhibited a perinuclear trapping of the FcRN in epoxomicin-exposed cells in comparison to vehicle-treated cells, where the FcRN trafficked to and from the cell border during the 60 min observation time (Fig. 7C). These data indicate that proteasome functionality affects IgG clearance from the GFB by altering FcRN recycling dynamics.

Besides the FcRN, we identified other, for the glomerulus hitherto unrecognized, endocytosis players to be preferentially abundant in podocytes on the protein (Supplementary Fig. 14A) and transcriptional (Supplementary Figs. 15B, C and 16) level. As such, especially the C-type mannose receptor 2 (Mrc2; also known as uPARAP, Endo180, or CD280) exhibited a substantial transcriptional regulation in podocytes and GEnCs in the setting of proteasome inhibition, with *Mrc2* mRNA being reduced in podocytes and increased in GEnCs (Supplementary Fig. 16C). Mrc2 is an endocytic receptor involved in extracellular matrix (ECM) remodeling through the uptake of collagen, large collagen fragments[34] and of thrombospondin 1[35]. Our evaluations substantiated the dependence of glomerular Mrc2 protein abundance on proteasomal functionality, as transcript and protein levels of Mrc2 were reduced in proteasome-inhibited glomeruli (upper blot Fig. 7D, quantification upper graph), which localized to the cell body and foot-processes of proteasome-inhibited podocytes (Fig. 7E). The essential podocyte protein nephrin, on the other hand, whose slit diaphragm localization is extensively regulated by endocytic recycling[32], showed no

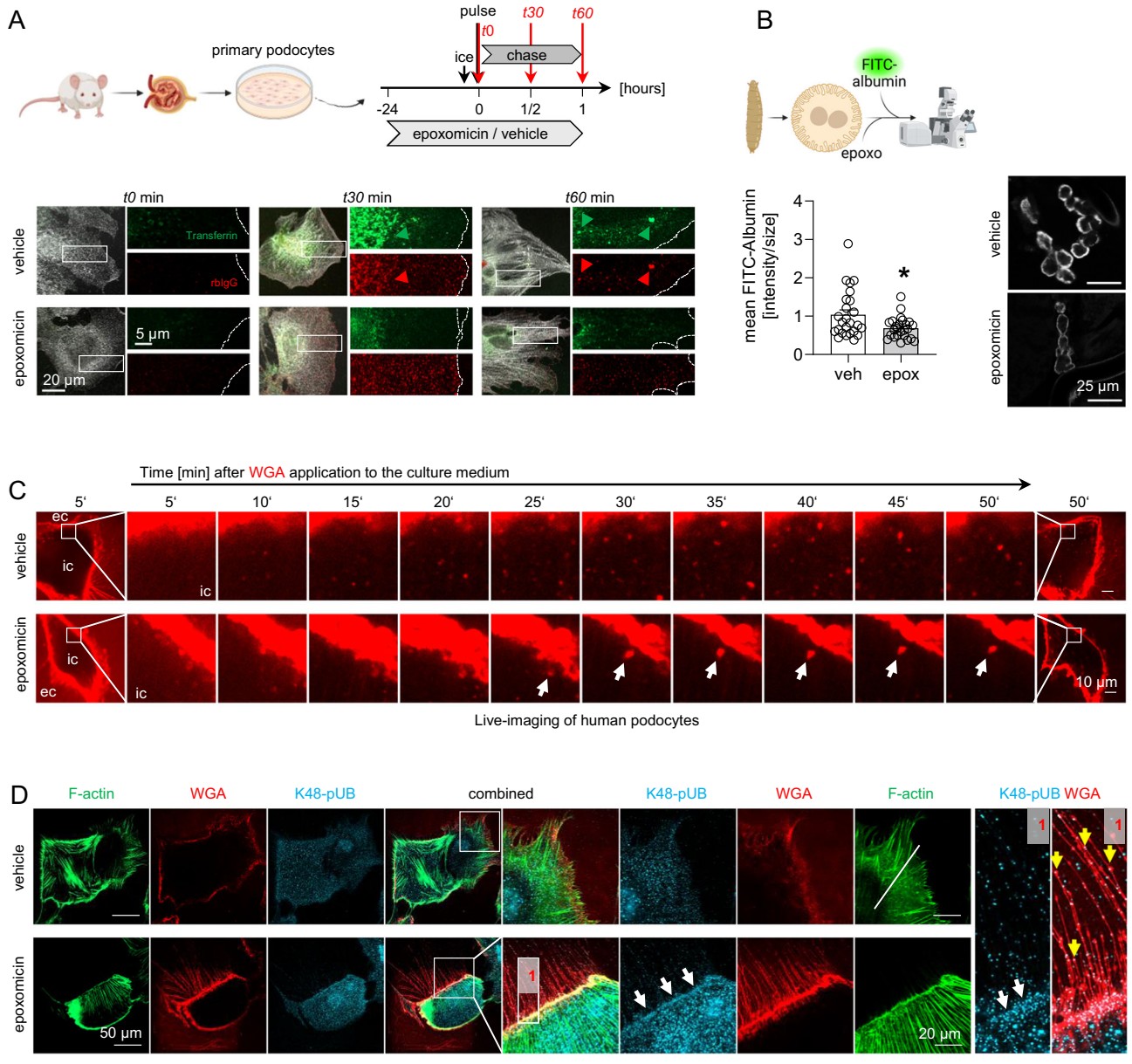

**Fig. 6 | Proteasome functionality affects podocyte endocytosis.** Micrographs were analyzed from 3 individual experiments, with 3 micrographs per group. **A** Primary podocytes were outgrown for 5 days and assessed for their endocytic activity. Cells were pretreated with either 10 nM epoxomicin or DMSO for 24 h, synchronized on ice prior to the addition of endocytic substrates. Cells were harvested after time point (*t*) 0, 30, and 60 min, fixed with 4% PFA and stained for synaptopodin (white) to demarcate podocytes. Internalization of FITC-transferrin (green) and Cy3-rabbit IgG (rbIgG, red) was assessed by immunofluorescence. Green and red arrows point toward endocytic cargo that was taken up; the dotted line indicates the cell border. **B** FITC-albumin endocytosis was assessed in isolated garland nephrocytes from *Drosophila* after pretreatment with 2 μM epoxomicin (epox) or equivalent amounts of DMSO (veh). High-resolution confocal images of FITC-albumin in nephrocytes, graph exhibits quantification of relative mean fluorescent intensity per nephrocyte area to vehicle, mean ± SEM, pooled data of 3 independent experiments with *n* = 25 per group, *\*p* = 0.0409, two-sided Mann–Whitney *U*. **C** Human podocytes were pretreated with 1 μM epoxomicin or

equivalent amounts of DMSO (vehicle) for 6 h. Wheat germ agglutinin (WGA)-rhodamine was applied at timepoint 0 min to the medium to mark glycoproteins of the plasma membrane. Time-lapse live-cell images were taken at 37°C and 5% $CO_2$ using a Nikon Spinning Disc microscope, a frame every 5 min is shown (live-imaging films are within the supplement). Note the reduced appearance of WGA-positive vesicles in epoxomicin pretreatment. The large vesicle depicted (white arrow) exhibits a reduced motility, ec extracellular space, ic intracellular space. **D** Following live-cell imaging, human podocytes were fixed and processed for high-resolution confocal microscopy for localization of the in vivo applied rhodamine-WGA (red) in conjunction with filamentous (F)-actin (green) and lysine (K)-48-polyubiquitinated proteins (K48-pUb, turquoise). Note the strong WGA signal as well as the accentuated signal for K48-pUb at the plasma membrane (white arrows) and at the thin processes (yellow arrows in [1]) of the epoxomicin-exposed human podocyte. Scheme was created with BioRender.com. Source data are provided as a Source Data file.

alterations on the total protein level (lower blot Fig. 7D, quantification lower graph) or by immunofluorescence (Fig. 7E) in proteasome inhibition. Using an established method to determine endocytosis by in vivo biotinylation of plasma membrane proteins

in mice[36], we could discern reduced abundances of both Mrc2 and nephrin within the biotinylated glomerular plasma membrane protein fraction of proteasome-inhibited mice (Fig. 7F), demonstrating a reduced in vivo plasma membrane-expression of Mrc2

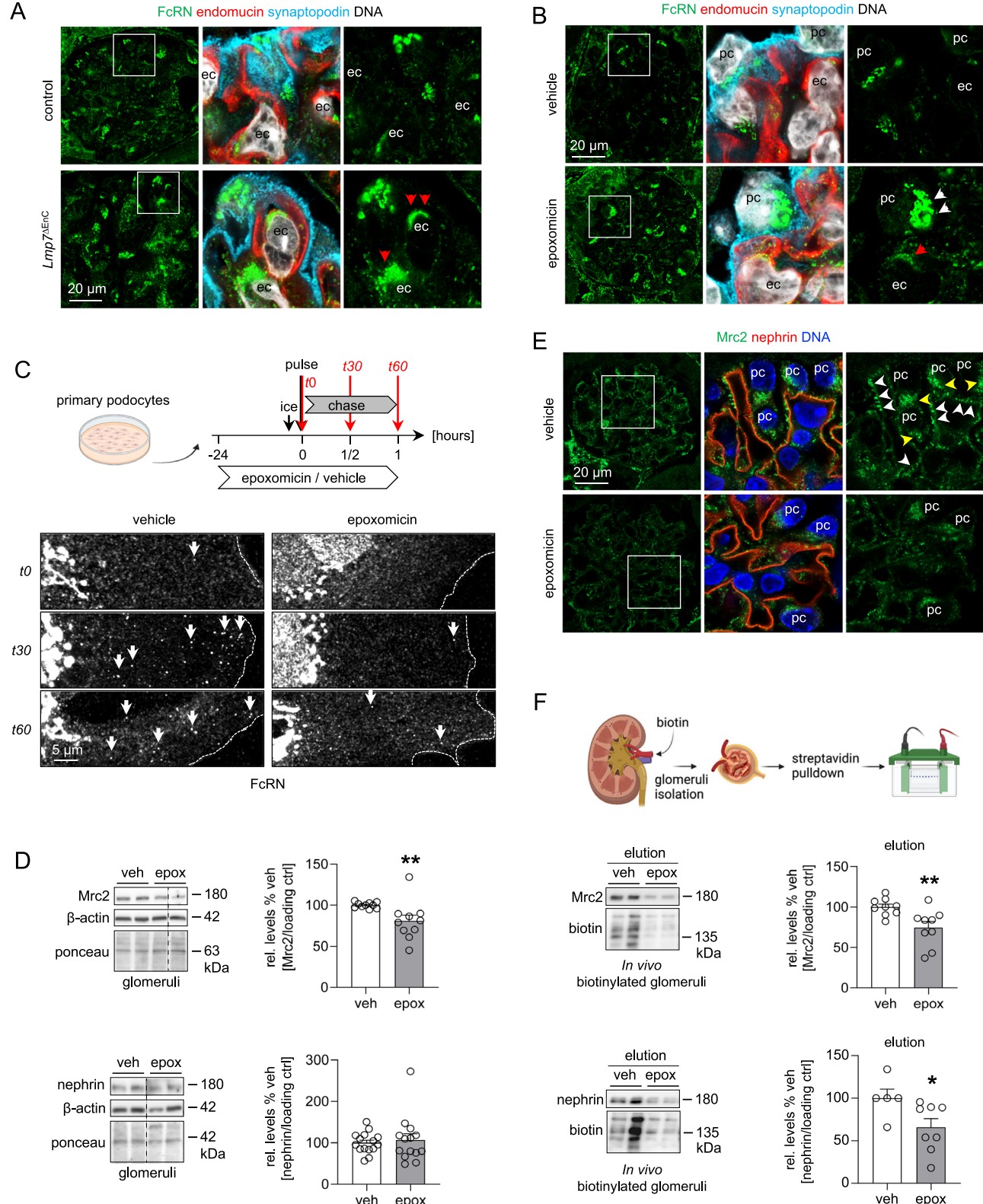

and nephrin in podocytes. These results demonstrate that proteasome inhibition affects endocytosis of GEnC and podocytes in a complex manner, by i.e., altering total and plasma membrane levels of specific endocytic receptors, as well as their trafficking.

Taken together, our study establishes the proteasome as a central regulator of GFB functionality by orchestrating the endocytic recycling of membrane proteins such as endocytic receptors. As summarized in Fig. 8B, modulation of β5i-subunit expression in GEnCs affects the

morphology of GEnCs as well as of podocytes and the GBM. Apart from accumulating immunoglobulins intracellularly, GEnCs exhibit decreased MHC-I surface levels and an intracellular retention of FcRN showing an impairment in membrane receptor recycling. Global proteasome inhibition as summarized in Fig. 8C results in functional impairment of the GFB. Podocyte proteostasis is severely affected with intracellular proteasome substrate accumulation and extracellular immunoglobulin accumulation at the slit diaphragm. Besides foot

**Fig. 7 | Plasma membrane abundance of endocytic receptors is modulated by proteasome functionality.** Micrographs were analyzed from 3 individual experiments, with 3 micrographs per group. High-resolution confocal micrographs of glomeruli from (**A**) $Lmp7^{\Delta EnC}$ and control kidneys after induction of β5i-knockout or from (**B**) BALB/c mice treated with epoxomicin or vehicle for 4 consecutive days demonstrates abundant FcRN (green) expression in GEnCs (ec, endomucin (red)) in a vesicular/granular pattern (red arrows) and in podocytes (pc, synaptopodin (turquoise)) in a perinuclear vesicular/granular pattern (white arrows), DNA (white, Hoechst). **C** Primary podocytes were assessed for FcRN trafficking, experimental procedure depicted in the scheme (corresponding panels for rbIgG and transferrin Fig. 6A). Cells were harvested after time point ($t$) 0, 30, and 60 min, fixed with 4% PFA and stained for FcRN. White arrows: FcRN trafficking to or from the plasma membrane, the dotted line indicates the cell border. **D–F** Glomeruli and glomerular cell types of epoxomicin- compared to vehicle-treated mice were evaluated for protein abundance of Mrc2 and nephrin. **D** Immunoblot quantification of glomerular Mrc2 and nephrin protein levels to vehicle, mean ± SEM, $n = 10$ (Mrc2),

$n = 15$ (nephrin veh) $n = 14$ (nephrin epox) mice, data from 5 independent experiments, **$p = 0.0014$, two-sided Mann–Whitney $U$ test. **E** High-resolution confocal micrographs of glomeruli derived from proteasome-inhibited mice depicting the localization of Mrc2 (green), nephrin (red) and DNA (blue, Hoechst), note the prominent Mrc2 expression within the podocyte (pc) cytoplasm (yellow arrows) and at foot processes lining and forming the GFB (white arrows) in the vehicle mouse, which is reduced upon proteasome inhibition. **F** In vivo biotinylation of glomeruli from epoxomicin- or vehicle-treated mice after streptavidin pulldown of biotinylated membrane proteins. Immunoblots depict Mrc2 and nephrin abundance within the biotinylated plasma membrane protein fraction, Mrc2 and nephrin densitometric analysis normalized to the amount of total precipitated biotinylated proteins, mean ± SEM, $n = 9$ (Mrc2), $n = 5$ (veh nephrin), $n = 8$ (epox nephrin) mice, pooled data from 2 independent experiments, *$p = 0.0179$, **$p = 0.0078$, two-sided Mann–Whitney $U$ test. Scheme was created with BioRender.com. Source data are provided as a Source Data file.

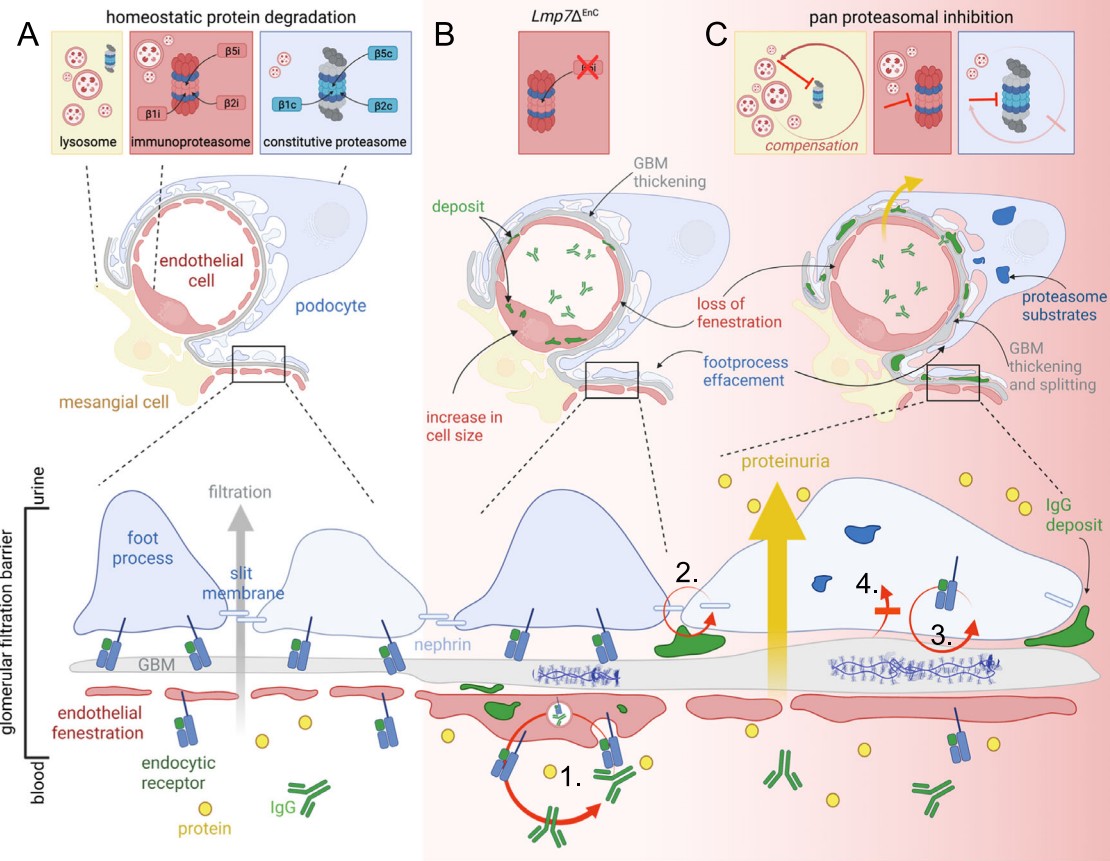

**Fig. 8 | Summary of the effects of 20S proteasome modulation in glomerular filtration barrier functionality. A** Glomerular cell type proteostasis differentially depends on the proteasome and lysosome system. Conserved between species, podocytes and glomerular endothelial cells exhibit a prominent expression and activity of the proteasome. Proteasome constitution differs: Podocytes exhibit a prominent abundance and activity of the constitutive (c20S) proteasome, whereas GEnCs exhibit a prominent abundance and activity of the immuno (i20S) proteasome. Varying between species, abundance of lysosome subtypes (Limp2 versus Lamp2-positive) differs. As a common theme, lysosome abundance is highest in mesangial cells and GEnCs. **B** Modulation of β5i-subunit expression in GEnCs affects the morphology of GEnCs as well as of podocytes and the GBM. GEnC cell size increases, fenestrations are lost and molecules such as IgG accumulate intracellularly. The GBM is thickened and focally split. Podocyte foot processes are effaced.

GEnCs exhibit decreased MHC-I at the surface and altered expression of FcRN (endocytic receptor to IgG and albumin) by histology but not total protein level, suggesting alterations in membrane protein trafficking (1.). **C** Global inhibition of the constitutive as well as of the immune β-subunit activities results in proteinuria. MCs show little alterations, whereas podocytes, GEnCs, and GBM are affected. Podocytes accumulate proteasome substrates intracellularly, and immunoglobulins in the subepithelial space accentuated at the slit diaphragm. The GBM is thickened and split. GEnCs lose their fenestration. Surface levels of the slit diaphragm protein nephrin (2.) and of the endocytic receptor to collagen Mrc2 (3.) are decreased, as is the uptake of substrates by endocytosis (4.). Podocyte plasma membrane dynamics and vesicle movement are reduced. Proteasome impairment in MCs is compensated by the lysosomes, compensation of proteasome impairment in podocytes by lysosomes is scant. Scheme was created with BioRender.com.

process effacement, the GBM is thickened and split, and GEnCs lose their fenestration. Surface levels of the slit diaphragm protein nephrin and of the endocytic receptor to collagen Mrc2 are decreased, as is the uptake of substrates by endocytosis. Podocyte plasma membrane dynamics and vesicle movement are reduced, demonstrating that proteasome functionality plays a crucial aspect in endocytic/recycling processes of GEnCs and podocytes which are essential for GFB permeability and maintenance. Proteasome impairment in MCs is compensated by the ALP, but compensation of proteasome impairment in podocytes is scant.

## Discussion

Our findings establish the proteasome as a central component of the podocyte and glomerular endothelial cell proteostasis network to ensure glomerular filtration barrier permeability and thus functionality through the orchestration of endocytosis. The (1) conserved abundance of distinct proteolytic 20S proteasome constitutions in podocytes and GEnCs between humans and mice, (2) the development of cell-specific morphologic and functional alterations upon differential 20S proteasome modulation, and (3) the cell-specific interplay with the lysosome system, altogether illustrate the intricacies as well as the relevance of an in-depth dissection of individual cellular proteostasis principles for our basic understanding of organ function and disease pathogenesis.

Proteome perturbations induce cellular stress responses and glomerular cells are differentially affected by proteome disbalances, suggesting the existence of complex diverging proteostasis principles. For example, metabolic stress as in diabetes[12], or lysosomal storage disorders affect the proteome across glomerular cell types and species to varying extents[37]. Our knowledge regarding the extent of these physiologic differences is scant as they can only be unraveled at the cell type level within the glomerular syncytium. This is related to the fact that cells in culture exhibit different proteostasis principles compared to when they are confined within the functional (glomerular) unit. As such, cultured mouse podocytes have an abundant lysosomal system[38], which contrasts the abundant proteasome system in podocytes within the glomerular syncytium identified here. Our findings of the strong dependence of podocytes on c20S function are in conjunction with a recent study, where a podocyte-specific knockout of the 19S regulatory cap protein Rpt3 of the 26S proteasome in mice resulted in a severe and early podocyte injury phenotype[39].

Illustrating the intricacies of this complex protease we find it intriguing that podocytes physiologically rely on the constitutive proteasome whereas GEnCs additionally rely on the immunoproteasome. This finding is conserved between humans and mice and thus relevant for translation. For example, the fact that GEnC health strongly depends on proteasome function yields an explanation as to why multiple myeloma patients treated with the proteasomal inhibitor carfilzomib (irreversible inhibitor of the β5c and β5i proteasomal subunits[40]) in phase II studies report nephrotoxic side effects including thrombotic microangiopathy, which led to the discontinuation of treatment[41].

Generally, it is thought that the immunoproteasome represents a proteasome constitution with enhanced proteolytic capacity and with a different cleavage specificity which is predestined for the generation of peptides for MHC-I presentation[17]. In line, MHC-I membrane abundance is diminished in β5i-deficient GEnCs, as already shown for other cell types[42]. Whether GEnCs require the proteolytic β5i-subunit for specific endothelial-immune regulatory functions is currently under investigation. However, since the β5c-subunit (which is also capable of generating peptides for MHC-I presentation) is drastically upregulated in β5i-deficient GEnCs, we considered distinct immunoproteasome functions within GEnCs beyond antigen presentation. As such, the involvement of the immunoproteasome in the preservation of cellular

proteostasis was initially demonstrated in fibroblasts in the setting of oxidative stress[43]. In line, the de novo expression of the immunoproteasome in the context of podocyte injury[11] is thought to represent a mechanism to preserve proteostasis[37,44]. Hence, the strong immunoproteasome abundance and activity in GEnCs could signify a prominent physiologic need for the degradation of proteasome substrates. However, contrasting the accumulation of proteasome substrates in podocytes, no significant accumulation was appreciated in proteasome-inhibited GEnCs by immunofluorescence or in β5i-deficient GEnCs by immunoblot, which would have been expected. In β5i-deficient GEnCs, the strong compensatory upregulation of the β5c-subunit might preclude proteasome substrate accumulations by shifting their removal by the i20S toward the c20S proteasome.

At this point it is further interesting to note that glomerular endothelial homeostasis was so severely disrupted in β5i-deficiency that podocytes were also affected, even though they reside on the urinary (opposite) side of the GFB. Within the scope of glomerular cell cross-talk, podocytes secrete factors that ensure GEnC health, such as VEGF[45], ephrin B2[46], and angiopoietin-1[47]. It is therefore conceivable that GEnCs secrete factors such as ECM and/or signaling proteins to ensure podocyte health[48] in an immunoproteasome dependent manner, as a role for the immunoproteasome in cytokine secretion has been shown[20]. Further, GEnCs and podocytes secrete ECM proteins for GBM synthesis and maintenance[49]. Hence the observed prominent GBM splitting in endothelial β5i-deficiency as well as in global proteasome inhibition could reflect an altered secretion/removal of ECM material by podocytes and GEnC in a proteasome-dependent manner, resulting in GBM alteration and subsequent podocyte disturbance.

Proteasome substrates prominently accumulated in podocytes of proteasome-inhibited mice as expected[44,50], however, not in MCs and GEnCs showing that podocytes cannot balance their proteome in the setting of proteasome dysfunction. It is increasingly getting acknowledged that the autophagosome-lysosome and proteasome system interact at multiple levels to balance the cellular proteome[12,13]. In MCs, the strong lysosome-dependence renders this cell type unsusceptible to proteasome impairment. Podocytes, on the other hand induce the proteasome system in the initial setting of impaired autophagy[51] or impaired lysosome[37] function, which then successfully balances proteostasis and thus precludes GFB dysfunctions. Why here, in the reverse scenario of initial proteasome impairment the arising proteostasis disturbances in podocytes are not successfully compensated for by the autophagy-lysosome pathway, necessitates further analyses but could in part be related to a suppression of autophagic activity as demonstrated in podocyte-specific Rpt3-deficient mice[39].

Mechanistically, we demonstrate that the (immuno)proteasome orchestrates endocytic turnover and thus clearance of proteins at the GFB. Of note, it is likely that a myriad of other pathways are also affected by proteosome inhibition. Interestingly, podocytes are the most affected cell type, as the injected rabbit IgG and intrinsic mouse IgG both deposit predominantly at the slit diaphragm in proteasome-inhibited mice. This deposition pattern is reminiscent of the IgG deposition described in the early phase of membranous nephropathy[52], an autoimmune glomerular disease in which autoantibodies target podocyte foot process antigens[53]. Podocyte homeostasis is known to strongly depend on an intact endocytic system[54], and the endocytic/transcytotic activity of GEnCs is starting to be understood[25]. Podocytes depend on endocytosis (1) for the turnover/recycling of proteins of the slit diaphragm such as nephrin[27] whose membrane levels (but not total protein abundance) were affected in proteasome-inhibited podocytes in this study; (2) for the transcytosis of molecules such as albumin[10] whose endocytic uptake was reduced in pulse-chase experiments of *Drosophila* nephrocytes in the setting of proteasome inhibition in this study; and (3) for GBM remodeling[55] and maintenance[56], with the GBM showing splitting and electron dense depositions in this study.

Endocytosis is orchestrated by ubiquitin at multiple node points, as essential proteins required for vesicular dynamics and vesicular fusion processes with the plasma membrane are regulated by ubiquitination/deubiquitination[57]. We show that the proteasome influences GEnCs and podocyte endocytosis at multiple node points as (1) clathrin as well as the non-clathrin mediated endocytosis pathways are affected, (2) as blockade of plasma membrane recycling is seen in live-imaging, and (3) as an altered expression and turnover of endocytic receptors at the membrane is present. The impact of the (immuno) proteasome on endocytosis could relate to an altered degradation of endocytosis regulating proteins such as Rab5b[58] or others. Further, proteasomes have recently been described to associate with the cytosolic surface of Golgi membranes and thus be involved in Golgi-homeostasis[59], hence providing a link to secretion-related pathologies. Of note, endocytosis can also be influenced by acid-base conditions i.e., downstream of PI3P generated by phosphatidylinositol 3 kinase catalytic subunit vps34, which plays a major role in podocyte endocytosis[60]. Therefore, the accumulation of ubiquitin and oxidized proteins in proteasome inhibited cells, could render the intracellular environment more acidic, thereby affecting endocytosis. To this end, our total transcript/protein analyses in conjunction with immunolocalization and membrane biotinylation assays could describe altered endocytic receptor localization, trafficking (neonatal Fc-receptor, FcRN), as well as membrane abundance (Mrc2) to underlie the endocytic alteration of podocytes and GEnCs with impaired proteasome functionality.

The FcRN salvages albumin and IgG from the degradative pathway by allowing their transcytosis or recycling. Both, a constitutive as well as a podocyte-specific knockout of FcRN results in glomerular size increase and IgG accumulation[28,33] corroborating our observation in the setting of proteasome impairment. The enhanced FcRN accumulation in a granular/vesicular pattern within GEnCs and podocytes, as well as the perinuclear retention in primary podocytes exposed to epoxomicin shows trafficking alterations "to and from" the plasma membrane to be causative for IgG accumulation in altered proteasome-functionality. Again, cell type specific reaction patterns were present at the filtration barrier. Total FcRN transcript/protein abundance was affected in proteasome-inhibited podocytes but not in β5i-deficient or proteasome-inhibited GEnCs, affirming both cell type and proteasome constitution dependent regulatory effects. Especially β5i-deficient GEnCs exhibited a prominent intracellular mouse IgG accumulation in combination with an intracellular FcRN retention clearly showing a role of the proteasome in membrane protein recycling.

Further, we de novo identified the Mrc2 as an endocytic receptor of podocytes and GEnCs, whose total transcript/protein as well as plasma membrane abundance at the filtration barrier depends on proteasome functionality. Since the Mrc2 is crucially involved in ECM remodeling in other non-glomerular renal cell types[61] through its role in the uptake of collagen, large collagen fragments[34], and thrombospondin 1[35], our findings link the observed ultrastructural GBM alterations in proteasome impairment to the altered membrane turnover of Mrc2.

In summary, our study provides the basis for understanding the intricacies of glomerular cell type related proteostasis and drug-related renal side-effects of proteasome inhibitors in patients and establishes a role for the proteasome in endocytosis regulation and clearance of the glomerular filtration barrier from deposited proteins. Our findings pave the way for future investigations of proteostasis disturbances in the pathogenesis of glomerulonephritis and may lead to new therapeutic principles to target disease-associated glomerular protein accumulations.

## Methods

The presented research complies with all relevant ethical regulations of the "Behörde für Justiz und Verbraucherschutz, Amt für Verbraucherschutz, Lebensmittelsicherheit und Veterinärwesen, Germany", and are conform to the requirements of the German Animal Welfare Act. Anonymous human kidney samples were derived from the healthy pool of tumor nephrectomy samples. Similar samples cannot be accessed by external users. Healthy pig kidneys were derived from the slaughterhouse Itzehoe, Germany.

### Antibodies

Primary antibodies used for the study were: rabbit anti-ubiquitin (immunofluorescence microscopy (IF) 1:300, Novus, #NB300-129); guinea pig anti-nephrin (IF 1:200, Origene, #BP5030); rabbit anti-β5c (IF 1:300, WB 1:5000, laboratory stock X. Wang, University of South Dakota, USA); rabbit anti-α2 (WB 1:1000, Cell Signaling, #2455); rabbit anti-α-actinin-4 (WB 1:1000, Immunoglobe, #0042-05); rabbit anti-p62 (WB 1:2000, Sigma-Aldrich, #P0067), guinea pig anti-p62 (IF 1:500, ProGene, #GP62-c); goat anti-Mrc2 (WB 1:1000, IF 1:200 R&D Systems, #AF4789); AF488 mouse anti H2Kd-(FACS-sort 1:250, BioLegend, Clone #SF1-1.1); BV421 rat anti-CD31 (FACS-sort 1:800, BD Horizon, #562939); AF647 rat anti-CD73 (FACS-sort 1:2000, BioLegend, #127208); AF700 rat anti-CD73 (FACS-sort 1:2000, BioLegend, #127230); PE hamster anti-podoplanin (FACS-sort 1:200, BioLegend, #127408); AF700 rat anti-CD45 (FACS-sort 1:100, BioLegend, #103128); APC-Cy7 rat anti-CD45 (FACS-sort 1:100, BioLegend, #103116); FITC mouse anti-pig CD31 (IF 1:30, Bio-Rad, #MCA1746F); goat anti-biotin (WB 1:1000, Thermo Fisher, #PA1-26792); AF546 rat anti-endomucin (IF 1:50, Santa Cruz, #sc-65495); rabbit anti-K48pUb (IF 1:300, WB 1:1000, Abcam, #ab140601); rabbit anti-LC3B (WB 1:5000, IF 1:50, Cell Signaling, #3868S) rabbit anti-Limp2 (WB 1:500, IF 1:1000, laboratory stock P. Saftig, CAU Kiel, Germany), rabbit anti-Lamp2 (WB 1:1000, IF 1:300, Sigma-Aldrich, #L0668), mouse anti-human LAMP1 (IF 1:200, DSHB, # H4A3); rabbit anti-p57 (IF 1:400, Santa Cruz, #sc-8298); rhodamine-wheat germ agglutinin (IF 1:400, WGA, Vector, #RL-1022); DyLight649-lycopersicon esculentum (tomato) lectin (i.v. injection of 10 ml of a 40 μg/ml solution, Vector, #DL-1178-1); DyLight488-Phalloidin (IF 1:200, Molecular Probes, #21833); AF488 donkey anti-rabbit IgG (IF 1:200, Cy2, WB 1:1000, Jackson ImmunoResearch, #711-545-152); Cy5 donkey anti-mouse IgG (IF 1:200, Cy5, Jackson ImmunoResearch, #715-175-151); guinea pig anti-synaptopodin (IF 1:400, Synaptic Systems, #163004); goat anti-collagen 4 (IF 1:400, Southern Biotechnologies, #1340-01); mouse anti-Rpt5 (WB 1:1000, Enzo, #BML-PW8770); rabbit anti-Lmp7 ((β5i) WB 1:5000, laboratory stock E. Krüger, IF 1:300); mouse anti-β-actin (WB 1:10,000, Sigma-Aldrich, #A5441); goat anti-cathepsin D (WB 1:100, Santa Cruz, #sc-6486); rabbit anti-calreticulin (WB 1:500, Abcam, #ab92516). All secondary antibodies used were either biotinylated, HRP- or fluorescent dye-conjugated affinity purified donkey antibodies (Jackson ImmunoResearch). All antibodies were either validated by us, the companies or controlled for by negative and positive controls.

### Animal experimentation

Mice were housed in a specific pathogen-free animal facility at the University Medical Center Hamburg-Eppendorf. Mice were kept at ambient temperature (20–24°C) and humidity (45–65%), had free access to water and standard animal chow (Altromin 1328P) and were synchronized to a 12 h light/12 h dark cycle.

**Generation of Lmp7^ΔEnC mice.** *Lmp7*^fl/fl mice were generated in C57BL/6J background by genOway (Lyon, France) by Prof. Dr. Frank Heppner, Charité Berlin, Germany. For the generation of an inducible endothelial cell-specific β5i-deficiency, *Lmp7*^fl/fl mice were crossed to the Cdh5-Cre-ERT2 mouse line[62] and fully back crossed to the C57BL/6 background. In C57BL/6 *Lmp7*^fl/fl cre+ mice, tamoxifen application (tamoxifen, Sigma, #T5648-1G, 75 mg/kg b.w. diluted in sunflower seed oil (Sigma, #S5007-250ml) i.p. on 5 consecutive days) induces Cre-mediated excision of *Lmp7* exon 3 in endothelial cells (*Lmp7*^ΔEnC). C57BL/6 *Lmp7*^fl/fl cre- mice

exposed to tamoxifen were used as control littermates (Ctrl). The naïve phenotype of mice was analyzed 5–50 weeks after the last tamoxifen injection. Both male and female mice were included within the experiments. Genotyping was performed as follows: the different genotypes were verified by PCR. Briefly, for PCR reaction, tail biopsies were lysed in DirectPCR (Tail) (Viagen, #102-T) with proteinase K (20 mg/ml, Sigma, #P6556) overnight at 55°C. Primers for the *Cdh5 cre*-locus were: *Cdh5*-transgene-fw 5′-GCG GTC TGG CAG TAA AAA CTA TC-3′, *Cdh5*-transgene rev: 5′-GTG AAA CAG CAT TGC TGT CAC TT-3′; *Cdh5*-control-fw: 5′-CTA GGC CAC AGA ATT GAA AGA TCT-3′, *Cdh5*-control-rev: 5′-GTA GGT GGA AAT TCT AGC ATC ATC C-3′; for the flox-locus of *Lmp7* were *Lmp7*-fw: 5′-GCT ATA ATG CCA GCT CTG TCT GAA CTT CG-3′, *Lmp7*-rev: 5′-TGC CTC TTG CAT CTC TTA GCC CAC C-3′.

**Inhibitor treatment.** Male BALB/c mice were purchased from Charles River. Mice were analyzed at 16–20 weeks of age. For inhibitor studies mice received the proteasomal inhibitor epoxomicin (Enzo #BML-PI127, 0.5 μg/g bodyweight) or the lysosomal inhibitor leupeptin A (Sigma #L5793, 40 μg/g bodyweight) by intra-peritoneal injection on 4 consecutive days, vehicle control mice received equal amounts (25%) DMSO in PBS. For quantification of glomerular immunoglobulin deposition, mice received 160 μl rbIgG (Bio & Sell, #RAB.SE.0100) by intra-venous injection, 7 days post rbIgG injection mice received the aforementioned inhibitors on 4 consecutive days. Animal euthanasia was performed following subcutaneous buprenorphine (0.1 mg/kg KG, Indivior, #IND00979) administration for analgesia 30 min prior to cervical neck dislocation under 3.5% isoflurane inhalation narcosis. All experimental procedures were performed according to the institutional guidelines.

**Sample collection, serum, and urine analysis.** For sample collection, mouse urine was collected in metabolic cages. Blood was drawn from the aorta at the time of sacrifice. Mice or isolated kidney packages were thoroughly perfused either through the aorta or through the heart with 20 ml PBS to remove blood components such as immunoglobulins from the renal circulation. Successful perfusion was controlled by the color of the kidney. Samples were stored at −80°C prior to further analysis. Mouse blood was centrifuged for 10 min at $1500 \times g$ to separate blood cells from serum. Mouse serum was analyzed for the determination of blood urea nitrogen (BUN) by standard methods using an autoanalyzer (Hitachi 717, Roche) in the Department of Clinical Chemistry at the University Hospital Hamburg. Urine albumin content was quantified using a commercially available ELISA system (Bethyl, #E99-134) according to the manufacturer's instructions, using an ELISA plate reader (BioTek), as described[63]. Urinary albumin values were standardized against urine creatinine values of the same individuals determined by Jaffé (Hengler analytic, #114444) and plotted.

## Glomeruli isolation
**Mouse glomeruli.** Glomeruli were isolated using Dynabead perfusion as described[64]. In brief, kidneys were perfused with 5 ml magnetic bead solution per kidney (magnetic bead solution: 50 μl Dynabeads™ M-450 Tosylactivated Invitrogen Thermo Fisher #14013 with 200 μl SPHERO Polystyrene Magnetic Particles, Spherotech, #PM-40-10 diluted in 50 ml HBSS, Gibco, #14170-138) and digested with collagenase (HBSS, 1.2 mg/ml collagenase 1A (Sigma, #C9891), 100 U/ml DNase I, (Roche, #10104159001)) for 15 min, 37°C, 1300 rpm. The solution was strained through a 100 μm filter, which was rinsed with HBSS, this was done 3 times. The solution was centrifuged for 5 min, $600 \times g$, 4°C. The pellet was resuspended in HBSS, and the glomeruli were separated from tubuli with a magnetic particle concentrator. The number of isolated glomeruli and contaminating tubuli was quantified under the phase contrast microscope prior to storage of glomeruli at −80°C.

**Pig glomeruli.** Prior to the isolation of glomeruli, pig kidneys were perfused with HBSS until they were light brown in color. After separation of the cortex from the medulla, the cortical tissue was weighed and added to PBS (Gibco, #14190-169 containing 0.05% BSA (Sigma, #A7906). The tissue was then homogenized with a blender. The homogenate was sieved through a 300 μm sieve placed on a beaker and pushed through with a small Erlenmeyer flask using low pressure. To finally separate glomeruli from tubular remnants, 90 μm sieves were used whereby intact glomeruli remained on the sieve. By using a syringe filled with PBS containing 0.05% BSA, a small cannula and high pressure the glomeruli were flushed into a clean beaker. To increase the purity grade of the glomerular isolate, the last step was performed four times, using clean 90 μm sieves each time. The collected suspension was then sieved through another 212 μm sieve to eliminate remaining tissue parts. The glomeruli were centrifuged for 10 min at $1000 \times g$, 4°C. The number and purity of isolated glomeruli was determined under a phase contrast microscope.

## Bulk-isolation of glomerular cell types
The isolation of podocytes, mesangial and endothelial cells was performed using the timMEP procedure as published[19]. Initially glomeruli were isolated as described above. The number of isolated glomeruli and contaminating tubuli was quantified under the phase contrast microscope. A fraction of the glomeruli was set aside as a positive control. The remaining glomeruli were processed for glomerular cell type isolation. Briefly, glomeruli were centrifuged for 5 min, $1000 \times g$, 4°C, the supernatant was discarded. Glomeruli were resuspended in 1 ml Liberase digest solution [(1000 μg/ml Liberase TL (Roche, #5401020001), 100 U/ml DNase I (Roche) in cell culture media (10% FCS (Gibco, #10500-064), 1% ITS (Pan-Biotech, #P07-03200), 1% Pen/Strep (Thermo, #15070063), 25 mM HEPES (Gibco, #15630056), RPMI Media 1640, (Gibco, #61870044))]. The glomeruli solution was received in a 2 ml reaction tube and incubated for 2 h, 37°C, 1400 rpm in a thermomix, while being disrupted mechanically by repeated cycles of vortexing and shearing. Following mechanical disruption, glomerular remnants were removed at the magnetic particle concentrator for 5 min. The supernatant containing the glomerular cells was collected in a 1.5 ml reaction tube. The reaction tube was topped off with MACS buffer (PBS, 0.5% BSA, 2 mM EDTA (Sigma, #E4884) and the cells were pelleted for 10 min, $1000 \times g$, 4°C, the supernatant was discarded. The cells were resuspended in 100 μl MACS buffer and the FACS antibodies were added. Depending on the mouse strain used, following FACS panels were applied for separation of podocytes, mesangial cells, and glomerular endothelial cells: BALB/c: CD31, CD73, podoplanin, live/dead stain (1:1000, Invitrogen, #L10119), and CD45; $Lmp7^{\Delta EnC}$: CD31, CD73, podoplanin, live/dead stain, MHC-I, and CD45. FACS antibodies were incubated at 4°C for 30 min in the dark. The cells were subsequently washed with 1 ml MACS buffer and pelleted for 10 min, $1000 \times g$, 4°C, the supernatant was discarded, and cells were resuspended in 300 μl PBS and filtered through a 40 μm filter into FACS tubes. The collection tubes contained 100 μl PBS. Glomerular cell types were sorted at the Aria IIIu or the BD FACS Fusion sorter (Becton Dickinson). APC-Cy7 was used as dump channel to remove CD45-positive immune cells as well as dead cells from the preparations. The cells were pelleted for 10 min, $1500 \times g$, 4°C, the supernatant was discarded. Pellets were stored at −80°C till further use. GEnC cell size (mean FSC-A) as well as MHC-I surface expression (MFI of the respective channel) was analyzed from the data collected in the course of FACS-sorting of the glomerular cell types using FlowJo.

## Cell cycle analysis via flow cytometry
Single cell suspensions were generated as described above. Cells were stained extracellularly with CD31 and live/dead stain as described above. Thereafter, cells were fixed and permeabilized with 70% EtOH

overnight. Cells were washed twice with MACS buffer and subsequently stained with propidium iodide (PI, Molecular Probes,) staining solution (20 μg/ml PI, 200 μg/ml RNase A in PBS, Molecular Probes, #F10797) for 30 min at RT. Cells were diluted in PBS and analyzed at a FACS Symphony A3 (BD) flow cytometer. The PI signal was detected in the PE channel. Cell cycle analysis was performed using FlowJo 10.9.0 using "Cell Cycle" function and a pragmatic Watson model.

## Cell culture
**Human immortalized podocytes.** Human immortalized podocytes (a kind gift of Moin Saleem, University of Bristol, Bristol, Great Britain) were cultured under permissive conditions [32°C, 5% $CO_2$, RPMI 1640 supplemented with 10% fetal calf serum, 1X insulin, transferrin selenium (ITS, PAN Biotech), 1 μg/ml Puromycin (Sigma, #P7255)] in uncoated, vented tissue culture flasks (Sarstedt) as described[65]. For differentiation, podocytes were cultured for 10 days under non-permissive conditions [38°C, 5% $CO_2$, RPMI 1640 supplemented with 10% FCS, 1X ITS] for live-cell imaging as detailed below. Cell density was kept below 80–90% to allow process development. DNA was extracted from cell pellets and controlled for the absence of mycoplasma infections at a 2–3 monthly basis according to the manufacturer's instructions (Minerva Biolabs), cellular passages below 45 were used for experiments.

**Primary mouse podocytes.** Decapsulated glomeruli were plated on 6-well plates (Sarstedt) or onto Corning BioCoat Collagen I 35 mm TC-treated culture dishes (Corning, #354456) in RPMI 1640 supplemented with 10% FCS, 15 mmol/l HEPES, 1 mmol/l sodium pyruvate (Gibco, #11360-039), 100 U/ml penicillin, 100 mg/ml streptomycin at 38°C, 5% $CO_2$. Four to 6 days after initial glomerular isolation, primary culture podocytes were used for the endocytosis assay detailed below.

## In vivo biotinylation and streptavidin pulldown
To quantify the in vivo abundance of nephrin and Mrc2 at the plasma membrane of glomerular cells (nota bene of podocytes, as both proteins are only expressed by podocytes within the glomerulus), in vivo biotinylation assays were performed as published[66,67]. In brief, kidneys were first perfused through the renal arteries with 10 ml PBSCM (PBS containing 1 mM $MgCl_2$ (Th. Geyer, #11222620) and 0.1 mM $CaCl_2$ (Sigma, #C-3881)) per kidney, then with 5 ml PBSCM supplemented with 0.5 mg/ml EZ-Link Sulfo-NHS-LC-Biotin (Thermo Scientific, #21335) per kidney, followed by perfusion with 5 ml PBSCM + 100 mM glycine (Roth, #3908.2) per kidney as a quencher. Following in vivo biotinylation, kidneys were perfused with 5 ml Dynabead solution per kidney through the renal artery for standard glomerular isolation as described above.

Biotinylated proteins were isolated via a streptavidin pulldown. Briefly, glomerular lysates were transferred to an appropriate amount of Streptavidin Sepharose High Performance matrix (Cytiva, #17511301) and incubated over night at 4°C. The samples were centrifuged for 30 s, 300 × g, 4°C, the supernatant was collected as the flowthrough fraction. The resin was washed with wash buffer I (0.5% sodium deoxycholate (Sigma, #30970), 1% Triton X-100 (Sigma, #T8787) in PBS, pH 7.4) one time, with Neufeld buffer (0.1% SDS (Roth, #2326.2), 0.05% NP-40 (Sigma, #N3516), 0.6 M NaCl (chemSolute, #6307), 10 mM Tris/HCl (Sigma, #T1503), pH 8.5) once, with wash buffer II (0.5% sodium deoxycholate, 1%Triton X-100 in PBS, 2 M KCl (Merck, #1049361000), pH 7.4) once, and lastly with 1% PBS, two times. After the last wash the supernatant was collected as wash fraction. The biotin was released from the matrix by denaturation with an equal amount of 2.5 x laemmli buffer. Samples were loaded on a SDS-PAGE under denaturing conditions for immunoblotting to Mrc2, nephrin, biotin and β-actin.

## Glomerular rabbit IgG pulldown
To quantify deposited rabbit IgG in glomeruli by immunoblot analysis, glomeruli-number-adapted lysates were transferred to an appropriate

amount of protein A/G resin (static binding capacity of 20 mg IgG per ml, Thermo Fisher, #53135) and incubated over night at 4°C. The samples were centrifuged for 10 min, 14,000 × g, 4°C, the supernatant was discarded. Rabbit IgG was released from the protein G resin by denaturation with an equal amount of 2 x laemmli buffer. Samples were loaded on a SDS-PAGE under denaturing conditions for immunoblotting to rabbit IgG. As the experiments depict the amount of rabbit IgG that was pulled down from the same number of glomeruli, an immunoblot loading control is technically not providable.

## Quantitative PCR
Total messenger RNA was extracted from isolated glomeruli and respective FACS-sorted podocytes, mesangial cells, and endothelial cells from the corresponding mouse with the commercially available NucleoSpin RNA, Plus XS kit for RNA purification (Macherey-Nagel, #740990.250). RNA was solved in purified $H_2O$. The whole amount of extracted RNA was reverse transcribed with random hexamer primer (Invitrogen, #SO142) and RevertAid reverse transcriptase (Thermo Scientific, #EP0442). The reverse transcription took place in a thermocycler (Biometra) at 25°C for 10 min, 42°C for 1 h, 70°C for 10 min, 4°C forever. mRNA expression was quantified with the QuantStudio 3 (Applied Biosystems) qPCR cycler using SYBR green. Exon spanning primer pairs to murine cDNA sequence were used (Supplementary Table 1). Relative gene expression was calculated using the ΔΔCT method.

## Immunoblot analysis
All immunoblots were performed with isolated glomeruli, bulk-isolated podocytes, mesangial or endothelial cells, or primary culture podocytes. To achieve equal loading for comparative investigations, following lysis protocols were applied: (1) 5000 glomeruli were lysed in 150 μl lysis buffer, 10 μl were loaded per well. (2) Glomerular cell types were loaded in a number-adapted manner according to the individual FACS-sort cell counts. In general, for glomerular cell types that were bulk-isolated from naïve BALB/c mice, 50,000 podocytes, 50,000 mesangial cells, and 50,000 GEnCs were lysed in 10 μl lysis buffer, 10 μl were loaded per well. As all cell types differ in their expression of housekeepers a normalization to housekeeper is not possible. (3) From naïve $Lmp7^{\Delta EnC}$ mice 50,000 GEnCs and 50,000 non-endothelial kidney cells were lysed in 10 μl lysis buffer, 10 μl were loaded per well.

For standard immunoblotting, samples were lysed in T-Per (Thermo Fisher Scientific, #78510) containing 1 mM sodium fluoride (ThGeyer, #2618.1), 1 mM sodium vanadate (Sigma, #450243-10G), 1 mM calyculin A (ThGeyer, #AB348875), Protease Inhibitor Sigma Fast, EDTA free (Sigma, #S8830) and denatured in SDS solubilization buffer. Samples were separated on a 4–12% MiniProtean TGX gel (BioRad) in a Tris-glycine migration buffer (0.25 M Tris, 1.92 M glycine, 1% SDS, pH 8.3). Protein transfer was performed in transfer buffer (0.192 M glycine, 25 mM Tris, 20% EtOH in $H_2O$) in a TransBlot Turbo System (BioRad). After the transfer, all proteins were visualized by ponceau staining. PVDF membranes (Millipore) were blocked (5% nonfat milk) prior to incubation with primary antibodies diluted in Superblock reagent (Thermo Fisher Scientific) or non-fat milk. Binding was detected by incubation with HRP-coupled secondary antibodies (1:10,000, 5% nonfat milk). Protein expression was visualized with ECL SuperSignal (Thermo Fisher Scientific, #34578) according to the manufacturer's instructions on an Amersham ImageQuant 600 or 800 (GE Healthcare, Cytiva). Ponceau, β-actin, α2, or biotin stainings of the same membrane are shown, fine dashed black lines indicate, where bands were not adjacent to another on the membrane. Quantification of immunoblots was performed using FIJI version 2.0. Proteins of interest were normalized to home keepers of the same membrane, depending on the system investigated, either β-actin, ponceau, α2, or biotin were used as home keepers for normalization. GEnCs of

*Lmp7*[ΔEnC] versus control littermates were not normalized to β-actin due to β-actin regulation in β5i-deficiency (Supplementary Fig. 13C, right graph). Hence, care was taken to load equal cell numbers between genotypes. Uncropped images of all gels and blots shown in the Figures are provided in the source data file.

## Morphological analysis

For conventional light, confocal, and high-resolution confocal microscopy kidney cortices from experimental mice were processed as follows: (1) For standard paraffin sections, kidneys were fixed in 4% PFA (Sigma, #441244) over night at 4 °C, embedded in paraffin, and cut on a rotation microtome into 1.5–3 μm thin sections and deparaffinated for staining with antigen retrieval; (2) for antigen-preservation, kidneys were fixed with zinc fixative (for 1 liter: 0.5 g calcium acetate (Sigma, #402850), 5 g zinc acetate (Sigma, #Z0625), 5 g zinc chloride (Sigma, #793523), with 0.1 M Tris all from Sigma pH 7.4) at 4 °C over night, embedded in paraffin, and cut on a rotation microtome into 3 μm thick sections, and deparaffinized for staining without antigen retrieval; or (3) for cryo-sections kidneys were exposed to 30% sucrose 4 °C for 6–12 h followed by freezing to −80 °C in OCT cryo-protection medium, and cut on a cryostat into 5–8 μm thick sections, and post-fixed with either 4% PFA 8 min at RT, with 100% acetone (precooled to −20 °C) for 5 min, or with 100% MeOH (precooled to −20 °C) for 5 min.

For light microscopic evaluation, 1.5 μm thick paraffin sections were stained with periodic-acid-Schiff reagent (Sigma, #1.08033.0500) according to the manufacturer's instructions. For immunofluorescent stainings, paraffin sections were deparaffinized and antigen retrieval was performed by steam cooker boiling for 40 min at 98 °C in pH6.1 buffer (DAKO, #S2369) or pH9 buffer (DAKO, #S2367) or by protease XXIV (Sigma, #8038, 5 μg/ml) digestion 15 min at 37 °C. Unspecific binding was blocked in 5% horse serum (Vector, VEC-S-2000) and 0.05% TritonX-100 for 30 min. Primary antibody incubations (in blocking buffer, o/n, 4 °C) were followed by incubation with biotinylated or AF488-, Cy3-, or Cy5-coupled secondary antibodies (1:200, 30 min) together with Hoechst (1:1000, Invitrogen, #H3570). Stainings were evaluated with an LSM800 with airyscan 1 microscope (ZEISS) for conventional and high-resolution confocal microscopy and a LSM980 with airyscan 2 microscope (ZEISS) using ZEN 3.0 software (ZEISS).

## Podocyte exact morphology measurement procedure (PEMP)

Quantification of filtration slit density (FSD) was performed as published[67,68]. In brief, 3 μm paraffin sections mounted on high-precision coverslips coated with Poly-L-lysine (Sigma, #P2636) were stained for nephrin. Stained sections were mounted with ProLong Gold (Thermo Fisher, #P10144) for imaging. 3D z-stacks from nephrin-stained kidney sections were acquired with sub diffraction limit resolution using a Visitron-SD-TIRF (with SoRa unit form Yokogawa) microscope. PEMP analysis was performed using the PEMP macro for FIJI[68]. For this, the capillary area was encircled, and the slit diaphragm length was determined. FSD values were calculated from the ratio of slit diaphragm length and capillary area. Per animal, 5 glomeruli with 9–22 regions of interest (ROIs) were analyzed.

## Electron microscopy

For electron-microscopical analyses, small cortical samples were fixed in 4% buffered paraformaldehyde with 1% glutaraldehyde (Roth, #4157.2) at 4 °C for at least 24 h. Tissue was post-fixed with 1% osmium in 0.1 M phosphate buffer (1 h at RT), stained with 1% uranylacetate (1 h at RT in 70% ethanol), dehydrated and embedded in epoxy-resin (Durcupan, Sigma-Aldrich). Ultrathin sections were cut (Ultra-microtome UC6, Leica) and contrasted with lead citrate. Micrographs were generated with a transmission-electron microscope (TEM 910, Zeiss, Oberkochen, Germany). Quantification of glomerular endothelial fenestrations and foot process effacement in *Lmp7*[ΔEnC] mice was performed using ImageJ, version 1.53t.

## Proteasomal activity assays

**Lysate chymotrypsin-like activity assay.** The main 20S core particle proteolytic activity (β5c and β5i, chymotrypsin-like activity) was measured in glomeruli as follows. Isolated glomeruli were lysed in T-Per (Thermo Fisher Scientific) supplemented with Protease Inhibitor Sigma Fast EDTA free. Protein concentration was determined by spectrophotometry (DeNovix DS-11). Ten μg total protein were diluted in incubation buffer (20 mM HEPES, 0.5 mM EDTA, 5 mM DTT (Sigma, #9779), 0.1 mg/ml ovalbumin (Sigma, #A5503) in $H_2O$, pH 7.8) to a final volume of 50 μl. Samples were pre-incubated in incubation buffer for 2 h at 4 °C. Following pre-incubation, the substrate Suc-LLVY-AMC (Bachem, #4011369) was added to the samples at a final concentration of 60 μM and to an end volume of 100 μl. Proteasomal activity was measured in triplicate at 355 and 460 nm in a fluorescent spectrophotometer (Mithras LB 940) after incubation for 2 h at 37 °C in the dark.

**β-subunit-specific proteasomal activity assay.** To measure the β-subunit-specific proteasomal activities of the 20S core particle in isolated glomerular cell types, cell types were lysed cell number adapted with TSDG buffer (10 mM Tris pH 7.5, 10 mM NaCl, 25 mM KCl, 1 mM $MgCl_2$, 0.1 mM EDTA, 1 mM DTT, 2 mM ATP (Sigma, #A3377) via eight freeze-thaw cycles. Samples were incubated for 1 h, 37 °C with 0.5 μM MVB003 (pan-proteasomal activity-based probe, assesses activity of all proteolytic β-subunits and does not differentiate between β5c and β5i activity due to very similar molecular weight of both β-subunits) or 1 μM GB514 (β5i-specific activity-based probe). As an assay control rat hybridoma cells (which are abundant in c20S as well as i20S proteolytic β-subunits) were incubated with or without 5 μM epoxomicin for 1 h, 37 °C prior to the incubation with activity-based probes. Samples were denatured in SDS solubilization buffer for 10 min at 70 °C, samples were separated on a 12.5% Tris-Glycine gel in a Tris-glycine migration buffer. The activity was visualized on a Fusion FX7EDGE V0.7 Imager (Vilbert Lourmat) using the appropriate filters. Following fluorescent visualization, activity gels were blotted onto PVDF membranes and further processed for conventional immunoblotting (as described above) to proteasomal subunits and loading controls. Uncropped images of all gels and blots shown in the Figures are provided in the source data file.

**Ex vivo pig glomerular proteasomal activity assay.** To measure the proteasomal activity in cultured pig glomeruli, glomeruli were incubated in medium (Willams Rodium, Glutamax, 2,7 g/dl glucose, Gibco, #32551020) containing 0.5 μM MVB003 for 1 h at 37 °C and 300 rpm. The immunoproteasome specific probe GB514 could not be used, as it is barely cell permeable. As an assay control, glomeruli were incubated with 2 μM epoxomicin for 1 h at 37 °C and 300 rpm before adding 0.5 μM MVB003 for another hour. Incubation with an equivalent amount of DMSO served as a negative control. To stop the reaction, glomeruli were washed with PBS + 50 mM EDTA. A fraction of glomeruli was fixed with 2% PFA and stained with the cell type specific markers CD31 for GEnCs and nephrin for podocytes, washed with 3 × 5 min with PBS, applied to super-frost slides, and mounted with Fluoromount mounting medium (Southern Biotech, #0100-01) for confocal analysis. Fluorescent signal intensity demonstrates over all β-subunit activity and does not differentiate which specific β-subunit is active. A second fraction of the glomeruli was used to further control for the assay, glomerular lysates were generated by 7 consecutive cycles of freeze-thawing in TSDG buffer. Samples were denatured in SDS solubilization buffer for 10 min at 70 °C, samples were separated on a 12.5% Tris-Glycine gel in a Tris-glycine migration buffer. The activity was visualized on a Fusion FX7EDGE V0.7 Imager (Vilbert Lourmat) using the appropriate filters. Following fluorescent visualization, activity gels were blotted onto PVDF membranes and further processed for conventional immunoblotting (as described above) to

proteasomal subunits and loading controls. Uncropped images of all gels and blots shown in the Figures are provided in the source data file.

### Lysosomal lysate activity assay

For photometric measurement of β-hexosaminidase activity in glomeruli, samples were lysed glomeruli number-adapted (T-Per containing protease inhibitor cocktail). Samples were incubated with substrate buffer (10 mM 4-nitrophenyl N-acetyl-β-D-glucosaminide (Sigma, #N9376), 0.1 M Na-citrate, 0.2% BSA, pH 4.6) for 1 h at 37 °C. The reaction was stopped with stop solution (0.4 M glycine, pH 10.4). Activity was measured using a microplate spectrophotometer (BioTek, EL 808) at 405 nm.

### Endocytosis assays

**Endocytosis assay in primary culture podocytes.** Isolated decapsulated glomeruli from naïve male BALB/c mice were seeded onto 6-well plates for WB analysis or onto Corning BioCoat Collagen I 35 mm TC-treated culture dishes (Corning, #354456) (1000 glomeruli per well) containing 2 ml cell culture medium (RPMI 1640 containing 10% FCS, 1% Pen/Strep, 1 mM Sodium pyruvate, 15 mM HEPES). Primary podocytes were grown out of glomeruli over 5–6 days at 37 °C, 5% CO₂. Podocytes were treated with either 10 nM epoxomicin (Enzo) or vehicle (equal concentration of DMSO) for 24 h. After the 24 h of inhibitor treatment, cells were synchronized on ice for 30 min during which endocytic substrates were applied, namely transferrin-biotin (5 μg/ml, Jackson ImmunoResearch, #2337204), transferrin-FITC (5 μg/ml, Jackson ImmunoResearch, #2337209), Cy3-rbIgG (1:200) or rabbit IgG (1:200, Bio & Sell, #Rab.SE.0100). Primary podocytes were then transferred to 37 °C, 5% CO₂. At the indicated time points (for 0, 30 and 60 min after transfer to 37 °C) primary podocytes were washed 3 times with PBS and harvested by applying 0.05% Trypsin-EDTA (Gibco, #25300054). For histology the cells were fixed in 4% PFA for 8 min. Primary podocytes were pelleted for 10 min, $1000 \times g$, 4 °C, the pellets were stored at −80 °C prior to processing for conventional immunoblotting.

**Live-cell imaging of human podocyte endocytic activity.** Immortalized human podocytes were seeded on collagen-IV coated glass-bottom well chamber slides (Ibidi μ-Slide 8-well, #80827) in a density of 5000 cells per well. After 10 days of differentiation, cells were treated with 1 μM epoxomicin (Enzo) for 6 h or with the corresponding amount of DMSO as control. Live-cell experiments were performed at 37 °C in a humidified atmosphere with controlled CO₂ level using a Nikon Ti2 based Spinning Disc microscope equipped with a Yokogawa CSU-W unit. Time-lapse images were acquired using a CFI Plan Apochromat λD 100x NA 1.45 oil immersion objective and an Andor iXON888 EMCCD camera (Oxford Instruments). Experiments were carried out in 4 independent imaging sessions while 8–12 individual cells were imaged per experiment and condition. Individual cells were chosen according to their morphology using brightfield imaging. Prior to the addition of rhodamine-WGA (10 μg/ml, Vector, #RL-1022) to visualize the plasma membrane as well as the internalized vesicles originating through endocytosis, a z-stack of 30 μm with a step size of 1 μm was acquired ensuring acquisition of the whole cells. Images were taken using the 561 nm excitation laser and a corresponding 609/54 bandpass emission filter. After the addition of rhodamine-WGA, automatic time-lapse and z-stack recordings were performed at 2.5-min intervals for 50 min.

After live-cell imaging (60 min after application of WGA), cells were immediately fixed with 4% PFA at RT for 20 min and washed with PBS for confocal microscopy to visualize the in culture applied rhodamine-WGA, proteasome substrates, and f-actin. For this, unspecific binding was blocked (5% horse serum (Vector Laboratories, #VEC-S-2000), 0.05% TX100, 30 min at RT) and primary antibody incubation

was performed in blocking buffer overnight, 4 °C. Following washes with PBS, an incubation of secondary antibodies (1:200 in blocking buffer, 1 h at RT) followed, before human podocytes were mounted in Fluoromount (Southern Biotechnology) for high-resolution confocal microscopy.

**FITC-albumin uptake assay in Drosophila nephrocytes.** FITC-albumin uptake assays have been performed as previously published[68,69]. Garland nephrocytes from 3rd instar larvae from the wildtype strain OregonR were isolated. Isolated nephrocytes were incubated with hemolymph-like buffer (HL3.1; 70 mM NaCl, 5 mM KCl, 1.5 mM CaCl₂, 4 mM MgCl₂, 10 mM NaHCO₃, 115 mM Sucrose and 5 mM HEPES), DMSO (vehicle control) or 2 nM epoxomicin for 15 min at RT on a shaker (300 rpm). Inhibitor and vehicle were diluted in HL3.1 buffer to achieve final concentrations. After incubation, nephrocytes were washed once for 1 min with HL3.1 buffer, followed by incubation with FITC-albumin (0.2 mg/ml; Sigma-Aldrich) for 1 min. After another washing step with HL3.1 for 1 min, nephrocytes were fixed with 4% formaldehyde for 20 min. The tissue was mounted using Prolong Gold mounting medium (Invitrogen). Confocal imaging was performed and the mean intensity/nephrocyte area was quantified using FIJI.

### Transcriptomic bioinformatics

Mouse glomerular single cell transcriptomics data published by He et al.[18] and human kidney single cell RNAseq data accessible within the Kidney Precision Medicine Project (KPMP, accessed 12/04/2023; https://www.kpmp.org) were analyzed for the differential expression of transcripts between podocytes, mesangial cells and glomerular endothelial cells. Data were analyzed using R 4.3.2 (R Core Team (2023). _R: A Language and Environment for Statistical Computing_. R Foundation for Statistical Computing, Vienna, Austria, <https://www.R-project.org/>) and Seurat V5[70]. Glomerular cell types in the raw datasets were identified via known marker genes (*Cdh5*, *Ehd3*, and *Pecam1* for GEnCs, *Thsd7a* and *Nphs1* for PCs, *Nt5e* and *Pdgfrb* for MCs)[70]. Pseudobulk analysis was performed using the AggregateExpression function and heatmaps were generated using the DoHeatmap function. Heatmaps depict the relative transcript levels of podocytes, glomerular endothelial and mesangial cells from the respective dataset.

### Proteomic bioinformatics

Proteomic raw data from Hatje et al.[19] were reanalyzed. Protein raw files were searched using MaxQuant and the LFQ algorithm[71,72] with search against a uniport mouse proteome reference database released in Jan 2018. Default criterions were used, meaning that PSM, peptide and protein FDRs were set at 0.01. LFQ algorithm was enabled, match between run was enabled. The data were analyzed using Perseus v 1.5.5.3 using filtering for the embedded annotations as contaminant, reverse or proteins identified by site only. Only proteins in at least 60% of samples were kept and missing values were imputed using default imputation parameters (downshift SD = 2, width 0.3). GO-term and uniprot keyword annotation and enrichment was performed using the embedded terms[73]. Radarplots were generated using ggradar package (Rstudio) using default settings. Differentially expressed proteins were defined by ANOVA with FDR-corrected *p* value of less than 0.01 to adjust for multiple testing.

### Statistical analysis

Results were expressed as mean ± SEM, and significance was set at *$p < 0.05$. The means were compared using the one-way ANOVA with corresponding post hoc test for multiple comparisons, specified in the figure legends using Prism 8.2.1 for Mac OS X, GraphPad Software, San Diego, California USA, www.graphpad.com. Replicates used were biological replicates, which were measured using different samples

derived from distinct mice. All animals were littermates and were blindly assigned to the experimental groups.

## Reporting summary

Further information on research design is available in the Nature Portfolio Reporting Summary linked to this article.

## Data availability

Source data are provided with this paper.

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

## Acknowledgements

This work was funded by the Deutsche Forschungsgemeinschaft (ME2108/10-1 to C.M.S.; SFB1192: project B3 to C.M.S.; KR1984/4-1 to O.K.). U.W. was supported by the graduate school iPRIME. We are grateful to the staff of the FACS Sorting Core Unit and the UKE imaging facility (UMIF) under the DFG Research Infrastructure Portal: RI_00489 for excellent technical assistance. The LSM980 Airyscan 2 was funded by the DFG (INST 152/952-1 FUGG to C.M.S.). KPMP is funded by the National Institute of Diabetes and Digestive and Kidney Diseases (Grant numbers: U01DK133081, U01DK133091, U01DK133092, U01DK133093, U01DK133095, U01DK133097, U01DK114866, U01DK114908, U01DK133090, U01DK133113, U01DK133766, U01DK133768, U01DK114907, U01DK114920, U01DK114923, U01DK114933, U24DK114886, UH3DK114926, UH3DK114861, UH3DK114915, UH3DK114937). We acknowledge financial support from the Open Access Publication Fund of UKE—Universitätsklinikum Hamburg-Eppendorf—and DFG—German Research Foundation.

## Author contributions

C.M.S. conceived the project, designed the experiments, performed stainings and confocal microscopy, analyzed data, supervised the project, and wrote the paper. W.S. did most of the experimental work in vivo and in vitro, work with primary podocytes, analyzed data and wrote the paper. L.B. experimented and analyzed and compiled the data of the Lmp7ΔEnC mice, performed the single cell RNAseq reanalyzes. D.L., C.C., and R.T. established, performed, and analyzed the live-imaging experiments of the human podocytes. S.K. performed endocytosis experiments in Drosophila, O.K. and D.L. performed the EM analyses. L.S. established and performed the ex vivo pig glomerular activity measurement, F.H. (UKE) and U.W. established bulk-isolated glomerular cell type samples and performed qPCR analyses. M.S., S.Z., and J.B. provided excellent technical assistance. B.I.F. provided activity-based probes and provided technical support. F.H. (Charité) generated the Lmp7fl/fl mice and E.K. provided technical support. M.M.R. performed sub analyses of the timMEP proteomic dataset[19].

## Funding

## Competing interests

The authors declare no competing interests.
