## [Peer Review File · Nature Communications]

The Proteasome Modulates Endocytosis Specifically in Glomerular Cells to Promote Kidney FiltrationREVIEWER COMMENTS

Reviewer #1 (Remarks to the Author):

In the presented manuscript, Sach et al., explore the role of proteasomes in maintaining homeostasis of the glomerular filtration barrier and prohibiting protein deposits from accumulating in the glomerulus. Importantly, the authors analyzed the three resident glomerular cell types in parallel to dissect the involvement of different degradation systems in their function. Indeed, understanding the role of both proteasome and lysosomal proteolysis across different cell types within a tissue remains an understudied area of research and the authors provide new insight into a specialized role for these systems in cell type-specific and proteasome-specific manner. Further, the authors identified a role for the immunoproteasome subunit Imp7 and the constitutive proteasome in diseases associated with glomerular protein accumulation.

The authors found a striking and novel effect of Imp7 knockout on the GBM, showing how immunoproteasome subunit knockout can affect the GBM, and increase sedimentation of immunoglobulins. The authors also focus on the aggregation of antibodies in the kidneys, a highly relevant clinical question in the context of poorly understood and highly morbid membranous and glomerular nephropathy. They show that endosomal uptake is disrupted by proteasome inhibition and genetic manipulation of the immunoproteasome. The authors show that treatment with proteasome inhibitors causes a decrease in the kidneys' ability to internalize proteins via a reduction in endocytic receptor abundance. However, mechanistic aspects of how the immunoproteasome exerts its effect on the kidney (as suggested in the discussion) were not described. Yet, this work offers an important insight into proteostasis control mechanisms at the kidney that would be of broad interest to the scientific community. Several of their key points need to be further substantiated or revised in the text to limit the scope of the conclusions to what is rigorously shown. Below I detail my comments and suggestions for improvements.

First, the authors characterized with several methods the expression and activity level of different proteasome subunits and lysosomal biomarkers, in the different glomerulus cell types. In their first statement, they claim the expression of ubiquitin and B5c is higher in human and murine podocytes. This is not observed in the figure.

Specifically, two podocytes are indeed high with ubiquitin. Yet, the only EC cell in the field of view also expresses ubiquitin. Interpretation based on one representative image is clearly not suitable for the statement made and certainly without quantification of numerous fields of view and cells. Further, the localization seems to be different between the human and murine cells, clearly observed for ubiquitin where in humans it looks nuclear and in the murine mostly not. The B5c expression also seems to be perinuclear in the murine EC cells (seen as yellow regions). Here, it does seem that the b5c is less expressed in EC and mc but cannot be concluded based on two or three cells. The same problem persists with determining the LC3-positive cells, in which the authors claim that their levels were balanced across all three types. Yet, in the human EC and mc, it is hardly stained. Also, podocytes in murine seem to express Lamp2, albeit cytosolic/vesicular, whereas humans do not. Caution is therefore warranted in re-

assessing the statements and inferences made in Fig 1B-C regarding the cell and species distribution of these markers based on IF. Quantification of more cells and secondary antibody control alone can help substantiate these findings. Clearly, confocal images may serve as exemplars but high-content imaging can help in generating the statistics required across numerous fields of view. Further, proteasome complex assemblies are inferred based on the level of a representative subunit. Multiple subunits should be examined by staining and complex composition can be inferred only by size exclusion chromatography or pulldown assays, after isolation of the different cell types. While Figure S1 (from a previous publication) reflects the levels of more subunits, one cannot compare between different cell types as the distribution will be different, therefore a low (blue) signal does not necessarily mean lower expression if not normalized. Normalizing by one of the structural units of the proteasome (PSMA1-7) is warranted.

Indeed in the qPCR analysis, Psmb5 is highly expressed in PC and the Limp2 transcript in the mc.

Then, the authors focused on the proteasomal system and examined immunoproteasome subunits. Again, here caution should be made with statements as only one subunit (i.e. B5i) represents a complex, and also its expression in murine samples is shown only for the IF (not qPCR and not wb. Thus, the statement “ depicting a conserved dissection of p5c and b5i expression among podocytes and GENCs across species should be avoided (see comment above regarding the proteomic data).

Regarding Figure 3 and the generation of the KO mice a control of secondary antibody alone should be used as while reduced, a signal against the b5i appears in the IF.

While several phenotypic aberrations are noted for ECs with Lmp7 knockout, the authors do not follow up or provide an explanation and move to focus on podocytes. In the electron microscopy image provided, proteasome inhibition seems to yield in broadening the nephrin meanders. Did the width of the leupeptin A were examined as well? They seem smaller (Fig. 4E).

The hypothesis that the increase in cell/glomerular size may be due to aggregation is intriguing. Yet, although the kidney cells are considered mostly terminally differentiated, the authors should consider including the results showing no effect on cell proliferation and add also cell cycle staining (PI) by FACS.

In Figure 5B a normalizing factor (based on a loading control) should be provided. Normalization of the untreated cells is not valid. Also, the wb actually shows higher levels in the leup than in the epox-treated cells. The altered distribution looks indeed different in C.

The pharmacological and genetic manipulations do not target the same proteasomal species, reducing the enthusiasm from all the drug-dependent effects in the manuscript. An immunoproteasome-specific inhibitor such as ONX-0914 should be examined to conclude directly on LMP7-dependent results. In addition, ex vivo experiments measuring antibody levels and secretion may be examined to substantiate the findings. A “rescue” experiment is warranted by induced expression of LMP7/ transfection of recombinant LMP7 and testing its impact on the IgG levels. This is a general note that hampered the ability to evaluate the contribution of the immunoproteasome on certain phenotypes (e.g. IgG-deposition and proteinuria) which are associated with the pathology, may have been examined by standard kits or from in vivo analysis and should be performed with an immunoproteasome-specific Ab.

In this regard, the authors also suggest in rows 447-458 that the EC secrete factors that are necessary for GBM formation and that this process is immunoproteasome-dependent, this should also be tested with an Lmp7-specific inhibitor.

Alternatively, instead of performing all of the experiments again with the relevant inhibitor, it is recommended to re-organize the text to limit the conclusions to proteasomal degradation and highlight a possible contribution of the immunoproteasome (where relevant). A follow-up on the striking effect of immunoproteasome on the GBM would have increased the strength and novelty of the paper regarding the pivotal role of Lmp7 for glomerulus (and EC-specific) homeostasis.

Minor points:

Of note, please use either PC, MC, and EC (or GEnC) or P,ec, mc consistently throughout the text across the paper and figures to help the reader (compare Fig1. B-C to E-F).

I would refrain from staining the terms “proteasomal-active” or lysosomal-active” cells based on immunofluorescence images. These systems are not mutually exclusive and may serve different dependencies based on different conditions.

2C highlights only the podocytes and not EC.

The calculation of the activity normalized to MC does not make sense to this reviewer. Why to normalize levels or activity across cell types? I would remove these graphs or explain the logic in the text. If at all, the relative distribution for the activity (bottom) shows a similar level in EC for both the constitutive or immunoproteasome activities- certainly not significant differences and therefore the conclusion that EC “distinguish themselves with a significant immunoproteasome activity” is problematic. It seems that ECs generally have more proteasomes expressed in them as seen by the levels of a2 and b5c (as well as more immunoproteasomes).

Fig. 2E-F should be presenting the GB514 as MVB003 is not specific to the immunoproteasome as well if the statement is related to immunoproteasome activity.

Further, the conservation between species based on staining in pig only against the probe which is for general proteasome activity should be removed and instead presented accurately such that pig glomeruli exhibited the highest proteasome activity in EC.

The Loading control of 3c seems too low and all sub-panels in Fig 3 should be labeled and referred to in the text (even if only quantitation).

Line 44- It is getting more and more clear – clearer.

Line 72- indirectly participate in participating.

Line 82 – for example, change to such as in... or in

Line 90 -especially the proteasome system, is unknown.

Reviewer #2 (Remarks to the Author):

The current paper examined the role of the proteasome system in the kidney glomerulus and show that endothelial immunoproteasome deficiency as well as proteasome inhibition disrupt the filtration barrier in mice, resulting in pathologic immunoglobulin deposition under the slit diaphragm and glomerular basement membrane alterations. Mechanistically, a reduced endocytic activity was identified, which relates to altered membrane recycling and turnover of endocytic receptors for collagen 4 and immunoglobulins.

The observation is novel, providing some new insight into renal physiology and pathology.

The key concern with the presented work is that observed phenotype with proteasome inhibition or genetic deletion is minimal at best. This is also consistent with clinical observations that kidney disease and proteinuria is not a major side effect of patients taking proteasome inhibitors. Overall casting some shadow on the significance or importance of the system.

It is also unclear whether change in proteasome system occur in kidney disease and whether this contributes to human pathology. Immunoglobulin deposition occurs in some forms of GN, but not uniformly observed.

The third key concern is that it lacks a clear mechanism that would link the proteasome to endocytosis and immunoglobulin accumulation. It is likely that a myriad of pathways are affected by proteasome inhibition, how do we know that immunoglobulin deposition is the most important one? How do we know that other pathways are not affected?

Figure 1 is a good analysis of the proteasome system but the team should analyze gene expression changes in published isolated glomeruli and single cell data in mice and humans. In addition, they should include the Deubiquitinating Enzymes (DUBs and Shuttling factors: These are proteins like Rad23 and Dsk2 that help in the delivery of the ubiquitinated protein to the proteasome.

Figure1F does not have a loading control.

The lysosome and ALP system is a bit more complicated to analyze, but again would suggest unbiased single cell studies, isolated glomeruli.

In addition, electron microscopical comparison would be needed to examine cell components.

The phenotype of the Lmp7 mice is mild at best. How do these animals behave under stress or following kidney disease?

Similarly mice treated with a variety of inhibitors only seems to show minimal proteinuria, but no glomerulosclerosis or kidney function decline. These observations strongly argue against the importance of the mechanism in kidney disease development.

Reviewer #3 (Remarks to the Author):

Thank you for the opportunity to review this manuscript by Wiebke Sachs. The authors revealed that podocytes rely on the constitutive proteasome, while endothelial cells rely on the immunoproteasome, as demonstrated through the isolation of glomerular cell types. The article also highlights that proteasome inhibition leads to the accumulation of immunoglobulin in glomeruli and provides evidence that it decreases endocytosis activity, affecting recycling and turnover. While there are other papers discussing proteasome and glomerular cells, it is novel that the authors investigated the topic from the viewpoint of endocytosis. This evidence is crucial in understanding glomerular disease and proteasome function.

However, there are some concerns and points that need to be corrected, listed below:

Major

#1 Figure 4 and Suppl Figure 3

The author administered the proteasome inhibitor systemically via intra-peritoneal injection instead of using a genetic model. Were there any symptoms such as diarrhea or weight loss that could have affected Cre or BUN? Regardless of the presence or absence of symptoms, the author should mention this in the study. Additionally, I am curious if the three types of cells were similarly affected by epoxomicin. To address this, I would suggest the authors conduct an experiment wherein they administer epoxomicin to the three types of primary cultured cells and evaluate the proteasome activity or ubiquitin accumulation.

#2 Supplemental 3 and 5

As p62, LC3, and Limp2 expression did not change between vehicle and epoxomicin, I understand that the authors concluded that lysosomes did not compensate for proteasome inhibition. However, since these expressions can be affected by both autophagy and lysosomes, I believe the authors should evaluate these molecules through the autophagy-lysosome flux. Furthermore, since the authors used glomerular lysates in Western blotting, the results do not solely reflect podocyte changes.

Generally, proteasome and autophagy-lysosome are considered to be related and can compensate for each other. To conclude that autophagy-lysosome did not compensate for proteasome inhibition, more experiments are required. To evaluate lysosome-autophagy flux accurately, the authors should inhibit autophagic flux in vivo and assess how LC3II and p62 respond to both the vehicle and epoxomicin.

#Figure6

Endocytosis can be influenced by acid-base conditions. Therefore, isn't it possible that the accumulation of ubiquitin and oxidized proteins, due to the inhibition of the proteasome, makes the intracellular environment more acidic, thereby affecting endocytosis?

Additionally, in Figure 6A, why is the signal of transferrin at time 0 so high in the Epoxomisin group, while the signal is very low in the vehicle group? Transferrin uptake in the epoxomicin group appears to decrease over time.

To analyze endocytosis under proteasome inhibition conditions, I would suggest the authors to attempt a pulse-chase experiment using transferrin-rhodamine, IgG-fluorescein. By comparing the uptake rates and patterns of each molecule, we should be able to gain a more detailed understanding of the kinetics of endocytosis within the cell.

Minor

#1 Figure 1C

The proteasome is expected to be localized in the nuclei more than in the cytoplasm. However, the staining of Beta5c (Psmb5) shows a significantly higher amount of proteasome in the cytoplasm compared to the nucleus. This result differs slightly from the one in Nature Communications volume 14, Article number: 2114 (2023), Figure 7c, which I consider to be the correct stain.

#2 Suppl 3D

Why does the beta-Hexosaminidase activity increase in lysosomal inhibition by leupeptin A in suppl Fig3D? Doesn't leupeptin A inhibit Hexosaminidase activity?

#3 Suppl Fig 6

Can the accumulated IgG lead to nephritis, or does it only accumulate in the glomerulus?

Dear reviewers

Thank you for the thorough and positive assessment of our manuscript. We have addressed every comment, performed extensive new experiments that were in the scope of the allocated timeframe of 3 months, and included new data. To assure a comfortable review of our rebuttal, we compiled a “**summary file of reviewer figures 1-17**” with corresponding figure legends. We feel that the manuscript has much improved thanks to the reviewers’ comments and hope that it is now suitable for publication in *Nature Communications*. Please find below our point-by-point response to the individual concerns raised. All changes within the manuscript and the supplemental appendix are highlighted in red.

Thank you for your enthusiasm, sincerely,
Catherine Meyer-Schwesinger, for the authors

Point-by-point response

REVIEWER 1

In the presented manuscript, Sachs et al., explore the role of proteasomes in maintaining homeostasis of the glomerular filtration barrier and prohibiting protein deposits from accumulating in the glomerulus. Importantly, the authors analyzed the three resident glomerular cell types in parallel to dissect the involvement of different degradation systems in their function. Indeed, understanding the role of both proteasome and lysosomal proteolysis across different cell types within a tissue remains an under-studies area of research and the authors provide new insight into a specialized role for these systems in cell type-specific and proteasome-specific manner. Further, the authors identified a role for the immunoproteasome subunit Imp7 and the constitutive proteasome in diseases associated with glomerular protein accumulation.

The authors found a striking and novel effect of Imp7 knockout on the GBM, showing how immunoproteasome subunit knockout can affect the GBM, and increase sedimentation of immunoglobulins. The authors also focus on the aggregation of antibodies in the kidneys, a highly relevant clinical question in the context of poorly understood and highly morbid membranous and glomerular nephropathy. They show that endosomal uptake is disrupted by proteasome inhibition and genetic manipulation of the immunoproteasome. The authors show that treatment with proteasome inhibitors causes a decrease in the kidneys' ability to internalize proteins via a reduction in endocytic receptor abundance. However, mechanistic aspects of how the immunoproteasome exerts its effect on the kidney (as suggested in the discussion) were not described. Yet, this work offers an important insight into proteostasis control mechanisms at the kidney that would be of broad interest to the scientific community. Several of their key points need to be further substantiated or revised in the text to limit the scope of the conclusions to what is rigorously shown. Below I detail my comments and suggestions for improvements.

Thank you for the thorough and positive assessment of our study. We hope to have adequately addressed the raised issues in the following paragraphs.

First, the authors characterized with several methods the expression and activity level of different proteasome subunits and lysosomal biomarkers, in the different glomerulus cell types. In their first statement, they claim the expression of ubiquitin and B5c is higher in human and murine podocytes. This is not observed in the figure. Specifically, two podocytes are indeed high with ubiquitin. Yet, the only EC cell in the field of view also expresses ubiquitin. Interpretation based on one representative image is clearly not suitable for the statement made and certainly without quantification of numerous fields of view and cells.

We agree that the histological investigations shown in **Fig. 1B** and **1C** are only suitable to depict the expression pattern of the stained proteins in between the glomerular cell types and do not allow a determination of the absolute protein abundances. This was clarified in the result section: “Therefore, comparative analyses of ubiquitin proteasome (UPS) and autophagosome-lysosome (ALP) marker protein (**Fig. 1A**) expression patterns were performed in human and murine glomerular cell types.” Further, we changed our wording from “abundance” to “expression pattern” in the description of the histological findings.

To accommodate the correct critique that the depicted single cells might not adequately reflect the complete glomerular situation, we added new **Suppl. Figs. 1** and **2** that include the overview pictures of the cropped images of **Fig. 1B** and **1C**. There, it can be appreciated that the cropped images reflect the general glomerular expression pattern and that podocytes have a strong immunosignal to ubiquitin and $\beta 5c$. The fact that GEnCs also have a prominent ubiquitin expression pattern was clarified in the result text: “High-resolution confocal analyses revealed an accentuated abundance of ubiquitin in podocytes and GEnCs and of the main proteolytic $\beta 5c$ subunit of the c20S proteasome in podocytes, a finding appreciable in healthy human (**Fig. 1B**, **Suppl. Fig 1**) as well as murine (**Fig. 1C**, **Suppl. Fig 2**) glomeruli.”

Determination of absolute protein levels between glomerular cell types requires quantitative techniques such as immunoblotting. This is shown in **Fig. 1F** for murine glomerular cells for $\beta 5c$ and other proteins. We now added an additional immunoblot to K48-polyubiquitinated

Reviewer Figure 1: Abundance of K48-polyubiquitin in bulk-isolated glomerular cells by immunoblotting. Equal cell numbers of podocytes (PC), mesangial cells (MC), and glomerular endothelial cells (EC) were loaded. Graph exhibits relative densitometric analysis of cells as percentage of MC levels, mean \pm SEM, One-Way ANOVA, n = 4 – 5.

proteasome substrates and of $\beta 1c$ to the original **Fig. 1F**. Please refer to **Reviewer Fig. 1** and **5** for these new additions to the figure. The result text and figure legend were adjusted accordingly.

Further, the localization seems to be different between the human and murine cells, clearly observed for ubiquitin where in humans it looks nuclear and in the murine mostly not.

Currently, we don't think that our data support a clear difference in subcellular ubiquitin expression pattern within human and mouse glomerular cell types. As seen in the overview micrographs of **Reviewer Fig. 2**, nuclear ubiquitin is also found in murine glomerular cells and cytoplasmic ubiquitin is also found in human glomerular cells. The differences of subcellular distribution could relate to differences in metabolic states of individual cells.

Reviewer Fig. 2: Distribution of ubiquitin (green) by high-resolution confocal images in a healthy human and murine glomerulus in relation to the slit diaphragm protein nephrin (red) and DNA (blue); pc = podocyte, mc = mesangial cell, ec = glomerular endothelial cell, white arrows point towards cytoplasmic ubiquitin and red arrows point towards nuclear ubiquitin.

The B5c expression also seems to be perinuclear in the murine EC cells (seen as yellow regions). Here, it does seem that the b5c is less expressed in EC and mc but cannot be concluded based on two or three cells.

Substantiated *via* the quantitative approaches of qPCR and immunoblotting shown in **Fig 1E** and **F**, indeed, the β 5c mRNA and protein abundance is lowest in GEnCs compared to podocytes and mesangial cells as reflected by the chosen histological panels of **Fig. 1B** and **C** and now additionally shown for more cells in the new **Suppl. Fig. 1** and **2**. Whether the expression pattern is perinuclear or cytoplasmic in GEnCs is difficult to say at this point, as GEnCs only have a very slim cytoplasm.

The same problem persists with determining the LC3-positive cells, in which the authors claim that their levels were balanced across all three types. Yet, in the human EC and mc, it is hardly stained.

We agree and were more precise with our description within the result section: “The abundance of LC3-positive autophagosomes was balanced between all three resident murine glomerular cell types whereas in humans, podocytes exhibited a prominent LC3 staining pattern.”

Also, podocytes in murine seem to express Lamp2, albeit cytosolic/vesicular, whereas humans do not.

To this end, we did not include LAMP2 staining of the human samples, as the 3 antibodies tested did not result in a convincing staining of LAMP2⁺ vesicles in the human paraffin embedded and frozen material. Therefore, LAMP1 stainings of the human samples are shown in **Fig. 1B**. The expression pattern of LAMP1 was highly abundant in PECs and appears balanced between podocytes, mesangials, and GEnCs. We added the sentence: “Parietal epithelial cells exhibited a strong LAMP1 expression pattern.”

Caution is therefore warranted in re-assessing the statements and inferences made in Fig 1B-C regarding the cell and species distribution of these markers based on IF. Quantification of more cells and secondary antibody control alone can help substantiate these findings. Clearly, confocal images may serve as exemplars but high-content imaging can help in generating the statistics required across numerous fields of view.

Quantification was performed using immunoblotting within the murine samples as histology is at most semiquantitative. To address this reviewers' concerns, immunoblot analyses to K48-polyubiquitin was added to **Fig. 1F**. Secondary antibody controls are standard in our lab and shown in **Reviewer Fig. 3A-D**. We attempted to histologically perform semiquantitative analyses of the cell distribution of target proteins, as suggested, by colocalization with cell-type specific markers such as synaptopodin for podocytes and endomucin for endothelial cells. We unfortunately encountered multiple problems as shown in **Reviewer Fig. 3E**. As the labeling of GEnCs with endomucin does not label the cytoplasmic/nuclear region of the cells, the proteins of interest are not detected in the cytoplasm/nucleus. Additionally, there is an overlap between endomucin and synaptopodin labeling due to the complex 3-dimensional architecture of the glomerular convolute, making it challenging to clearly quantify signals in the cell types, especially in the very delicate GEnCs, which together falsifies the GEnC measurements (**Reviewer Fig. 3E**).

Reviewer Fig. 3: Confocal images of ubiquitin (A), K48-polyubiquitin (K48pUB) (B), β 5c (C) and β 5i (D) distribution in a healthy murine and human glomerulus. All proteins of interest are shown in green in relation to the slit diaphragm protein nephrin (red) and DNA (blue). Lower panel rows depict the secondary antibody only negative controls (w/o $^{\circ}$ 2Ab) to each staining condition. (E) Exemplary attempt to differentiate signals in glomerular endothelial cells *via* endomucin staining (light blue) and podocytes *via* synaptopodin staining (red). A GEnC mask is generated based on endomucin signal. Overlay of this GEnC mask over DNA staining demonstrates “failure” to integrate the cytoplasmic/nuclear fraction of GEnCs within the mask (problem 1). Overlay of the GEnC mask over the synaptopodin staining demonstrates inclusion of podocyte signals within the GEnC mask (problem 2). Together, these limitations falsify the histological GEnC measurements.

Further, proteasome complex assemblies are inferred based on the level of a representative subunit. Multiple subunits should be examined by staining and complex composition can be inferred only by size exclusion chromatography or pull-down assays, after isolation of the different cell types. While Figure S1 (from a previous publication) reflects the levels of more subunits, one cannot compare between different cell types as the distribution will be different, therefore a low (blue) signal does not necessarily mean lower expression if not normalized.

Normalizing by one of the structural units of the proteasome (PSMA1-7) is warranted. Indeed in the qPCR analysis, Psmb5 is highly expressed in PC and the Limp2 transcript in the mc.

The reviewer is correct that different expression values of proteasome values might be a function of total protein abundance and, specifically, proteasome abundance. As common for proteomics data, we here normalize the total protein signals to total cell protein levels and MS signals using a label-free quantification proteomics approach. Thus, the data are quantified and normalized according to total MS signal using a label-free quantitative proteomic approach using MaxLFQ,² a method cited over 4000 times. We agree that this might have been a misunderstanding from our description of the data, thus we have further cited and added additional information to **Supplemental Fig 3, 6 and 14** and have added color bars that make this point clear. Following sentence was added to the figure legends: “Protein values were obtained by label-free quantification results using the MaxQuantLFQ algorithm <https://pubmed.ncbi.nlm.nih.gov/24942700/>. The heatmaps depict euclidian distance clustering...”.

We also recalculated the proteasome subunit abundance in murine bulk-isolated glomerular cell types following normalization to Psma1 of the structural proteasome 20S core particle (**Reviewer Fig. 4**). The differences between podocytes and GEnCs regarding proteolytic constitutive and standard proteasome subunit abundance become even more apparent. Therefore, this was added as a new panel **A** to **Fig. 2**.

Reviewer Fig. 4: Comparison of protein abundance in murine bulk-isolated glomerular cell types using (A) proteomic label-free quantification (original **Suppl. Fig. 3A**) and (B) proteomic label-free quantification normalized to Psma1 of the structural proteasome 20S core particle; podocytes (PC), mesangial cells (MC) and glomerular endothelial cells (EC).

Then, the authors focused on the proteasomal system and examined immunoproteasome subunits. Again, here caution should be made with statements as only one subunit (i.e. B5i) represents a complex, and also its expression in murine samples is shown only for the IF (not qPCR and not wb. Thus, the statement “depicting a conserved dissection of p5c and b5i expression among podocytes and GEnCs across species” should be avoided (see comment above regarding the proteomic data).

We apologize for the misunderstanding, but the abundance of $\beta 5i$ in the murine samples was not only shown by immunofluorescence (**Fig. 2D**), but also by immunoblotting (**Fig. 2B**) and qPCR (**Fig. 2C**). The protein quantification of $\beta 5i$ in the human material is currently not feasible beyond immunofluorescence. We, however, now included re-analyses of proteasome subunit expression levels from published human glomerular single cell RNAseq analyses (KPMP

dataset), that are now included as a new **Suppl. Fig 3C**. The clear presence of all proteolytic immunosubunits in GENCs in comparison to podocytes and mesangial cells can be appreciated in human GENCs, a finding further supported on the transcript (**Suppl. Fig. 3B**) and protein level in mice using our proteomic approach (**Suppl. Fig. 3A**).

Additionally, as both, $\beta 5c$ and $\beta 5i$ are the predominant/main proteolytic subunits of the constitutive ($\beta 5c$) or of the immunoproteasome ($\beta 5i$), we think this reductionist approach is valid based on technical challenges: The simultaneous detection of all proteolytic β -subunits is not feasible in the very limited protein material of bulk-isolated glomerular cells, as they all run at approximately the same height by immunoblotting, thus necessitating parallel sample loading on multiple membranes for a clean detection. Further, abundance of the $\beta 1c$, $\beta 1i$, $\beta 2c$, $\beta 2i$ subunits is much lower than of the $\beta 5c/i$ subunits, making their detection very challenging in this biological material. This was better explained in the result section: “The proteasome is a highly versatile multi-subunit complex, which exists in different constitutions depending on the proteolytic subunits incorporated within the 20S core particle, especially of the main proteolytic $\beta 5$ subunits. Cells exposed to oxidative stress and/or inflammatory stimuli have been shown to induce the immunoproteasome (marked by the main proteolytic $\beta 5i$ subunit), which differs from the constitutive proteasome by an enhanced proteolytic efficiency and a different cleavage specificity, thus generating peptides best for MHC-I presentation (Basler et al., 2013).”

To accommodate this reviewer, we attempted to detect the other proteolytic β -subunits in new preparations of bulk isolated glomerular cell types and unfortunately only succeeded for $\beta 1c$ (**Reviewer Fig. 5**). Again, $\beta 1c$ abundance was highest in podocytes. These data were not included to the **Fig. 2B**, as we cannot show corresponding high quality immunoblots to $\beta 1i$, $\beta 2c$, or $\beta 2i$. However, they were added as new data to **Fig. 1F**.

In our view, the overall combination of all applied techniques lets us state that “Together, these analyses depict a conserved dissection of $\beta 5c$ and $\beta 5i$ expression among podocytes and GENCs across species.” Hence, we would like to keep this modified statement within the manuscript.

Reviewer Figure 5: Abundance of the constitutive proteasome subunits $\beta 1c$, $\beta 5c$, of the immunoproteasome subunit $\beta 5i$ and of the 20S core complex subunit $\alpha 2$ in bulk-isolated glomerular cells by immunoblotting. Equal cell numbers of podocytes (PC), mesangial cells (MC), and glomerular endothelial cells (EC) were loaded. Graphs depict densitometric quantification of protein abundances normalized to MCs within individual experiments. Values are depicted as mean \pm SEM, pooled values of 4 independent experiments are shown for $\beta 5c$, $\beta 5i$ and $\alpha 2$, and of 2 experiments with $n=4$ biological samples for $\beta 1c$; * $p<0.05$, One Way ANOVA.

Regarding Figure 3 and the generation of the KO mice a control of secondary antibody alone should be used as while reduced, a signal against the $\beta 5i$ appears in the IF.

As shown in **Reviewer Fig. 6**, the negative control using secondary antibodies alone shows no unspecific staining signal at the same laser intensities of the confocal microscope. Of note, the KO model analyzed in **Fig. 3** is an endothelial cell specific KO of *Lmp7*, hence a residual histological *Lmp7* signal is present from other non-endothelial cells within glomeruli, whilst the

endothelial-derived Lmp7 signal is completely gone in **Fig. 3B**. Additionally, as the immunoblots in **Fig. 3C** depict, endothelial Lmp7 knockout is complete.

Reviewer Fig. 6: High-resolution confocal images of β5i knockout was evaluated by high-resolution confocal microscopy to β5i (green) in relation to nephrin (red) and DNA (blue) in glomeruli from *Lmp7^{ΔEnC}*. A negative control using secondary antibody alone (w/o 1°Ab), Cy2-rbIgG (green) and Cy3-gpIgG (red) shows absence of unspecific staining from the secondary antibodies used.

While several phenotypic aberrations are noted for ECs with Lmp7 knockout, the authors do not follow up or provide an explanation and move to focus on podocytes. In the electron microscopy image provided, proteasome inhibition seems to yield in broadening the nephrin meanders. Did the width of the leupeptin A were examined as well? They seem smaller (**Fig. 4E**).

We have now quantified the filtration slit density (FSD) using the Podocyte Exact Morphology Measurement Procedure (PEMP) analysis. The FSD in the leupeptin A treated mice is indeed higher than in vehicle treated mice, demonstrating narrower nephrin meanders. In the course of these investigations, we also included the epoxomicin treated mice, for completeness. FSD of epoxomicin treated mice was not lower than of vehicle mice, even though these mice exhibit significant proteinuria. As focal areas of foot process effacement are visible by EM and super-resolution microscopy of nephrin, we conclude, that the leakiness of the GFB in epoxomicin treated mice results from focal broadening of nephrin meanders.

The PEMP analysis of leupeptin and epoxomicin-treated mice was added as new **Suppl. Fig. 9B** to the manuscript. We changed the text within the result section as follows: “Ultrastructural investigations further confirmed that podocytes were affected by proteasome inhibition, as nephrin meanders were focally broadened by high-resolution microscopy (**Fig. 4E**) and foot processes focally effaced by electron microscopy (**Fig. 4F**, panel 1). Interestingly, nephrin meanders in leupeptin A treated mice were tighter than in vehicle treated mice (**Suppl. Fig. 9B**).”

The hypothesis that the increase in cell/glomerular size may be due to aggregation is intriguing. Yet, although the kidney cells are considered mostly terminally differentiated, the authors should consider including the results showing no effect on cell proliferation and add also cell cycle staining (PI) by FACS.

Thank you for this comment! We set up cell cycle analyses using propidium iodide measurements by FACS in bulk-isolated GEnCs from *Lmp7^{ΔEnC}* mice, as suggested, as there the increase in cell size was the most obvious phenotypic alteration. However, no differences in cell cycle phase distributions were discernible between control and *Lmp7^{ΔEnC}* GEnCs. These data were included as a new panel **F** to **Fig. 3** and a sentence was added to the result section: “To this end, GEnCs isolated from *Lmp7^{ΔEnC}* kidneys exhibited an increased cell size compared to control GEnCs (**Fig. 3E**), which was not due to differences in the cell cycle state of the cells (**Fig. 3F**).” To be able to fit these data to the figure, we removed the graph with the densitometric quantification of K48-polyubiquitin in GEnCs, as there was no significant difference between control and *Lmp7^{ΔEnC}* cells, as clearly seen in the immunoblot and added to the result text: “GEnC β5i-deficiency did not translate to an enhanced abundance of proteasome substrates (lysine-48-polyubiquitinated proteins) by immunoblotting (densitometric analysis not shown)...”.

In Figure 5B a normalizing factor (based on a loading control) should be provided. Normalization of the untreated cells is not valid. Also, the wb actually shows higher levels in the leup than in the epox-treated cells. The altered distribution looks indeed different in C. Unfortunately, there is no other adequate way to normalize this experiment than the one provided. We pull the rabbit IgG *via* protein A/G resin from a defined (counted) number of isolated glomeruli per mouse. So, normalization occurs at the level of glomerular number. The pulled rabbit IgG is then eluted from the extensively washed protein A/G resin and quantified by immunoblotting. Due to this approach, a loading control on the same immunoblot is technically not providable. This explanation was added to the method section: "As the experiments depict the amount of rabbit IgG that was pulled down from the same number of glomeruli, an immunoblot loading control is technically not providable.". We changed the presentation of **Fig. 5B** by adding the information "pulldown from number-adapted glomeruli" underneath the experiment and we replaced the immunoblot by an immunoblot from another experiment. Of note: the mesangial accumulation of rblgG is enhanced in leupeptin A treated mice in comparison to vehicle treated mice, hence the pulldown for rblgG indeed shows overall higher glomerular rblgG levels in leupeptin A than in vehicle mice, albeit not as high as under epoxomicin exposure.

The pharmacological and genetic manipulations do not target the same proteasomal species, reducing the enthusiasm from all the drug-dependent effects in the manuscript. An immunoproteasome-specific inhibitor such as ONX-0914 should be examined to conclude directly on LMP7-dependent results.

Thank you for the comment. We performed immunoproteasome specific inhibitor experiments in a set of 6 mice using the immunoproteasome β 5i-subunit specific inhibitor ONX0914, as suggested (**Reviewer Fig. 7**). However, inhibition using ONX0914 was not restricted to the β 5i subunit but also affected other proteasome subunits such as β 1c and β 2c. Hence, these experiments were not included to the manuscript. Again, use of ONX0914 resulted in leakiness of the GFB to albumin, corroborating the importance of proteasome functionality for the GFB.

The rationale behind the use of epoxomicin was that we wanted to affect the constitutive proteasome in podocytes, as healthy podocytes do not express significant amounts of the immunoproteasome. This was explained in the result section: "Healthy podocytes do not express the i20S in a detectable manner, hence a podocyte-specific knockout of β 5i did not result in morphologic and functional glomerular/podocyte alterations under homeostatic conditions (data not shown). Genetic targeting of β 5c is not feasible to date, therefore we set out to modulate constitutive β -subunit activities by chemical modulation using epoxomicin. This pan-proteasome inhibitor binds covalently to all proteolytic β -subunits, with a prevalence for the β 5c- and β 5i-subunits (Kim and Crews, 2013)."

Reviewer Fig. 7: Male BALB/c mice were treated with either vehicle or ONX-0914 (10 mg/kg bodyweight) on four consecutive days. Urine was collected for the assessment of glomerular filtration barrier integrity and glomeruli were isolated for biochemical analyses. **(A)** In-gel activity using the pan-proteasomal activity-based probe MVB003 in isolated glomeruli of experimental mice. Graphs depict densitometric analysis of the prominent β 2c and β 5c/ β 5i activities in relation to α 2 amounts, values are shown as mean \pm SEM, percent of vehicle. **(B)** Albumin/creatinine ratio of vehicle and ONX-0914 treated mice after 4 days of treatment.

In addition, *ex vivo* experiments measuring antibody levels and secretion may be examined to substantiate the findings. A "rescue" experiment is warranted by induced expression of LMP7/transfection of recombinant LMP7 and testing its impact on the IgG levels.

We apologize, but we do not know what is meant by the point of “ex vivo experiments measuring antibody levels and secretion.”

Concerning the comment on “rescue experiments”: *In vivo* rescue experiments that would ensure an induced rescue expression of Lmp7 in mice with underlying Lmp7 knockout in endothelial cells is technically not feasible in our hands, especially not in the allocated time of 3 months for this review, as for one we don't have the required S2-permission to do this kind of experiment and secondly, specific GEnC transduction *in vivo* is not readily established. We apologize for this shortcoming.

To accommodate this reviewer, we attempted rescue experiments in cultured human *PSMB8* (CRISPR/Cas9) knockout endothelial cells, which unfortunately repeatedly failed, as LMP7 expression was not reestablished upon transient transfection with an *PSMB8*-encoding vector. Therefore, we cannot provide these data in the given timeframe for this rebuttal. We are sorry for this. Nonetheless, these experiments substantiated an endo-lysosomal phenotype of human LMP-7 deficient endothelial cells including RAB5 alterations, and especially alterations of the lysosomal system proteins with LAMP2 and LIMP2 (please refer to **Reviewer Fig. 12 and 13**).

This is a general note that hampered the ability to evaluate the contribution of the immunoproteasome on certain phenotypes (e.g. IgG-deposition and proteinuria) which are associated with the pathology, may have been examined by standard kits or from *in vivo* analysis and should be performed with an immunoproteasome-specific Ab. In this regard, the authors also suggest in rows 447-458 that the EC secrete factors that are necessary for GBM formation and that this process is immunoproteasome-dependent, this should also be tested with an Lmp7-specific inhibitor. Alternatively, instead of performing all of the experiments again with the relevant inhibitor, it is recommended to re-organize the text to limit the conclusions to proteasomal degradation and highlight a possible contribution of the immunoproteasome (where relevant). A follow-up on the striking effect of immunoproteasome on the GBM would have increased the strength and novelty of the paper regarding the pivotal role of Lmp7 for glomerulus (and EC-specific) homeostasis.

We thank this reviewer for all the fruitful comments, which we were mostly able to adequately address. Concerning the last comment, as recommended, the discussion was reorganized where relevant:

- 1) “Hence the observed prominent GBM splitting in endothelial $\beta 5i$ -deficiency as well as in global proteasome inhibition could reflect an altered secretion/removal of ECM material by podocytes and GEnC in a proteasome-dependent manner, resulting in GBM alteration and subsequently to podocyte disturbance.”
- 2) “To this end, our total transcript/protein analyses in conjunction with immunolocalization and membrane biotinylation assays could describe altered endocytic receptor localization, trafficking (neonatal Fc-receptor, FcRN), as well as membrane abundance (Mrc2) to underlie the endocytic alteration of podocytes and GEnCs with impaired proteasome functionality.”
- 3) “Especially $\beta 5i$ -deficient GEnCs exhibited a prominent intracellular mouse IgG accumulation in combination with an intracellular FcRN retention clearly showing a role of the proteasome in membrane protein recycling.”

Minor points:

Of note, please use either PC, MC, and EC (or GEnC) or P,ec, mc consistently throughout the text across the paper and figures to help the reader (compare Fig1. B-C to E-F).

This was corrected throughout the figures.

I would refrain from staining the terms “proteasomal-active” or “lysosomal-active” cells based on immunofluorescence images. These systems are not mutually exclusive and may serve different dependencies based on different conditions.

We agree and changed the result text sections to: “As illustrated within the radar plot (**Fig. 1D**), our analyses support the view of murine podocytes as cells with a high proteasome abundance.” and “Corroborating the histological analyses, single cell RNAseq and bulk proteomic analysis of ALP marker proteins also revealed a glomerular cell type distinct clustering in mice and humans (**Suppl. Fig. 6**) supporting the view of GEnCs as cells with endosomal abundance, and mesangial and podocytes as cells with abundance in lysosomes.”

2C highlights only the podocytes and not EC.

The arrows within **Fig. 2D** highlight the *Lmp7* expression within GEnCs. For more clarity, GEnC nuclei are now additionally indicated by “ec” within the figure.

The calculation of the activity normalized to MC does not make sense to this reviewer. Why to normalize levels or activity across cell types? I would remove these graphs or explain the logic in the text.

In **Fig. 2E** protein abundance and activity measurements were quantified from 4 independent glomerular cell type sorts. To pool the individual experiments for graphical representation, an internal normalization is required. MCs were chosen as a reference point within the individual experiments, as 1) they had the lowest proteasome protein abundance and activity and 2) focus of the figure were podocytes and GEnCs.

For simplification, we followed this reviewers’ advice and removed most of the graphs from the figure and result text, leaving only the quantification of the MVB003 signal, and compiled a new **Fig. 2**.

If at all, the relative distribution for the activity (bottom) shows a similar level in EC for both the constitutive or immunoproteasome activities- certainly not significant differences and therefore the conclusion that EC “distinguish themselves with a significant immunoproteasome activity” is problematic. It seems that ECs generally have more proteasomes expressed in them as seen by the levels of a2 and b5c (as well as more immunoproteasomes).

We apologize for the misunderstanding and clarify our conclusion: “Taken together our data show that within the glomerular syncytium GEnCs and podocytes have the strongest proteasome activity. Whereas podocytes exhibit mostly constitutive proteasome-related activity, GEnCs distinguish themselves with an additional immunoproteasome activity.”

Fig. 2E-F should be presenting the GB514 as MVB003 is not specific to the immunoproteasome as well if the statement is related to immunoproteasome activity.

We agree. However, GB514 is barely cell permeable, hence this experimental setup of *ex vivo* immunoproteasome activity measurement did not result in specific activity measurements in our hands and was therefore not included to the manuscript. This information was added to the method section. The *ex vivo* use of MVB003 (**Fig. 2F**) resulted in specific activity measurements in isolated glomeruli. This experiment was included to the manuscript to demonstrate that the results of abundant proteasome activity in GEnCs and podocytes shown in **Fig. 2B-E**, were not a technical artifact but could in essence be reproduced with another technique. One must keep in mind that bulk isolation of glomerular cell types is a mechanically harsh and time-consuming technique which potentially could alter proteasome activities.

Further, the conservation between species based on staining in pig only against the probe which is for general proteasome activity should be removed and instead presented accurately such that pig glomeruli exhibited the highest proteasome activity in EC.

We apologize for the misunderstanding. The conservation between species is not only based on staining in pig, but rather on the comparable findings between human – mouse – pig in respect to proteasome transcript/protein expression and activity. To accommodate this reviewer, we removed: “This proteasome distribution is conserved between species.” from the overall conclusion of this paragraph.

The Loading control of 3c seems too low and all sub-panels in Fig 3 should be labeled and referred to in the text (even if only quantitation).

The loading control in **Fig. 3C** is not low, only a very short exposure was shown. We replaced this by a longer exposed β -actin. We included referrals to the quantitation subpanels in the result text and within the figure legends.

Line 44- It is getting more and more clear – clearer. We apologize, but we cannot find this in the text.

Line 72- indirectly participate in participating.

This was changed to: “Mesangial cells (MCs) are contractile cells that constitute the central stalk of the glomerulus to provide structural support and, as specialized pericytes, indirectly participating in filtration by reducing the glomerular surface area by contraction.”³

Line 82- for example, change to such as in... or in

This was changed to: “...glomerular cell types themselves might contribute to glomerular protein clearance, such as MCs by phagocytosis...”

Line 90 -especially the proteasome system, is unknown.

We apologize, but is it not clear to us, what we should change here.

REVIEWER 2

The current paper examined the role of the proteasome system in the kidney glomerulus and show that endothelial immunoproteasome deficiency as well as proteasome inhibition disrupt the filtration barrier in mice, resulting in pathologic immunoglobulin deposition under the slit diaphragm and glomerular basement membrane alterations. Mechanistically, a reduced endocytic activity was identified, which relates to altered membrane recycling and turnover of endocytic receptors for collagen 4 and immunoglobulins.

The observation is novel, providing some new insight into renal physiology and pathology. Thank you for the thorough and positive assessment of our study. We hope to have adequately addressed the raised issues in the following paragraphs.

The key concern with the presented work is that observed phenotype with proteasome inhibition or genetic deletion is minimal at best. This is also consistent with clinical observations that kidney disease and proteinuria is not a major side effect of patients taking proteasome inhibitors. Overall casting some shadow on the significance or importance of the system.

Even though we respect this assessment, we beg to disagree. In general, inducing GFB alterations in mice is quite challenging. As such many groups over the last decades had to experience that knockout of major proteins frequently fail to result in a GFB phenotype in mice. Here, pan-proteasome inhibition for 4 days results in both significant functional and morphologic alterations of the GFB. This does not occur if for example autophagy is genetically impaired⁴, mitochondrial biology is genetically substantially altered⁵, or if lysosomal function is severely disrupted⁶. Further, the genetic deletion of *Lmp7* in endothelial cells in this study represents in our opinion a “subtle” alteration of proteasome constitution, as all other 5 proteolytic subunits are present and expressed. Nonetheless, *Lmp7* Δ^{EnC} mice exhibit essential morphologic GFB alterations.

We agree that until now kidney disease and proteinuria are not recognized major side effects of patients taking Velcade. They have, however, been reported in patients receiving Carfilzomib as mentioned in the discussion, resulting in discontinuation of treatment in these

patients. Velcade is most frequently used to treat patients with multiple myeloma and is rapidly taken up by the tumor cells potentially resulting in a low exposure of glomerular cell types to this drug (oral communication Prof. E. Krüger, Greifswald) thus potentially precluding negative side-effects on glomerular function. Carfilzomib has a different bioavailability than Velcade.

It is also unclear whether change in proteasome system occur in kidney disease and whether this contributes to human pathology.

This lack of knowledge exactly justifies our basic investigations. In the past we could demonstrate proteasome system changes in rats, mice, and patients with glomerular disease,⁷⁻¹⁰ justifying our current attempt to provide a basic understanding of the physiologic significance of this complex degradative system for glomerular cell function. An example is shown in **Reviewer Fig. 8** in human ANCA⁺ GN, where a significant induction of the immunoproteasome can be appreciated, not only in the infiltrating immune cells but also in resident glomerular cells, especially in podocytes, suggesting an involvement in human glomerular pathology.

Reviewer Figure 8: Representative confocal images in human biopsy specimens. LMP7 (green) expression is restricted to GEnCs (white arrows) in the healthy control biopsy. In anti-neutrophil cytoplasmic antibody (ANCA)+ glomerulonephritis (GN), LMP7 expression is additionally found in podocytes (red arrows) and in periglomerular leukocytes (yellow arrows). Nephrin (red) demarcates the glomerular filtration barrier, DNA is stained with Hoechst (blue).

Immunoglobulin deposition occurs in some forms of GN, but not uniformly observed.

This is correct. Future investigations will hopefully provide insight into the underlying (differing) mechanisms of GN, the mechanisms underlying immunoglobulin deposition and the extent and type of proteasomal involvement in these processes.

The third key concern is that it lacks a clear mechanism that would link the proteasome to endocytosis and immunoglobulin accumulation.

We agree that we did not provide a specific single link of the proteasome to endocytosis and immunoglobulin accumulation at this point. We now added data demonstrating, that in addition to altered plasma membrane dynamics and plasma membrane levels of endocytic receptors, proteasome inhibition results in a reduced trafficking of the FcRN from the perinuclear region to the cell border in pulse chase experiments in primary podocytes (new **Fig. 7C**).

As the uptake of multiple types of cargoes are affected, the effects of proteasome alterations on endocytosis can be attributed to a multitude of mechanisms, beyond the turnover/trafficking of plasma membrane receptors we have been focusing on here. To name a few aspects we are currently considering:

- Altered expression of actin-cytoskeletal proteins
- Altered plasma membrane dynamics.

- Altered expression/turnover of membrane receptors at the plasma membrane.
- Alterations of the endo-lysosomal pathway (especially of the late endo-lysosomal pathway).

These considerations are based on bulk transcriptomic (see **Reviewer Fig. 9**), protein biochemical and histological analyses especially of Lmp7-deficient GEnCs or human LMP7-deficient endothelial cells (see **Reviewer Fig. 13** and **14**). These and other analyses were not included to this manuscript, as the content of the manuscript is already quite dense as is.

A

B

Wrap73: SSX2IP:WRAP73 complex is proposed to act as regulator of spindle anchoring at the mitotic centrosome; may promote docking of RAB8A

Rlp: Rab effector playing a role in late endocytic transport to degradative compartments. Involved in the regulation of lysosomal morphology and distribution. Induces recruitment of dynein-dynactin motor complexes to Rab7A-containing late endosome and lysosome compartments. Promotes centripetal migration of phagosomes and the fusion of phagosomes with the late endosomes and lysosomes

Dynl1c: Dynein light chain Tctex-type 1: Acts as one of several non-catalytic accessory components of the cytoplasmic dynein 1 complex that are thought to be involved in linking dynein to cargos and to adapter proteins that regulate dynein function. Cytoplasmic dynein 1 acts as a motor for the intracellular retrograde motility of vesicles and organelles along microtubules.

Dynlr1b1: Dynein light chain roadblock-type 1: Cytoplasmic dynein 1 acts as a motor for the intracellular retrograde motility of vesicles and organelle

Fcrlb: Fc receptor-related protein Y (FcRY)

Ncln: Component of the multi-pass translocon (MPT) complex that mediates insertion of multi-pass membrane proteins into the lipid bilayer of membranes.

Tmem52b: category: membrane, extracellular exosome

Dlg2: Disks large homologue 2, Regulates surface expression of NMDA receptors

Sfrp1: secreted frizzled related protein 1: Has antiproliferative effects on vascular cells, in vitro and in vivo, and can induce, in vivo, an angiogenic response. In vascular cell cycle, delays the G1 phase and entry into the S phase.

Pla2g4f: Cytosolic phospholipase A2 zeta, Has calcium-dependent phospholipase and lysophospholipase activities with a potential role in membrane lipid remodeling and biosynthesis of lipid mediators

Ecrg4: Augurin: Probable hormone that may attenuate cell proliferation and induce senescence of oligodendrocyte and neural precursor cells in the central nervous system

P2ry12: P2Y purinoceptor 12: Receptor for ADP and ATP coupled to G-proteins that inhibit the adenylyl cyclase second messenger system. Required for normal platelet aggregation and blood coagulation

Negr1: Neuronal growth regulator 1: May be involved in cell-adhesion.

Clcnkb: Chloride channel protein CIC-Kb: Voltage-gated chloride channel. Chloride channels have several functions including the regulation of cell volume; membrane potential stabilization, signal transduction and transepithelial transport.

Slc12a3: Electroneutral sodium and chloride ion cotransporter

Fxyd2: Sodium/potassium-transporting ATPase subunit gamma

Reviewer Fig. 9: Bulk RNAseq of naïve GEnCs sorted from Lmp7 Δ EnC mice and control littermates. **(A)** Volcano plots exhibiting significantly down (blue) or upregulated (red) transcripts. **(B)** List of selected transcripts and their putative function based on www.uniport.org, demonstrating downregulation of i.e., essential proteins of the late endo-lysosomal system required for vesicle docking and trafficking/transport and upregulation of i.e., essential proteins involved in membrane remodeling, in the expression of plasma membrane proteins, in chloride permeability and cell swelling.

We are in the process of investigating the molecular (links) between the immunoproteasome and the altered endo-lysosomal pathway, however, identification of immunoproteasome

specific substrates in GEnC and/or podocyte has proven very challenging, and we are not expecting “one single clear link”. Therefore, we decided to focus the manuscript on providing a detailed and clear dissection of the proteasome system in the glomerular syncytium and by showing its functional relevance for glomerular filtration barrier integrity, especially in orchestrating endocytosis at the level of the turnover/trafficking of plasma membrane proteins involved in cargo endocytosis.

It is likely that a myriad of pathways are affected by proteasome inhibition, how do we know that immunoglobulin deposition is the most important one? How do we know that other pathways are not affected?

We completely agree with this assessment, as originally stated in the discussion: “We show that the proteasome influences GEnCs and podocyte endocytosis at multiple node points...”

We now added a further sentence to the discussion to clearly acknowledge this aspect: “Mechanistically, we demonstrate that the (immuno)proteasome orchestrates endocytic

turnover and thus clearance of proteins at the GFB. Of note, it is likely that a myriad of other pathways are also affected by proteasome inhibition.”

Reviewer Fig. 10: Comparison of expression levels of proteasome related transcripts in two different accessible human single cell RNAseq datasets. (A) Single cell transcripts of PC, MC, and GEnC derived from a published human glomerular single cell RNAseq dataset (from He, B. et al. Nat Commun 2021 Vol. 12 Issue 1 Pages 2141¹) or (B) derived from human kidney single cell RNAseq data accessible within the Kidney Precision Medicine Project (accessed 12/04/2023; <https://www.kpmp.org>) were analyzed for the expression of proteasome-related transcripts. Heatmaps depict the relative transcript levels of podocytes, glomerular endothelial and mesangial cells to total murine glomerular cell transcript levels (mouse) of the preparations or to normalized transcript levels of PC, MC, and GEnCs in the KPMP database.

Figure 1 is a good analysis of the proteasome system but the team should analyze gene expression changes in published isolated glomeruli and single cell data in mice and humans. In addition, they should include the Deubiquitinating Enzymes (DUBs and Shuttling factors: These are proteins like Rad23 and Dsk2 that help in the delivery of the ubiquitinated protein to the proteasome.

Changes in the proteasome system in the disease setting are frequently only observed on the protein level and not on the transcriptional level in the systems we work with (glomerular cells, neurons, and cardiomyocytes). To accommodate this reviewers' suggestion, we included detailed analyses of published naïve single cell data sets of murine and human glomerular cell types on the basic expression of proteasome subunits (new **Suppl. Fig. 3B** and **C**), of endo-lysosomal proteins (new **Suppl. Fig. 6B** and **C**), of phagocytosis related proteins (new **Suppl. Fig. 14B** and **C**) and included them side-by-side to the proteomic analyses. Transcript expression of DUBs and shuttling factors were also analyzed and included as new **Suppl. Fig. 4A-D** to the manuscript.

For human scRNAseq analyses we included reanalyses of available data from the Kidney Precision Medicine Project (accessed 12/04/2023; <https://www.kpmp.org>), as in this dataset the cell numbers obtained from 20 healthy patients were higher than in comparison to the human scRNAseq analyses from isolated human glomerular published by He, B. et al. Nat Commun 2021 Vol. 12 Issue 1 Pages 2141¹ (**Reviewer Fig. 10**). Further, the transcript data extracted from KPMP fit to our protein observations in human glomerular cells and to the protein and mRNA observations in mice, which was not the case for the human data set published by He, B. et al.¹ This could be the consequence of different single cell isolation approaches (less mechanically challenging in KPMP) as well as of the number of individual cells and patients included (more in KPMP data set).

Figure1F does not have a loading control.

As stated within the manuscript, including a loading control such as β -actin, tubulin, calreticulin, or GAPDH is not helpful in this experimental setup, as the different glomerular cell types do not physiologically express the same amount of these housekeepers. This is exemplarily shown in **Reviewer Fig. 11**. Hence, this problem was solved by loading the same number of PC, MC, and GEnCs, as established during FACS sorting within the individual cell sorts. We addressed this approach in the method section and clarified in the result text: "Glomerular cell types were loaded in a cell number adapted manner, as they did not express the same quantity of common housekeepers (**Suppl. Fig. 5**)". Additionally, **Reviewer Fig. 11** was included as a new **Suppl. Fig. 5** to the manuscript.

Reviewer Fig. 11: Immunoblot of β -actin, calreticulin and ponceau staining in bulk-isolated glomerular cells. PC = Podocytes, MC = mesangial cells, EC = glomerular endothelial cells. Equal cell numbers were loaded.

The lysosome and ALP system is a bit more complicated to analyze, but again would suggest unbiased single cell studies, isolated glomeruli. In addition, electron microscopical comparison would be needed to examine cell components.

As requested by this reviewer, we performed unbiased analyses of published naïve single cell data sets of murine and human glomerular cell types on the expression of endo-lysosomal transcripts (new **Suppl. Fig. 6B** and **C**) and of phagocytosis related transcripts (new **Suppl. Fig. 14B** and **C**) within glomerular cell types and included them side-by-side to the existing proteomic analyses.

Cell components of glomerular cells have been analyzed by EM under healthy conditions by many experts in the field.¹¹⁻¹³ Unfortunately, we don't feel that we would add significantly new findings to the already existing excellent ultrastructural analyses of the glomerulus.

The phenotype of the *Lmp7* mice is mild at best. How do these animals behave under stress or following kidney disease?

To answer this reviewers' question, *Lmp7* ^{Δ EnC} mice were challenged with nephrotoxic serum, which primarily targets the GBM and glomerular endothelial cells and serves as a model for rapid progressive GN (RPGN) (**Reviewer Fig. 12**). To this end, *Lmp7* ^{Δ EnC} mice exhibited an exacerbated disease course in comparison to control littermates. Glomerular endothelial cells were lost in *Lmp7* ^{Δ EnC} mice, clinically resulting in an increased protein loss to the urine and increased serum triglycerides in comparison to control littermates. Morphologically, the percentage of glomerular crescents was elevated and peri- and intra-glomerular matrix deposition (smooth muscle actin) exacerbated. The mechanisms behind this phenotype in *Lmp7* ^{Δ EnC} mice is currently being elaborated and is beyond the scope of this manuscript.

Reviewer Figure 12: *Lmp7^{ΔEnC}* mice exhibit an exacerbated disease course in the nephrotoxic nephritis model. **(A)** *Lmp7^{ΔEnC}* and control mice were induced with tamoxifen, urine was collected before induction, mice were injected with nephrotoxic serum on day 0, urine was collected on days -1, 2, 9 and 10. Mice were sacrificed on day 10. **(B)** Cell numbers of bulk isolated glomerular cells were compared. To assess disease severity urine albumin / creatinine ratio **(C)** as well as serum blood urea nitrogen (BUN, **D**) and serum triglycerides **(E)** were measured during disease development. The crescent formation was assessed **(F)** and measured **(G)** via PAS staining. High-resolution confocal images of Kidney injury marker 1 (Kim1, green) in relation to smooth muscle actin (red) and DNA (Hoechst, blue).

Similarly mice treated with a variety of inhibitors only seems to show minimal proteinuria, but no glomerulosclerosis or kidney function decline. These observations strongly argue against the importance of the mechanism in kidney disease development.

Thank you for this comment. We agree that the leakiness of the GFB does not reach a nephrotic range in epoxomicin-treated mice in comparison to for example our plethora of membranous nephropathy models.^{10,14-18} Nonetheless, proteinuria is not low, considering that

mg/mg albumin/creatinine are plotted and is higher than in other publications, where for example autophagy is impaired by ATG5 knockout in podocytes.⁴ Glomerulosclerosis is a condition that develops in the setting of severe glomerular injury which results in glomerular cell type loss, usually from loss of GEnCs and/or podocytes. The development of glomerulosclerosis cannot be realistically expected in the setting of the experimental approach in the present study, where mice were only exposed for 4 days to inhibitor treatment to assess the physiological involvement of the proteasome for glomerular filtration barrier function. Even in adriamycin nephritis, a model for focal segmental glomerulosclerosis, where podocytes are lost due to toxic injury, development of glomerulosclerosis necessitates at least 10-14 days after adriamycin injection.

The aim of the current study is to provide a first basic understanding of protein degradative principles in glomerular cells with focus on the proteasome system, a major homeostatic system which is involved in all kinds of cellular reactions to injury. We know from our various published and ongoing unpublished studies, that the proteasome system is heavily involved in glomerulonephritis. Continuing investigations by others and us are, however, hampered by lack of a basic understanding of this system in normal glomerular biology. We start to close the knowledge gap with this study.

REVIEWER 3:

Thank you for the opportunity to review this manuscript by Wiebke Sachs. The authors revealed that podocytes rely on the constitutive proteasome, while endothelial cells rely on the immunoproteasome, as demonstrated through the isolation of glomerular cell types. The article also highlights that proteasome inhibition leads to the accumulation of immunoglobulin in glomeruli and provides evidence that it decreases endocytosis activity, affecting recycling and turnover. While there are other papers discussing proteasome and glomerular cells, it is novel that the authors investigated the topic from the viewpoint of endocytosis. This evidence is crucial in understanding glomerular disease and proteasome function.

Thank you for the thorough and positive assessment of our study. We hope to have adequately addressed the raised issues in the following paragraphs.

However, there are some concerns and points that need to be corrected, listed below:

Major

#1 Figure 4 and Suppl Figure 3

The author administered the proteasome inhibitor systemically via intra-peritoneal injection instead of using a genetic model. Were there any symptoms such as diarrhea or weight loss that could have affected Cre or BUN? Regardless of the presence or absence of symptoms, the author should mention this in the study.

Thank you for this important comment. To this end, no obvious symptoms such as diarrhea or weight loss were discernable in the 4 days of proteasome inhibition (**Reviewer Fig. 13**). Mice were however less agile and exhibited dulled fur. This finding was included to the result section as follows: "No obvious symptoms such as diarrhea or weight loss were discernable in the 4 days of inhibitor treatment, mice with proteasome inhibition were, however, less agile and exhibited dulled fur."

Reviewer Fig. 13: Weight change after inhibitor treatment. Percent weight change after 4 consecutive days of inhibitor treatment.

Additionally, I am curious if the three types of cells were similarly affected by epoxomicin. To address this, I would suggest the authors conduct an experiment wherein they administer epoxomicin to the three types of primary cultured cells and evaluate the proteasome activity or ubiquitin accumulation.

As the ultrastructural analyses of **Fig. 4F** and **G** demonstrate, MCs are not affected by epoxomicin, whereas GEnCs and podocytes exhibit ultrastructural alterations. Additionally, transcript levels of selected genes (i.e., **Suppl. Fig. 15B, C**) or protein levels of selected proteins (i.e., **Suppl. Fig. 13F**) are differentially affected in the three glomerular cell types upon proteasome inhibition. Therefore, we are confident that a differential susceptibility to proteasome inhibition is present across glomerular cell types.

The proposed experiments in primary cultured glomerular cell types after bulk isolation will not help answer this question. As mentioned in the discussion, glomerular cell types alter their predominance of degradative systems when placed in culture. Additionally, primary culture of mesangial cells and glomerular endothelial cells following bulk-isolation is not successful in terms of generating enough healthy cells that survive in culture for subsequent biochemical investigations. This is the consequence of the mechanical challenging bulk-isolation procedure and the lack of adequate medium compositions and proliferative capacity of the highly differentiated glomerular cell types (especially podocytes and GEnCs).

#2 Supplemental 3 and 5

As p62, LC3, and Limp2 expression did not change between vehicle and epoxomicin, I understand that the authors concluded that lysosomes did not compensate for proteasome inhibition. However, since these expressions can be affected by both autophagy and lysosomes, I believe the authors should evaluate these molecules through the autophagy-lysosome flux. Furthermore, since the authors used glomerular lysates in Western blotting, the results do not solely reflect podocyte changes.

Confocal analyses demonstrate that of the three glomerular cell types, podocytes are the cell type with the most obvious induction in p62 and ubiquitin in epoxomicin treatment, compared to GEnCs and MCs, which show no expression changes of these two proteins. Limp2 expression on the other hand is not changed in podocytes in comparison to MC and GEnCs in the setting of epoxomicin treatment. Hence, integrating these histological expression findings with the respective quantitative immunoblot analyses of isolated glomeruli that indeed reflect total glomerular changes, we feel that one can suggest that the changes of abundance assessed by immunoblot are mostly derived from the podocyte compartment (p62 and ubiquitin) or from the MC and GEnC compartment (Limp2).

This reviewer correctly states that changes of especially p62, LC3 can be affected both by autophagy and lysosomes, as an intact ALP is required. We corrected our statement in the result section title as follows: "20S alteration results in intra- and extracellular glomerular protein accumulation, which is not prevented by the autophagosome lysosome pathway.". Further, the figure legend title of **Suppl. Fig. 10** was changed to: "Proteasome impairment in podocytes is not compensated by the autophagosome lysosome pathway."

Indeed, flux measurements are the gold standard to assess autophagosome formation and fusion of autophagosomes to lysosomes and would certainly be of interest. Unfortunately, ethical regulations in animal work are severe in Germany. We currently do not have the allowance to administer lysosomal inhibitors such as bafilomycin or leupeptin A in addition to proteasomal inhibitors in the same mouse. Application and receipt of such an allowance was attempted, but still has not been granted. As we are beyond the given rebuttal time of 3 months for this manuscript, we will not be able to perform these experiments. We hope that this reviewer will accept the changes in wording detailed above (and below in the next point) as a sufficient correction of the manuscript.

Generally, proteasome and autophagy-lysosome are considered to be related and can compensate for each other. To conclude that autophagy-lysosome did not compensate for

proteasome inhibition, more experiments are required. To evaluate lysosome-autophagy flux accurately, the authors should inhibit autophagic flux *in vivo* and assess how LC3II and p62 respond to both the vehicle and epoxomicin.

See last paragraph above regarding the timely feasibility of the suggested *in vivo* flux measurements. To adequately address the concerns of this reviewer, we included following statement to the result section: “Of note, our cumulative investigations indicate that proteostasis disturbances within podocytes of proteasome-inhibited mice were not compensated by the autophagosome-lysosome pathway (**Suppl. Figs. 8, 10**). A firm conclusion, however, necessitates additional *in vivo* flux measurements in proteasome inhibited mice.”

#3 Figure 6

Endocytosis can be influenced by acid-base conditions. Therefore, isn't it possible that the accumulation of ubiquitin and oxidized proteins, due to the inhibition of the proteasome, makes the intracellular environment more acidic, thereby affecting endocytosis?

Indeed, this is a possibility. Thank you for the suggestion. We included this possibility to the discussion as follows: “Of note, endocytosis can also be influenced by acid-base conditions i.e., downstream of PI3P generated by phosphatidylinositol 3 kinase catalytic subunit vps34, which plays a major role in podocyte endocytosis.¹⁹ Therefore, the accumulation of ubiquitin and oxidized proteins in proteasome inhibited cells, could render the intracellular environment more acidic, thereby affecting endocytosis.”

Additionally, vesicle marker abundances and localizations are changed i.e. in endothelial cells

Reviewer Fig. 14: LMP7 knockout was induced by CRISPR/Cas9 technology in human endothelial cells (Δ LMP7) and compared to parental control cells (Ctrl). **(A)** The immunopatterns of endo-lysosomal vesicle markers were assessed by confocal microscopy and show no major alterations for early endosomal antigen (EEA)1, RAB7, RAB11, or LAMP1 expression. **(B)** Protein abundance of EEA1, RAB7, RAB11 was assessed by immunoblotting, graphs depict densitometric quantification of 1 experiment, n = 3 per condition, pooled values are shown, the line indicates the mean.

in the setting of LMP7 deficiency, as seen in the new experiments performed in a human endothelial cell line with CRISPR/Cas9 mediated LMP7 deficiency (**Reviewer Fig. 14 and 15**). Whereas the abundance and localization of EEA+, LAMP1+, RAB7+, and RAB11+ vesicles were not strongly altered (**Reviewer Fig. 14**), we noted a decreased protein abundance of RAB5, LAMP2 and LIMP2 in LMP7-deficient human endothelial cells (**Reviewer Fig. 15B**), as well as individual changes in localization or size of respective vesicles by immunofluorescence (**Reviewer Fig. 15A**). Whereas RAB5+ vesicles were prominent at the cell border of LMP7-deficient human endothelial cells, LAMP2+ vesicles were reduced and “disorganized” within the cytoplasm of LMP7-deficient human endothelial cells. LIMP2+ vesicles were occasionally large and also appeared “disorganized”. The alteration of the late endo-lysosomal pathway is further substantiated by our bulk transcriptomic analyses (please refer to **Reviewer Fig. 9**). As these analyses are not complete yet and are turning out to be very high in data content and

most likely not pin-pointable to one mechanistic process, this aspect was not included, as we feel it is out of the scope of the current manuscript.

Reviewer Fig. 15: LMP7 knockout was induced by CRISPR/Cas9 technology in human endothelial cells (Δ LMP7) and compared to parental control cells (Ctrl). **(A)** The immunopatterns of endo-lysosomal vesicle markers were assessed by confocal microscopy and show alterations for late endo-lysosomal proteins including RAB5, LAMP2 or LIMP2 expression. **(B)** Protein abundance of these vesicular marker proteins was assessed by immunoblotting, graphs depict densitometric quantification of 1 experiment, $n = 3$ per condition, pooled values are shown, the line indicates the mean.

Additionally, in Figure 6A, why is the signal of transferrin at time 0 so high in the Epoxomicin group, while the signal is very low in the vehicle group? Transferrin uptake in the epoxomicin group appears to decrease over time. To analyze endocytosis under proteasome inhibition conditions, I would suggest the authors to attempt a pulse-chase experiment using transferrin-rhodamine, IgG-fluorescein. By comparing the uptake rates and patterns of each molecule, we should be able to gain a more detailed understanding of the kinetics of endocytosis within the cell.

Thank you for this comment, which we address in **Reviewer Fig. 16**. The experimental setup of the endocytosis assay of **Fig 6A** for the timepoint 0 samples was as follows: Primary podocytes derived from glomerular outgrowths were pretreated with and without proteasomal inhibition and synchronized on ice. Then endocytic cargo (biotin-transferrin and rblgG) was mixed into the medium. For timepoint 0, the medium with biotin-transferrin and rblgG was directly removed again, cells were washed, scraped, and pelleted for immunoblot analyses. Hence, the timepoint 0 cells did theoretically come into contact with biotin-transferrin and rblgG. To follow up on the correct observation that the signal for transferrin appears high in the epoxomicin group, we quantified the amount of biotin-transferrin and rblgG signal at time 0 min in the epoxomicin group in comparison to vehicle of all the biological replicates and independent experiments performed. Indeed, this quantification shows that transferrin levels in the epoxomicin group are significantly higher, whereas rblgG levels are (not significantly) elevated (**Reviewer Fig. 16A and B**). This leads us to speculate that the biotin-transferrin and rblgG might adhere better to the primary podocyte membrane in epoxomicin condition than in

the vehicle condition. This could be either through unspecific binding (“stickiness”) to the plasma membrane or due to higher levels of receptors that bind the biotin-transferrin / rblgG at the membrane or due to higher levels of receptors that bind the biotin-transferrin / rblgG at the membrane and then fail to internalize.

As suggested, we therefore performed pulse chase experiments with transferrin-488 and IgG-Cy3, which was evaluated by confocal microscopy. High-resolution confocal images show reduced amount of transferrin, rblgG uptake in epoxomicin treated cells in comparison to vehicle treated cells over the course of 60 min of substrate exposure (**Reviewer Fig. 16C**). Additionally, the FcRN did not translocate to or from the perinuclear region of the cells to the cell border in epoxomicin treated podocytes corroborating trafficking problems of this receptor. We could not observe an enhanced fluorescent signal to transferrin or rblgG at the plasma membrane at timepoint 0 in epoxomicin treated primary podocytes using this approach.

Reviewer Fig. 16: (A) Densitometric analysis of transferrin-biotin immunoblot of epoxomicin treated cells after 0 min of substrate exposure in relation to the amounts found in vehicle treated cells. (B) Densitometric analysis of rblgG immunoblot of epoxomicin treated cells after 0 min of substrate exposure in relation to the amounts found in vehicle treated cells. (C) High-resolution confocal images of vehicle and epoxomicin treated cells in a pulse-chase experiment over the course of 60 min of substrate exposure. Uptake of FITC-transferrin (green) and Cy3-rabbit IgG (rblgG, red) was monitored, FcRN (light blue) and synaptopodin (white) was detected after fixation with 4% PFA by indirect immunofluorescence. Synaptopodin identifies and outlines podocytes. Green and red arrows point towards endocytic cargo that was taken up. Note the presence of FcRN redistribution towards the cell border (dotted line) in the vehicle cell, that does not occur in the epoxomicin treated cell.

The fluorescent assay detects very specifically the endocytic activity of podocytes in the glomerular outgrowths, since only podocytes are assessed based on synaptopodin expression. We therefore moved the original immunoblot measurements of endocytic activity to the supplement, together with the timepoint *t0* quantifications of transferrin and rblgG abundance (new **Suppl. Fig. 12**). We included the fluorescent endocytic assay of rblgG and transferrin as a new **Fig. 6A**. The FcRN staining was not included in this figure, as it disrupts

the fluidity of the manuscript at that stage. The FcRN trafficking data were separately included as a new **Fig. 7C**. The qPCR analyses of Mrc2 expression were for this moved to **Suppl. Fig. 15C**. The result section and method section were changed accordingly.

Minor

#1 Figure 1C

The proteasome is expected to be localized in the nuclei more than in the cytoplasm. However, the staining of Beta5c (Psm5) shows a significantly higher amount of proteasome in the cytoplasm compared to the nucleus. This result differs slightly from the one in Nature Communications volume 14, Article number: 2114 (2023), Figure 7c, which I consider to be the correct stain.

Thank you for this comment. There is no experimental evidence to the best of our knowledge that should support the expectation that the proteasome system is localized more in nuclei than in the cytoplasm of podocytes. The subcellular proteasome localization is very likely to differ, depending on the cell type evaluated (i.e. neurons in the soma and post-synaptic densities, synapses²⁰) and of the metabolic status.²¹ Further, the predominant subcellular localization of the proteasome certainly depends on many factors such as transcriptional activity and cellular stress/age.

The antibody to $\beta 5c$ used in the above-mentioned Nat Commun article 2114¹⁰ and in the present manuscript is the same (same lot), as is the staining protocol. What differs is 1) the mouse background, which was C75BL/6 in article 2114 and in the present study is BALB/c and 2) the age of the mice used for experiments (younger in the current manuscript) and 3) the disease model shown in the above-mentioned Nat Commun article 2114¹⁰ is from a mouse treated with anti-podocyte nephritis serum (APN) while in the present study naïve mice are shown .

Of note, nuclear $\beta 5c$ expression can be appreciated in selected podocytes within other glomeruli (**Reviewer Fig. 17**), hence there seems to be a certain degree of physiologic dynamics in subcellular proteasome localization in podocytes. As the aim of **Fig. 1C** is to demonstrate the predominance of $\beta 5c$ expression in podocytes in comparison to MCs and GEnCs, we would at this point not want to stress and focus on the subcellular localization of $\beta 5c$.

Reviewer Fig. 17: High-resolution confocal images of $\beta 5c$ (green) in relation to nephrin (red) and DNA (blue) in healthy mouse (**A**) and human (**B**) kidney. Red arrows point to nuclear $\beta 5c$ localization and white arrows point at cytoplasmic $\beta 5c$.

#2 Suppl 3D

Why does the beta-Hexosaminidase activity increase in lysosomal inhibition by leupeptin A in suppl Fig3D? Doesn't leupeptin A inhibit Hexosaminidase activity?

Leupeptin A indeed inhibits hexosaminidase activity as a protease inhibitor that primarily inhibits lysosomal and some cysteine proteases. Leupeptin A forms a covalent bond with the active sites of the enzymes it inhibits. Binding however is reversible and competitive.²² Additionally Leupeptin A is not very stable *in vivo* and only has a short half-life.²³ The hexosaminidase activity shown in **Suppl. Fig. 8D** was assessed using an *ex vivo* lysate activity measurement assay in isolated glomeruli and was significantly increased in Leupeptin A treated mice. Currently, we can only formulate the following explanation for this clear finding: Lysosomal inhibition in mice resulted an induction of glomerular lysosomes (i.e., Limp2 **Suppl. Fig. 10A, B**) and of lysosomal enzyme abundance (i.e., Cathepsin D **Suppl. Fig. 10C**), hence potentially also of hexosaminidase. Dosage of Leupeptin A was certainly not maxed out to fully inhibit lysosomal enzymes in mice, including hexosaminidase. Therefore, overall glomerular hexosaminidase activity was increased in the *ex vivo* lysate assay of isolated glomeruli from Leupeptin A treated mice.

#3 Suppl Fig 6

Can the accumulated IgG lead to nephritis, or does it only accumulate in the glomerulus?

The strongest accumulation of mslgG (**Suppl. Fig. 11**) and rblgG (**Fig. 5**) was observed within the glomerulus. Within the timeframe of the performed experiments, no clinical or histologic signs of nephritis such as leukocyte infiltration were observed. This does, however, not exclude the possibility that GN could potentially ensue if subepithelial deposition persists. On the other hand, in mice with constitutive or podocyte-specific FcRN deficiency,^{24,25} nephritis does not ensue in unchallenged mice despite a substantial renal/glomerular IgG deposition.

LITERATURE

1. He, B., Chen, P., Zambrano, S., Dabaghie, D., Hu, Y., Moller-Hackbarth, K., Unnersjo-Jess, D., Korkut, G.G., Charrin, E., Jeansson, M., et al. (2021). Single-cell RNA sequencing reveals the mesangial identity and species diversity of glomerular cell transcriptomes. *Nat Commun* 12, 2141. 10.1038/s41467-021-22331-9.
2. Cox, J., Hein, M.Y., Lubner, C.A., Paron, I., Nagaraj, N., and Mann, M. (2014). Accurate proteome-wide label-free quantification by delayed normalization and maximal peptide ratio extraction, termed MaxLFQ. *Mol Cell Proteomics* 13, 2513-2526. 10.1074/mcp.M113.031591.
3. Schlondorff, D., and Banas, B. (2009). The mesangial cell revisited: no cell is an island. *J Am Soc Nephrol* 20, 1179-1187. 10.1681/ASN.2008050549.
4. Hartleben, B., Godel, M., Meyer-Schwesinger, C., Liu, S., Ulrich, T., Kobler, S., Wiech, T., Grahammer, F., Arnold, S.J., Lindenmeyer, M.T., et al. (2010). Autophagy influences glomerular disease susceptibility and maintains podocyte homeostasis in aging mice. *J Clin Invest* 120, 1084-1096. 10.1172/JCI39492.
5. Brinkkoetter, P.T., Bork, T., Salou, S., Liang, W., Mizi, A., Ozel, C., Koehler, S., Hagmann, H.H., Ising, C., Kuczkowski, A., et al. (2019). Anaerobic Glycolysis Maintains the Glomerular Filtration Barrier Independent of Mitochondrial Metabolism and Dynamics. *Cell Rep* 27, 1551-1566 e1555. 10.1016/j.celrep.2019.04.012.
6. Sachs, W., Sachs, M., Kruger, E., Zielinski, S., Kretz, O., Huber, T.B., Baranowsky, A., Westermann, L.M., Voltolini Velho, R., Ludwig, N.F., et al. (2020). Distinct Modes of Balancing Glomerular Cell Proteostasis in Mucopolysaccharidosis Type II and III Prevent Proteinuria. *J Am Soc Nephrol* 31, 1796-1814. 10.1681/ASN.2019090960.
7. Meyer-Schwesinger, C., Meyer, T.N., Munster, S., Klug, P., Saleem, M., Helmchen, U., and Stahl, R.A. (2009). A new role for the neuronal ubiquitin C-terminal hydrolase-L1 (UCH-L1) in podocyte process formation and podocyte injury in human glomerulopathies. *J Pathol* 217, 452-464. 10.1002/path.2446.

8. Meyer-Schwesinger, C., Meyer, T.N., Sievert, H., Hoxha, E., Sachs, M., Klupp, E.M., Munster, S., Balabanov, S., Carrier, L., Helmchen, U., et al. (2011). Ubiquitin C-terminal hydrolase-11 activity induces polyubiquitin accumulation in podocytes and increases proteinuria in rat membranous nephropathy. *Am J Pathol* 178, 2044-2057. 10.1016/j.ajpath.2011.01.017.
9. Beeken, M., Lindenmeyer, M.T., Blattner, S.M., Radon, V., Oh, J., Meyer, T.N., Hildebrand, D., Schluter, H., Reinicke, A.T., Knop, J.H., et al. (2014). Alterations in the ubiquitin proteasome system in persistent but not reversible proteinuric diseases. *J Am Soc Nephrol* 25, 2511-2525. 10.1681/ASN.2013050522.
10. Reichelt, J., Sachs, W., Frombling, S., Fehlert, J., Studencka-Turski, M., Betz, A., Loreth, D., Blume, L., Witt, S., Pohl, S., et al. (2023). Publisher Correction: Non-functional ubiquitin C-terminal hydrolase L1 drives podocyte injury through impairing proteasomes in autoimmune glomerulonephritis. *Nat Commun* 14, 2453. 10.1038/s41467-023-38206-0.
11. Mundel, P., and Kriz, W. (1995). Structure and function of podocytes: an update. *Anat Embryol (Berl)* 192, 385-397. 10.1007/BF00240371.
12. Schmidt, M.E., Abdelbaki, Y.Z., and Tu, A.T. (1976). Variations in kidney ultrastructure with a variety of paraformaldehyde fixation techniques. *Acta Anat (Basel)* 95, 468-479. 10.1159/000144634.
13. Zimny, M.L., and Levy, E.D., Jr. (1971). Ultrastructure of mesangial and juxtaglomerular cells in the kidney of a hibernator. *Z Zellforsch Mikrosk Anat* 118, 326-332. 10.1007/BF00331191.
14. Tomas, N.M., Meyer-Schwesinger, C., von Spiegel, H., Kotb, A.M., Zahner, G., Hoxha, E., Helmchen, U., Endlich, N., Koch-Nolte, F., and Stahl, R.A.K. (2017). A Heterologous Model of Thrombospondin Type 1 Domain-Containing 7A-Associated Membranous Nephropathy. *J Am Soc Nephrol* 28, 3262-3277. 10.1681/ASN.2017010030.
15. Meyer-Schwesinger, C., Tomas, N.M., Dehde, S., Seifert, L., Hermans-Borgmeyer, I., Wiech, T., Koch-Nolte, F., Huber, T.B., and Zahner, G. (2020). A novel mouse model of phospholipase A2 receptor 1-associated membranous nephropathy mimics podocyte injury in patients. *Kidney Int* 97, 913-919. 10.1016/j.kint.2019.10.022.
16. Tomas, N.M., Dehde, S., Meyer-Schwesinger, C., Huang, M., Hermans-Borgmeyer, I., Maybaum, J., Lucas, R., von der Heide, J.L., Kretz, O., Kollner, S.M.S., et al. (2022). Podocyte expression of human phospholipase A2 receptor 1 causes immune-mediated membranous nephropathy in mice. *Kidney Int*. 10.1016/j.kint.2022.09.008.
17. Seifert, L., Zahner, G., Meyer-Schwesinger, C., Hickstein, N., Dehde, S., Wulf, S., Kollner, S.M.S., Lucas, R., Kyllies, D., Froembling, S., et al. (2023). The classical pathway triggers pathogenic complement activation in membranous nephropathy. *Nat Commun* 14, 473. 10.1038/s41467-023-36068-0.
18. Tomas, N.M., Dehde, S., Meyer-Schwesinger, C., Huang, M., Hermans-Borgmeyer, I., Maybaum, J., Lucas, R., von der Heide, J.L., Kretz, O., Kollner, S.M.S., et al. (2023). Podocyte expression of human phospholipase A2 receptor 1 causes immune-mediated membranous nephropathy in mice. *Kidney Int* 103, 297-303. 10.1016/j.kint.2022.09.008.
19. Bechtel, W., Helmstadter, M., Balica, J., Hartleben, B., Kiefer, B., Hrnjic, F., Schell, C., Kretz, O., Liu, S., Geist, F., et al. (2013). Vps34 deficiency reveals the importance of endocytosis for podocyte homeostasis. *J Am Soc Nephrol* 24, 727-743. 10.1681/ASN.2012070700.
20. Ribeiro, F.C., Cozachenco, D., Heimfarth, L., Fortuna, J.T.S., de Freitas, G.B., de Sousa, J.M., Alves-Leon, S.V., Leite, R.E.P., Suemoto, C.K., Grinberg, L.T., et al. (2023). Synaptic proteasome is inhibited in Alzheimer's disease models and associates with memory impairment in mice. *Commun Biol* 6, 1127. 10.1038/s42003-023-05511-9.
21. Baumann, K. (2023). mTOR inhibits starvation-induced nuclear export of the proteasome. *Nat Rev Mol Cell Biol* 24, 854. 10.1038/s41580-023-00678-9.

22. Billinger, E., Viljanen, J., Lind, S.B., and Johansson, G. (2020). Inhibition properties of free and conjugated leupeptin analogues. *FEBS Open Bio* 10, 2605-2615. 10.1002/2211-5463.12994.
23. Maes, K., Testelmans, D., Powers, S., Decramer, M., and Gayan-Ramirez, G. (2007). Leupeptin inhibits ventilator-induced diaphragm dysfunction in rats. *Am J Respir Crit Care Med* 175, 1134-1138. 10.1164/rccm.200609-1342OC.
24. Blaine, J. (2021). Generating a Podocyte-Specific Neonatal F Receptor (FcRn) Knockout Mouse. *Methods Mol Biol* 2224, 123-132. 10.1007/978-1-0716-1008-4_9.
25. Akilesh, S., Huber, T.B., Wu, H., Wang, G., Hartleben, B., Kopp, J.B., Miner, J.H., Roopenian, D.C., Unanue, E.R., and Shaw, A.S. (2008). Podocytes use FcRn to clear IgG from the glomerular basement membrane. *Proc Natl Acad Sci U S A* 105, 967-972. 10.1073/pnas.0711515105.

Reviewer Fig 1:

Abundance of K48-polyubiquitin in bulk-isolated glomerular cells by immunoblotting. Equal cell numbers of podocytes (PC), mesangial cells (MC), and glomerular endothelial cells (EC) were loaded. Graphs exhibit relative densitometric analysis of cells as percentage of MC levels, mean \pm SEM, One-Way ANOVA, $n = 4 - 5$.

Reviewer Fig 2:

mouse kidney

human kidney

Ubiquitin
nephrin
DNA

↑ nuclear ubiquitin ↑ cytoplasmic ubiquitin pc = podocyte ec = endothelial cell
mc = mesangial cell

Distribution of ubiquitin (green) by high-resolution confocal images in a healthy human and murine glomerulus in relation to the slit diaphragm protein nephrin (red) and DNA (blue); pc = podocyte, mc = mesangial cell, ec = glomerular endothelial cell, white arrows point towards cytoplasmic ubiquitin and red arrows point towards nuclear ubiquitin.

Reviewer Fig 3:

Confocal images of ubiquitin (**A**), K48-polyubiquitin (K48pUB) (**B**), β5c (**C**) and β5i (**D**) distribution in a healthy murine and human glomerulus. All proteins of interest are shown in green in relation to the slit diaphragm protein nephrin (red) and DNA (blue). Lower panel rows depict the secondary antibody only negative controls (w/o 2Ab) to each staining condition. (**E**) Exemplary attempt to differentiate signals in glomerular endothelial cells *via* endomucin staining (light blue) and podocytes *via* synaptopodin staining (red). A GEnC mask is generated based on endomucin signal. Overlay of this GEnC mask over DNA staining demonstrates “failure” to integrate the cytoplasmic/nuclear fraction of GEnCs within the mask (problem 1). Overlay of the GEnC mask over the synaptopodin staining demonstrates inclusion of podocyte signals within the GEnC mask (problem 2). Together, these limitations falsify the histological GEnC measurements.

Supplementary Figure 3C

Comparison of protein abundance in murine bulk-isolated glomerular cell types using (A) proteomic label-free quantification (original Suppl. Fig. 3A) and (B) proteomic label-free quantification normalized to Psma1 of the structural proteasome 20S core particle; podocytes (PC), mesangial cells (MC) and glomerular endothelial cells (EC).

Reviewer Fig 5:

Abundance of the constitutive proteasome subunits $\beta 1c$, $\beta 5c$, of the immunoproteasome subunit $\beta 5i$ and of the 20S core complex subunit $\alpha 2$ in bulk-isolated glomerular cells by immunoblotting. Equal cell numbers of podocytes (PC), mesangial cells (MC), and glomerular endothelial cells (EC) were loaded. Graphs depict densitometric quantification of protein abundances normalized to MCs within individual experiments. Values are depicted as mean \pm SEM, pooled values of 4 independent experiments are shown for $\beta 5c$, $\beta 5i$ and $\alpha 2$, and of 1 experiment with $n=4$ biological samples for $\beta 1c$; * $p<0.05$, One Way ANOVA.

Reviewer Fig 6:

High-resolution confocal images of β 5i knockout was evaluated by high-resolution confocal microscopy to β 5i (green) in relation to nephrin (red) and DNA (blue) in glomeruli from *Lmp7 Δ EnC*. A negative control using secondary antibody alone (w/o 1° Ab), Cy2-rbIgG (green) and Cy3-gpIgG (red) shows absence of unspecific staining from the secondary antibodies used.

Male BALB/c mice were treated with either vehicle or ONX-0914 (10 mg/kg bodyweight) on four consecutive days. Urine was collected for the assessment of glomerular filtration barrier integrity and glomeruli were isolated for biochemical analyses. **(A)** In-gel activity using the pan-proteasomal activity-based probe MVB003 in isolated glomeruli of experimental mice. Graphs depict densitometric analysis of the prominent β 2c and β 5c/ β 5i activities in relation to α 2 amounts, values are shown as mean \pm SEM, percent of vehicle. **(B)** Albumin/creatinine ratio of vehicle and ONX-0914 treated mice after 4 days of treatment.

Reviewer Fig 8:

Representative confocal images in human biopsy specimens. LMP7 (green) expression is restricted to GEnCs (white arrows) in the healthy control biopsy. In anti-neutrophil cytoplasmic antibody (ANCA)+ glomerulonephritis (GN), LMP7 expression is additionally found in podocytes (red arrows) and in periglomerular leukocytes (yellow arrows). Nephrin (red) demarcates the glomerular filtration barrier, DNA is stained with Hoechst (blue).

A

B

Wrap73: SSX2IP:WRAP73 complex is proposed to act as regulator of spindle anchoring at the mitotic centrosome; may promote docking of RAB8A

Rilp: Rab effector playing a role in late endocytic transport to degradative compartments. Involved in the regulation of lysosomal morphology and distribution. Induces recruitment of dynein-dynactin motor complexes to Rab7A-containing late endosome and lysosome compartments. Promotes centripetal migration of phagosomes and the fusion of phagosomes with the late endosomes and lysosomes

Dynl1c: Dynein light chain Tctex-type 1: Acts as one of several non-catalytic accessory components of the cytoplasmic dynein 1 complex that are thought to be involved in linking dynein to cargos and to adapter proteins that regulate dynein function. Cytoplasmic dynein 1 acts as a motor for the intracellular retrograde motility of vesicles and organelles along microtubules.

Dynlr1b1: Dynein light chain roadblock-type 1: Cytoplasmic dynein 1 acts as a motor for the intracellular retrograde motility of vesicles and organelle

Fcrlb: Fc receptor-related protein Y (FcRY)

Ncln: Component of the multi-pass translocon (MPT) complex that mediates insertion of multi-pass membrane proteins into the lipid bilayer of membranes.

Tmem52b: category: membrane, extracellular exosome

Dlg2: Disks large homolog 2, Regulates surface expression of NMDA receptors

Sfrp1: secreted frizzled related protein 1: Has antiproliferative effects on vascular cells, in vitro and in vivo, and can induce, in vivo, an angiogenic response. In vascular cell cycle, delays the G1 phase and entry into the S phase.

Pla2g4f: Cytosolic phospholipase A2 zeta, Has calcium-dependent phospholipase and lysophospholipase activities with a potential role in membrane lipid remodeling and biosynthesis of lipid mediators

Ecr4: Augurin: Probable hormone that may attenuate cell proliferation and induce senescence of oligodendrocyte and neural precursor cells in the central nervous system

P2ry12: P2Y purinoreceptor 12: Receptor for ADP and ATP coupled to G-proteins that inhibit the adenylyl cyclase second messenger system. Required for normal platelet aggregation and blood coagulation

Negr1: Neuronal growth regulator 1: May be involved in cell-adhesion.

Clnkb: Chloride channel protein ClC-Kb: Voltage-gated chloride channel. Chloride channels have several functions including the regulation of cell volume; membrane potential stabilization, signal transduction and transepithelial transport.

Slc12a3: Electroneutral sodium and chloride ion cotransporter

Fxyd2: Sodium/potassium-transporting ATPase subunit gamma

Bulk RNAseq of naïve GEnCs sorted from Lmp7 Δ EnC mice and control littermates. (A) Volcano plots exhibiting significantly down (blue) or upregulated (red) transcripts. (B) List of selected transcripts and their putative function based on www.uniprot.org, demonstrating downregulation of i.e., essential proteins of the late endo-lysosomal system required for vesicle docking and trafficking/transport and upregulation of i.e., essential proteins involved in membrane remodeling, in the expression of plasma membrane proteins, in chloride permeability and cell swelling.

He, B. et al. Nat Commun 2021
Vol. 12 Issue 1 Pages 2141

Kidney Precision Medicine
Project (accessed 12/04/2023;
<https://www.kpmp.org>)

Comparison of expression levels of proteasome related transcripts in two different accessible human single cell RNAseq datasets. (A) Single cell transcripts of PC, MC, and GEnC derived from a published human glomerular single cell RNAseq dataset (from He, B. et al. Nat Commun 2021 Vol. 12 Issue 1 Pages 2141) or (B) derived from human kidney single cell RNAseq data accessible within the Kidney Precision Medicine Project (accessed 12/04/2023; <https://www.kpmp.org>) were analyzed for the expression of proteasome-related transcripts. Heatmaps depict the relative transcript levels of podocytes, glomerular endothelial and mesangial cells to total murine glomerular cell transcript levels (mouse) of the preparations or to normalized transcript levels of PC, MC, and GEnCs in the KPMP database.

Reviewer Fig 11:

Immunoblot of β -actin, calreticulin and ponceau staining in bulk-isolated glomerular cells. PC = Podocytes, MC = mesangial cells, EC = glomerular endothelial cells. Equal cell numbers were loaded.

Lmp7^{ΔEnC} mice exhibit an exacerbated disease course in the nephrotic nephritis model. (A) *Lmp7^{ΔEnC}* and control mice were induced with tamoxifen, urine was collected before induction, mice were injected with nephrotoxic serum on day 0, urine was collected on days -1, 2, 9 and 10. Mice were sacrifice on day 10. (B) Cell numbers of bulk isolated glomerular cells were compared. To assess disease severity urine albumin / creatinine ratio (C) as well as serum blood urea nitrogen (BUN, D) and serum triglycerides (E) were measured during disease development. The crescent formation was assessed (F) and measured (G) via PAS staining. High-resolution confocal images of Kidney injury marker 1 (Kim1, green) in relation to smooth muscle actin (red) and DNA (Hoechst, blue).

Reviewer Fig 13:

Weight change after inhibitor treatment. Percent weight change after 4 consecutive days of inhibitor treatment.

LMP7 knockout was induced by CRISPR/Cas9 technology in human endothelial cells (Δ LMP7) and compared to parental control cells (Ctrl). **(A)** The immunopatterns of endo-lysosomal vesicle markers were assessed by confocal microscopy and show no major alterations for early endosomal antigen (EEA)1, RAB7, RAB11 or LAMP1 expression. **(B)** Protein abundance of EEA1, RAB7, RAB11 was assessed by immunoblotting, graphs depict densitometric quantification of 1 experiment, $n = 3$ per condition, pooled values are shown, the line indicates the mean.

A

B

LMP7 knockout was induced by CRISPR/Cas9 technology in human endothelial cells (Δ LMP7) and compared to parental control cells (Ctrl). (A) The immunopatterns of endo-lysosomal vesicle markers were assessed by confocal microscopy and show no major alterations for early endosomal antigen RAB5, LAMP2 or LIMP1 expression. (B) Protein abundance of these vesicular marker proteins was assessed by immunoblotting, graphs depict densitometric quantification of 1 experiment, $n = 3$ per condition, pooled values are shown, the line indicates the mean.

(A) Densitometric analysis of transferrin-biotin immunoblot of epoxomycin treated cells after 0 min of substrate exposure in relation to the amounts found in vehicle treated cells. (B) Densitometric analysis of rblgG immunoblot of epoxomycin treated cells after 0 min of substrate exposure in relation to the amounts found in vehicle treated cells. (C) High-resolution confocal images of vehicle and epoxomycin treated cells in a pulse-chase experiment over the course of 60 min of substrate exposure. Uptake of FITC-transferrin (green) and Cy3-rabbit IgG (rblgG, red) was monitored, FcRN (light blue) and synaptopodin (white) was detected after fixation with 4%PFA by indirect immunofluorescence. Synaptopodin identifies and outlines podocytes. Green and red arrows point towards endocytic cargo that was taken up. Note the presence of FcRN redistribution towards the cell border (dotted line) in the vehicle cell, that does not occur in the epoxomycin treated cell.

A mouse kidney

B human kidney

↑ nuclear β5c

↑ cytoplasmic β5c

pc = podocyte

ec = endothelial cell

High-resolution confocal images of β5c (green) in relation to nephrin (red) and DNA (blue) in healthy mouse (A) and human (B) kidney. Red arrows point to nuclear β5c localization and white arrows point at cytoplasmic β5c.

REVIEWERS' COMMENTS

Reviewer #1 (Remarks to the Author):

Thank you for the detailed response addressing our comments with satisfaction. A few minor corrections are suggested:

Please note that Fig. 1F is still missing a loading control.

In addition, in the cell cycle analysis, it is not clear why are there several populations for different cell cycle stages. A canonical analysis should contain only G1, G2/M, and S phase (if relevant). I would recommend adding the flow cytometry data to the supplementary materials so the readers can view it as well.

FACS in the text should be used only when sorting is done, otherwise, flow cytometry is the accurate term.

Reviewer #2 (Remarks to the Author):

My concerns have been responded. The team has performed a large number of experiments to make this a wonderful manuscript.

Reviewer #3 (Remarks to the Author):

I have reviewed their point-by-point response and the revised manuscript. I think they conducted the suggested experiments within a limited timeframe.

Also, the revisions made to the manuscript addressed my concern regarding the compensation of autophagy-lysosome for proteasome deficiency.

I was satisfied with their opinion regarding the localization of proteasome in podocytes.

I support the publication of this paper as it is an important study in understanding the role of the proteasome in glomerular diseases.

Reviewer #1 (Remarks to the Author):

Thank you for the detailed response addressing our comments with satisfaction.

Thank you for your positive assessment of our manuscript and the helpful comments. We appreciate the time put into this.

A few minor corrections are suggested:

Please note that Fig. 1F is still missing a loading control.

Please excuse our oversight in this regard, we had addressed this for Reviewer 2 as follows:

As stated within the manuscript, including a loading control such as β -actin, tubulin, calreticulin, or GAPDH is not helpful in this experimental setup, as the different glomerular cell types do not physiologically express the same amount of these housekeepers. This is exemplarily shown in **Reviewer Fig. 9**. Hence, this problem was solved by loading the same number of PC, MC, and GEnCs, as established during FACS sorting within the individual cell sorts. We addressed this approach in the method section and clarified in the result text: "Glomerular cell types were loaded in a cell number adapted manner, as they did not express the same quantity of common housekeepers (**Suppl. Fig. 5**).". Additionally, **Reviewer Fig. 9** was included as a new **Suppl. Fig. 5** to the manuscript.

Reviewer Fig. 9: Immunoblot of β -actin, calreticulin and ponceau staining in bulk-isolated glomerular cells. PC = Podocytes, MC = mesangial cells, EC = glomerular endothelial cells. Equal cell numbers were loaded.

In addition, in the cell cycle analysis, it is not clear why are there several populations for different cell cycle stages. A canonical analysis should contain only G1, G2/M, and S phase (if relevant).

Thank you for your comment, we have adjusted the presentation of the cell cycle stages in the manuscript.

I would recommend adding the flow cytometry data to the supplementary materials so the readers can view it as well.

We have added the cell cycle flow cytometry data as Supplementary Figure 7

FACS in the text should be used only when sorting is done, otherwise, flow cytometry is the accurate term.

Thank you for this comment, we have changed FACS to flow cytometry.

Reviewer #2 (Remarks to the Author):

My concerns have been responded. The team has performed a large number of experiments to make this a wonderful manuscript.

Thank you for your positive assessment of our manuscript, as well as your work and input towards the manuscript.

Reviewer #3 (Remarks to the Author):

I have reviewed their point-by-point response and the revised manuscript. I think they conducted the suggested experiments within a limited timeframe.

Also, the revisions made to the manuscript addressed my concern regarding the compensation of autophagy-lysosome for proteasome deficiency. I was satisfied with their opinion regarding the localization of proteasome in podocytes. I support the publication of this paper as it is an important study in understanding the role of the proteasome in glomerular diseases.

Thank you for your positive assessment of our manuscript. We appreciate your time and helpful input towards our manuscript.